# Atypical integrative element with strand-biased circularization activity assists interspecies antimicrobial resistance gene transfer from *Vibrio alfacsensis*

**Lisa Nonaka**[1,2]*, **Michiaki Masuda**[1], **Hirokazu Yano**[3,4¤]*

**1** Department of Microbiology, Dokkyo Medical University School of Medicine, Mibu, Tochigi, Japan,
**2** Faculty of Human Life Sciences, Shokei University, Kumamoto, Japan, **3** Faculty of Life and Environmental Sciences, University of Tsukuba, Tsukuba, Japan, **4** Graduate School of Life Sciences, Tohoku University, Sendai, Japan

¤ Current address: Antimicrobial Resistance Research Center, National Institute of Infectious Diseases, Higashimurayama, Japan
* nonaka20@shokei-gakuen.ac.jp (LN); h-yano@niid.go.jp (HY)

**Data Availability Statement:** All DNA sequence files are available from DDBJ/NCBI/EMBL (AP024165, AP024166, AP024167, AP024168, AP019849, AP019850, AP019851, AP019852).

## Abstract

The exchange of antimicrobial resistance (AMR) genes between aquaculture and terrestrial microbial populations has emerged as a serious public health concern. However, the nature of the mobile genetic elements in marine bacteria is poorly documented. To gain insight into the genetic mechanisms underlying AMR gene transfer from marine bacteria, we mated a multidrug-resistant *Vibrio alfacsensis* strain with an *Escherichia coli* strain, and then determined the complete genome sequences of the donor and the transconjugant strains. Sequence analysis revealed a conjugative multidrug resistance plasmid in the donor strain, which was integrated into the chromosome of the recipient. The plasmid backbone in the transconjugant chromosome was flanked by two copies of a 7.1 kb unclassifiable integrative element harboring a β-lactamase gene. The 7.1 kb element and the previously reported element Tn*6283* share four coding sequences, two of which encode the catalytic R-H-R-Y motif of tyrosine recombinases. Polymerase chain reaction and sequencing experiments revealed that these elements generate a circular copy of one specific strand without leaving an empty site on the donor molecule, in contrast to the movement of integron gene cassettes or ICE/IMEs discovered to date. These elements are termed SEs (strand-biased circularizing integrative elements): SE-6945 (the 7.1 kb element) and SE-6283 (Tn*6283*). The copy number and location of SE-6945 in the chromosome affected the antibiotic resistance levels of the transconjugants. SEs were identified in the genomes of other *Vibrio* species. Overall, these results suggest that SEs are involved in the spread of AMR genes among marine bacteria.

The raw reads for strains 04Ya108, 04Ya249, and LN95 are available from Sequence Read Archive under accession numbers DRA011098, DRA008632, and DRA011762, respectively. Complete genome of transconjugant TJ249 is available at URL:https://doi.org/10.6084/m9.figshare.13332467).

**Funding:** This research is supported by JSPS KAKENHI 18K05790 (LN), 22K05790 (LN), research grant from Mishima Kaium Memorial Foundation (HY). The funders had no role in study design, data collection and analysis, decision to publish, or preparation of the manuscript.

**Competing interests:** The authors have declared that no competing interests exist.

## Introduction

Antimicrobials have been used globally in aquaculture to control fish diseases. Although this approach helps to maintain a stable supply of aquacultural products, misuse of antimicrobials has led to the emergence of antimicrobial-resistant microbes and the accumulation of resistance genes at aquaculture sites [1, 2]. As the spread of multidrug-resistant (MDR) bacteria and pan-drug-resistant bacteria threatens human life [3, 4], it is important to obtain clues about whether and how aquatic and terrestrial microbial populations exchange genetic materials.

Conjugative plasmids [5] and integrative and conjugative elements (ICE) [6] are DNA units that can move from one cell to another through the conjugation machinery that they encode. They can directly move between species [7, 8], and indirectly mobilize genes on specific mobile elements [9–11]. Plasmids can be classified into families according to the replicon type or the mobilization machinery type for epidemiological purposes [12, 13].

Transposons have been defined as specific DNA segments that can repeatedly insert into one or more sites in one or more genomes [14]. Several distinct strand exchange mechanisms and the routes used for their location changes, i.e., *transposition*, have been discovered to date [15]. In a comprehensive review by Curcio and Derbyshire [15], transposons were classified into five families based on the DNA strand exchange enzymes used. In prokaryotes, the mode of transposition was largely classified into 'cut-out paste-in' (often called 'excision integration' for the ICEs [6]), 'copy-out paste-in,' or 'copy-in' for which replication is necessary to complete transposition; e.g., Tn3, IS91. DNA insertion involving DNA replication can also be seen in the integration of integron gene cassettes into *attI* [16] and the phage CTX into *dif* [17]; both use tyrosine recombinase(s) as the strand exchange enzyme. In transposon classification, ICEs are classified as transposons that use a Y-transposase (tyrosine recombinase) or an S-transposase (serine recombinase) [15]. Other than ICEs, DNA segments that can move to a location in other cells via the excision–transfer (rolling-circle replication)–integration route have been termed IME (integrative mobilizable element) [18], MGI (mobilizable genomic island) [19], CIME (*cis*-mobilizable element) [20], or MTn (mobilizable transposon) [21] (hereafter collectively referred to as IME). IMEs encode site-specific recombinase or DDE transposase as strand exchange enzyme [22]. They have been recognized as degraded forms of ICEs [20]. ICEs/IMEs so far studied in the laboratory seem to change their location via the 'excision integration' route only [23–26]. Genes embedded in these mobile genetic elements are readily shared among bacteria and provide genetic resources for microbial adaptation in changing environments.

To better understand the genetic mechanisms underlying the spread of antimicrobial resistance (AMR) genes at aquaculture sites, we previously collected resistant bacteria from sediments at an aquaculture site in Kagawa, Japan [27–29]. We identified a self-transmissible MDR plasmid of the MOB$_H$-family, named pAQU1, that can replicate in both the original host *Photobacterium damselae* subsp. *damselae* and in *Escherichia coli* [28]. Subsequently, another MOB$_H$-family MDR plasmid, named pSEA1, was identified in a *Vibrio alfacsensis* isolate (previously identified as *V. ponticus*) [30]. pSEA1 carries a 12 kb unclassifiable integrative element, Tn6283, in addition to the AMR genes. Although pSEA1 could not replicate in *E. coli* at 42˚C, following conjugation, it could cointegrate with the *E. coli* chromosome by homologous recombination between two Tn6283 copies: one on pSEA1 and another that moved from pSEA1 into the chromosome [30]. Further, Tn6283 was suggested to circularize one specific strand without producing an empty site, presumably using tyrosine recombinases. However, it is not known whether interspecies gene transfer assisted by Tn6283-like elements is common in nature.

To obtain further insights into the mechanisms behind genetic exchange among aquaculture-associated bacteria, in this study we mated another MDR *Vibrio* isolate, *V. alfacsensis* 04Ya249, with *E. coli* strains in the laboratory and then determined the genome sequences of both the donor and transconjugant strains. We identified a Tn*6283*-like element involved in the cointegration of the multidrug resistance plasmid and the *E. coli* chromosome. Based on the findings in the previous study and this study, we have termed Tn*6283* and a related element as a new class of mobile elements: <u>s</u>trand-biased circularizing integrative <u>e</u>lement (SE). Tn*6283* is hereafter referred to as SE-6283.

In this study, several types of intracellular DNA movement need to be discussed. For clarity, the terms *transposition*, *cointegration*, and *integration* are used here based on the following rules. We call the movement of a DNA segment from one location to the other location in one or two genomes, mediated by any DNA strand exchange enzymes, transposition. The physical incorporation of a plasmid into a chromosome is called cointegration regardless of the molecular mechanisms and genomic locations involved. The physical incorporation of a DNA molecule into a specific site in another molecule mediated by a site-specific recombinase (so-called integrase) is called integration.

## Results

### Identification of SE carrying a β-lactamase gene

We previously isolated *V. alfacsensis* strain 04Ya249 from sea sediment at an aquaculture site [27] (Table 1). 04Ya249 showed resistance to erythromycin, tetracycline, and ampicillin [31]. To identify active mobile elements in this strain that are relevant to its AMR, the strain was mated with macrolide-sensitive *E. coli* JW0452, and transconjugants were selected on erythromycin at 42˚C. One transconjugant was named strain TJ249.

The complete genome sequences of strains 04Ya249 and TJ249 were determined using the PacBio RS II platform. The genome of strain 04Ya249 consists of four replicons: two chromosomes, a 306 kb putative conjugative plasmid pSEA2, and an 82 kb putative conjugative plasmid pVA249. Strain 04Ya249 has a very similar genome architecture to the database strain *V. alfacsensis* CAIM 1831 based on gene synteny (S1A Fig) and average nucleotide identity (S1B Fig). Seven AMR genes were identified on pSEA2 (S2A Fig). One β-lactamase gene (60% product identity to $bla_{VHH-1}$ in the CARD database [32]) was located within a 7.1-kb repeat region found in both chromosome 1 and pSEA2 (S2B Fig). This β-lactamase gene has been assigned a new family name, $bla_{GMA}$ (<u>G</u>ammaproteobacterial <u>M</u>obile Class <u>A</u> β-lactamase), and allele name, $bla_{GMA-1}$, in NCBI.

The chromosome of *E. coli* TJ249 contains two notable insertions (Fig 1A), a smaller insertion of the 7.1-kb element alone immediately downstream of the putative transcription termination site for *yjjN* (*yjjNt*) located between *yjjN* and *yjjO* (in GenBank accession no. U00096.3), and a larger insertion into *insJ* of IS*3*, containing a nearly complete copy of pSEA2 and an additional copy of the 7.1 kb element. These indicate the movement between different genomic locations (pSEA2, *yjjNt*, *insJ*), thereby transposition, of the 7.1 kb element in *E. coli*. Thus, the 7.1 kb element was labelled a transposon and assigned a four digit identifier, 6945, by the transposon registry [33]. As the 7.1-kb element was confirmed to generate a circular copy of one specific strand without leaving an empty site like SE-6283 in this study (see below), the 7.1 kb element was regarded as a new SE and designated SE-6945.

As pSEA2 showed > 99.9% nucleotide identity in the aligned region to pSEA1 (accession no. LC081338.1) from strain 04Ya108 [30], we also determined the complete genome of strain 04Ya108 for comparison with strain 04Ya249. As predicted from Southern hybridization [30], strain 04Ya108 possessed two copies of SE-6283, one in chromosome 1 and the other in

**Table 1. Strains and plasmids used.**

| Strains or plasmids | Relevant genotype, phenotype, descriptions[a] | Reference |
|---|---|---|
| *Vibrio alfacsensis* | | |
| 04Ya249 | Aquaculture isolate harboring conjugative plasmid pSEA2; Tc[r], Ap[r], Ery[r] Cm[r] | [27] |
| 04Ya108 | Aquaculture isolate harboring conjugative plasmid pSEA1; Tc[r], Ap[r], Ery[r] Cm[r] | [27] |
| LN95 | pSEA2 free derivative of strain 04Ya249; Ap[r] | This study |
| *E. coli* | | |
| DH5α | F[-], Φ80d*lacZ*ΔM15, Δ(*lacZYA-argF*)U169, *deoR*, *recA*1, *endA*1, *hsdR*17(r$_K$[-], m$_K$[+]), *phoA*, *supE*44, λ[-], *thi*-1, *gyrA*96, *relA*1 | Takara Bio Inc., Shiga, Japan |
| JW0452 | λ- *rph*-1 Δ(*rhaD-rhaB*)568 *hsdR*514 Δ*lacZ*4787(::*rrnB*-3) Δ(*araD-araB)*567 Δ*acrA*748::*kan*; Km[r] | [68] |
| JW0452rif | Spontaneous rifampicin-resistant mutant of JW0452 | This study |
| JW0452Δ*recA* | JW0452Δ*recA*::*cat*; Km[r], Cm[r] | This study |
| JW0452Δ*recA*rif | Spontaneous rif resistant mutant of JW0452Δ*recA*; Km[r], Cm[r], Rif[r] | This study |
| Transconjugants[c] | | |
| TJ249 | JW0452 *insJ*::SE-6945::pSEA2 *yjjNt*::SE-6945; Ap[r] Tc[r] Ery[r] Cm[r] | This study |
| LN28, LN32, LN36, LN86 | JW0452 chr::pSEA2[b]; Ap[r] Tc[r] Ery[r] Cm[r] | This study |
| LN29, LN30, LN82, LN88, LN91 | JW0452 *yjjNt*::SE-6945::pSEA2; Ap[r] Tc[r] Ery[r] Cm[r] | This study |
| LN33, LN34, LN35, LN37, LN58, LN84, LN89, LN90 | JW0452 *insJ*::SE-6945::pSEA2; Ap[r] Tc[r] Ery[r] Cm[r] | This study |
| LN52 | JW0452 *yjjNt*::SE-6945; Ap[r] | This study |
| LN52rif | Spontaneous rifampicin-resistant mutant of LN52; Rif[r], Ap[r] | This study |
| Plasmids | | |
| pGEM-T | Cloning vector; pUC replicon, Ap[r] | Promega, Madison, WI, USA |
| pGEM-attS | pGEM-T carrying the 206 bp *attS* fragment; pUC replicon, Ap[r] | This study |
| pGEM-int | pGEM-T carrying the 103 bp *intA* fragment; pUC replicon, Ap[r] | This study |
| pGEM-gyrB | pGEM-T carrying the 110 bp *gyrB* fragment from strain 04Ya249; pUC replicon, Ap[r] | This study |
| pKD46 | Lambda-Red recombinase expression plasmid; pSC101ts replicon, Ap[r] | [69] |
| pKD3 | Donor of the *cat* gene for the Lambda-Red recombination experiment; R6K replicon, Cm[r] | [69] |

*a*. Antimicrobial resistance phenotypes ([r]) listed were confirmed by experiments. Abbreviations are as follows: Ap, ampicillin; Tc, tetracycline: Ery, erythromycin; Cm, chloramphenicol; Rif, rifampicin; *kan*, kanamycin resistance gene. *yjjNt* denotes the putative transcription termination region for the *yjjN* gene. It is unclear whether the resistance genes on pSEA2 are still in the original locations for other transconjugants besides TJ249.

*b*. Cointegration of plasmid pSEA2 with chromosome, independent of SE-6945 transposition. Location of pSEA2 in the transconjugant genome could not be defined based on Southern hybridization results.

*c*. The 19 transconjugants listed (TJ249, LN28, LN29, LN30, LN32, LN33, LN34, LN35, LN36, LN37, LN52, LN58, LN82, LN84, LN86, LN88, LN89, LN90, LN91) were obtained from 19 independent mating experiments. Selection was done at 42˚C. Antibiotic selection used was as follows: Tc for LN28, LN29 LN30, LN32, LN33, LN34, LN35, LN36, and LN37; Ap for LN52; Ap Tc for LN58 LN82, LN84, LN86, LN88, LN89, LN90, and LN91; Ery for TJ249.

plasmid pSEA1. Strain 04Ya108 also carries two copies of SE-6945, in the same locations on the chromosome and plasmid as strain 04Ya249 (S1A and S2B Figs). The four coding sequences in SE-6945 showed homology to the four coding sequences clustered at one end of SE-6283 (Fig 1B). The biochemical functions of the products of the four genes were predicted using Pfam and InterProScan in EMBL. The *intA* and *intB* products generated a hit on phage_integrase (CL0382) in Pfam and were further aligned with known tyrosine recombinases using structure-aware alignment program PROMAL3D [34] to confirm the presence of conserved motifs. The *intA* and *intB* products contained the catalytic R-H-R-Y motif (G-R motif in box I and H-R-H-Y motif in box II (Fig 1C)) conserved in tyrosine recombinases [35], though the N-terminal region of IntB is extraordinarily long among tyrosine recombinases.

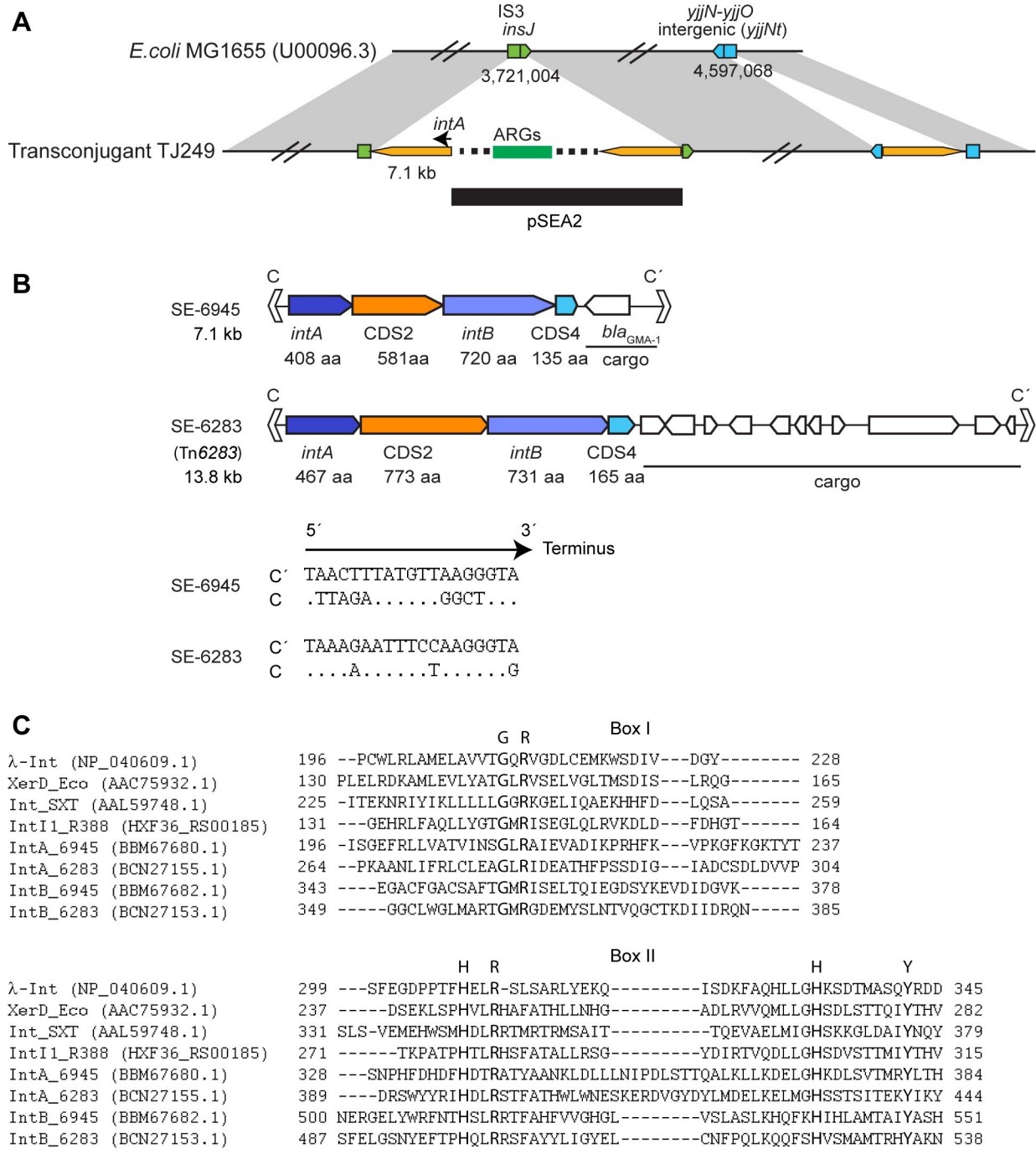

**Fig 1. Genetic features of SEs.** (A) Schematic representation of the 7.1 kb element and pSEA2 insertion locations in *E. coli* TJ249. Yellow pentagons are the 7.1 kb element. The sequence from pSEA2 is indicated by a horizontal line. (B) Upper panel: Genetic organization of SE-6283 and SE-6945. Lower panel: Nucleotide sequences of imperfect inverted repeat motifs C and C'. Dots indicate that bases in C are identical to those in C'. (C) Presence of R-H-R-Y motif in IntA and IntB. IntA and IntB sequences are aligned with sequences of known tyrosine recombinases based on secondary structure prediction using PROMAL 3D [34]. The aligned regions corresponding to box I and box II in [35] are shown.

CDS2 and CDS4 products did not generate hits in the Pfam or InterProScan databases. SE-6283 and SE-6945 carry imperfect 19 bp inverted repeat motifs at their ends (Fig 1B), termed C and C' based apparent on functional equivalence to the core-type sites of phages [36]. Gene

product identities between the SE-6945 and SE-6283 counterparts were 22.5% for IntA, 20.5% for CDS2, 23.2% for IntB, and 26.6% for CDS4 in blastp.

The border regions between SE-6945 and its target locations were termed *attL* (*intA*-proximal side) and *attR* (*intA*-distal side) (Fig 2A). The nucleotide sequence alignments of *attL* and *attR* from the two *E. coli* chromosomal locations and the two locations in 04Ya249 genomes (pSEA2 and chr 1) are shown in S3 and S4 Figs, respectively. Sequence comparison of the *attL*, *attR*, and unoccupied target site in the host (*attB*) in *E. coli* (Fig 2B) revealed that the 6 bp next to the 3′ end of C′ in *attR* in *E. coli* (5′-TTTTCT-3′) was identical to the 6 bp next to the 5′ end of C at *attL* on pSEA2, while the 6 bp next to the 5′ end of C in *attL* (5′-TTCTTT-3′ in *insJ*, 5′-TTTTTT-3′ in *yjjNt*) was from *attB* (Figs 2B and S3 and S4). Therefore, SE-6945 inserts itself, ending with 5′-GTA-3′ (Fig 1B) along with an additional 6 bp next to the 5′ end of C at *attL* on the donor molecule into the target site. The 6 bp from the donor molecule is placed next to the 3′ end of C′ in the newly formed *attR*.

**Target site preference of SE-6945 and SE-6283.**    After the isolation and genome sequence determination of TJ249, additional mating experiments between *E. coli* JW0452 and *V. alfacsensis* 04Ya249 were performed to test whether pSEA2 insertion into a specific location in the *E. coli* chromosome accompanying SE-6945 transposition is reproducible. Independent mating was performed more than 17 times, and transconjugants were screened on tetracycline (Tc) alone, ampicillin (Ap) alone, or both Ap and Tc. The SE-6945 integration pattern was investigated by Southern hybridization for 16 randomly chosen Tc resistant transconjugants, and one Tc sensitive but Ap-resistant transconjugant obtained from selection with Ap alone (S5C Fig). Among 16 Tc resistant transconjugants, 12 strains had *intA* at the *yjjNt*-SE-6945 junction or the *insJ*-SE-6945 junction. These 16 strains also generated pSEA2-derived fragments (S5C Fig). The remaining four strains generated pSEA2-derived fragments and contained only one copy of *intA*, suggesting SE-6945-independent pSEA2 cointegration with the chromosome; autonomous replication of pSEA2 seems unlikely given the high frequency of pSEA2 cointegration with the chromosome in other transconjugants. Strain LN52, obtained from selection with Ap alone, carried only one SE-6945 copy, and it was integrated into *yjjNt* (S5A and S5C Fig). Taken together, these observations suggest that pSEA2 integration relies on SE-6945 transposition, and that SE-6945 prefers *yjjNt* or *insJ* of IS*3*, rather than random sites in the *E. coli* genome.

To obtain further insights into the target site of SE-6945, bacterial genomes carrying SE-6945 were searched for based on blastn in the NCBI nucleotide collection (nr/nt) database using the SE-6945 core gene region (*intA*-CDS2-*intB*-CDS4) as the query. The genome of *Vibrio harveyi* strain WXL538 contains a segment showing 88.9% nucleotide sequence identity to SE-6945 core genes. Genome structure comparisons between WXL538 and 04Ya249 revealed that WXL538 carries a 16.4 kb SE-6945-like SE (provisionally termed SE-VhaWXL538) containing a different set of cargo genes (Fig 2A). SE-VhaWXL538 is integrated into the promoter region of a *yaaA* gene encoding an $H_2O_2$_YaaD_Superfamily protein (NCBI conserved domain accession no. cl01143) (locus_tag F9277_14400 in CP045070.1). This location corresponds to the SE-6945 insertion location in *V. alfacsensis* chromosome 1 (Fig 2A). Sequences of target sites (*attB*) in *E. coli*, *V. alfacsensis*, and *V. harveyi* are shown in Fig 2B. The conservation level of nucleotides in *attB* is presented in Fig 2C as sequence logo information contents [37]. The central 6 bp putative crossover region in *attB* is T rich. However, other motifs, such as imperfect inverted repeat structures observed in *attC/attI* of integrons (IntI target) [38, 39], or *dif* of CTX phage integration site (XerC/XerD target) [17], could not be detected in *attB*, *attL* (S3 Fig), or *attR* (S4 Fig) for SE-6945 in the current data set.

We previously reported SE-6283 integration into the *bcp* gene encoding peroxiredoxin (locus_tag b2480) in *E. coli*. SE-6283 also carries a *bcp* homolog (locus_tag VYA_19800 in AP024165.1) as if it complements the disrupted *bcp* gene in the host genome upon insertion

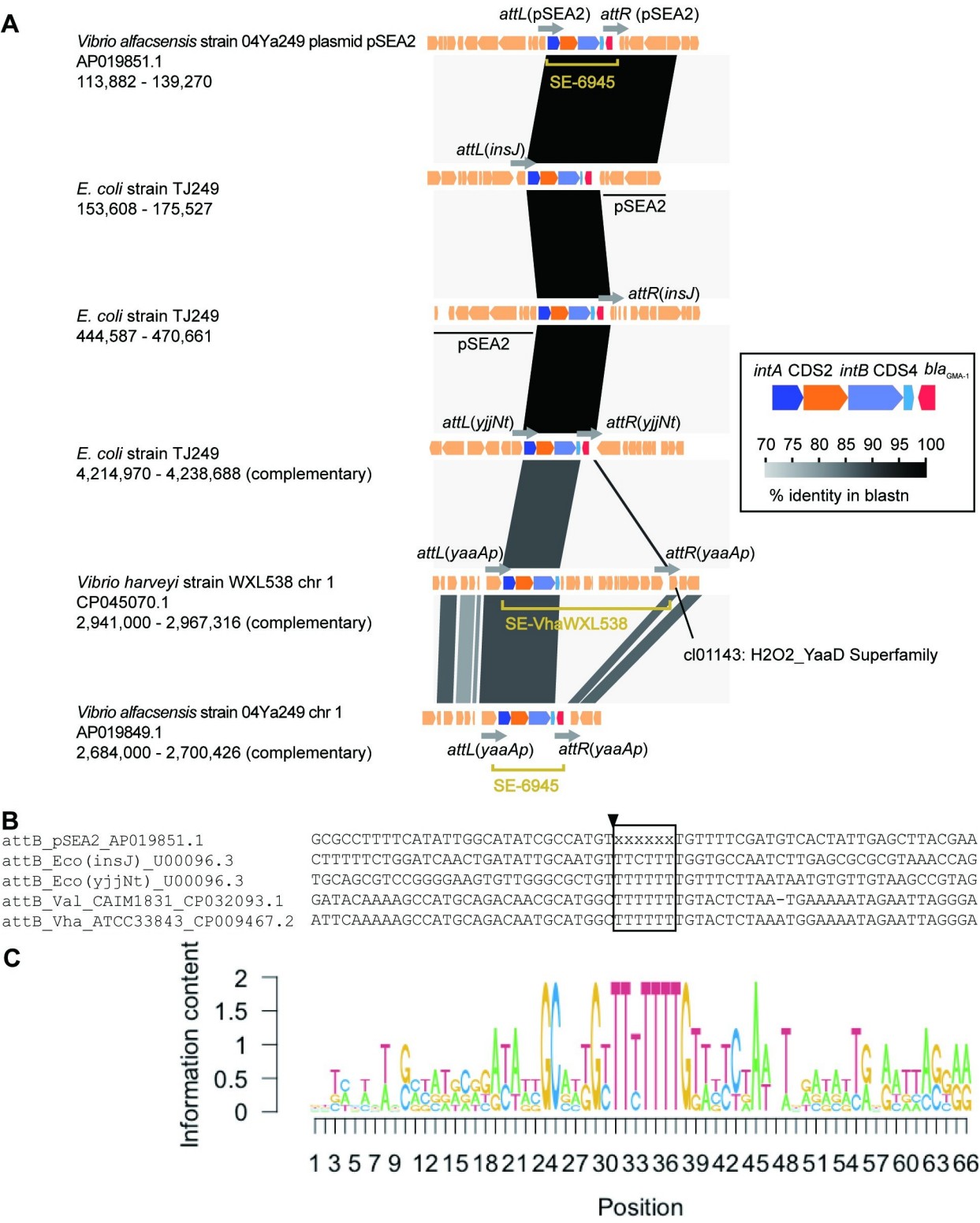

**Fig 2. Locations of SE-6945 and its related elements.** (A) Comparisons of genetic organization around SE insertion locations. GenBank accession numbers (*Vibrio* strains only) and the coordinates of genomic segments compared are shown below the strain name. (B) Alignment of *attB* sequences. The sequences are retrieved from SE-free genomes. Since the SE-6945-free pSEA2 ancestor could not be found in the database, the bases in the putative crossover region could not be defined and are shown as x. Nicking is expected to take place next to the central box: position is shown by black arrowhead. (C) Sequence logo representation of conservation levels of *attB* sequence.

[30]. Genome sequencing of strain 04Ya108 in this study revealed that one SE-6283 copy was integrated into a *bcp* gene homolog in chromosome 1 (Fig 3A). To obtain further insight into the target site preference of SE-6283, SE-6283-like elements in public databases were screened for using blastn. SE-6283-like SEs were identified in three other *Vibrio* species (Fig 3A). The sizes of SE-6283 and related SEs ranged from 13.8 kb (SE-6283) to 41.3 kb (SE-ValK09K1 in Fig 3A). Although the motifs C and C′ of the three SE-6283-like elements were slightly different from those of SE-6283 (see *attL* in S6 and *attR* in S7 Figs), all are integrated into *bcp* orthologs in chromosome 1: chromosome 1 (or I) of *V. alginolyticus* strain K09K1 is incorrectly named as chromosome II in CP017919. Comparison of *attL*, *attR*, and unoccupied *attB* found in the SEs-free genome of each *Vibrio* species suggests the integration of SE-6283-like elements and the 6 bp sequence into the central 6 bp putative crossover region in *attB* (Fig 3B). Nucleotide sequences around putative nicking sites in *attB* for SE-6283 are G and A-rich and, like *attB* of SE-6945 did not possess an inverted repeat structure (Fig 3C). These observations collectively suggest that, like ICEs, SEs target a few selected locations.

## Transposition of SE-6945 aids pSEA2-chromosome cointegration

The direct repeat structure of SE-6945 in the TJ249 chromosome suggested that pSEA2 was integrated into the recipient chromosome through the transposition of SE-6945, with subsequent cointegration of pSEA2 with the chromosome by homologous recombination between SE-6945 copies: one on the chromosome and the other on pSEA2 (referred to as a two-step gene transfer mechanism in the previous study [30]). If this scenario holds, integration of SE-6945 alone should occur more frequently than pSEA2-chromosome cointegration upon mating, and the cointegration would be reduced in frequency or abolished altogether in a *recA*-null mutant recipient. To test these possibilities, we conducted additional mating assays.

Quantitative mating assays were performed using a rifampicin-resistant recipient strain, JW0452rif, and its isogenic strain carrying *recA* deletion JW0452Δ*recA*rif. Note that Tc resistance is on pSEA2, while Ap resistance is on SE-6945. The transfer of Tc Ap resistance from *V. alfacsensis* 04Ya249 to *E. coli* JW0452rif was detected only in 2 out of 4 replicated mating experiments, and the transfer frequency was determined to be approximately $10^{-9}$ per donor, indicating a very rare event, generating 0–3 transconjugant colonies per mating (Fig 4). The transfer of Ap resistance alone was observed at >300-fold higher frequency than the transfer of both Tc and Ap resistance (compare the first and the second groups from left in Fig 4), indicating that integration of SE-6945 alone occurs more frequently than pSEA2-chromosome cointegration during mating.

The transfer of Tc Ap resistance to JW0452Δ*recA*rif could not be detected in any of the four independent mating assays (below detection limit of $1.02 \times 10^{-9}$ per donor). Unexpectedly the mean transfer frequency of Ap resistance to JW0452Δ*recA*rif was about 17-fold lower than that to JW0452rif (compare the second and the fourth groups from the left in Fig 4). Thus, the direct effect of *recA* knockout on the cointegration could not be evaluated due to the reduced detection sensitivity of gene transfer. However, the mean transfer frequency of Tc Ap resistance to LN52rif (which already carries one copy of SE-6945 and wild type *recA*) was about 56-fold higher than that to JW0452rif (compare the first and fifth groups in Fig 4). This indicates that the presence of the 7.1 kb long identical sequence in both target DNA and pSEA2 promotes cointegration of pSEA2 and the recipient chromosome.

Tn*3*-like cointegrate formation [15] might happen if SE-6945 could simultaneously introduce nicks at one site on the top strand at *attL* (or *attR*) and one site on the bottom strand at *attR* (or *attL*), and DNA-protein crosslink (DPC) repair [40] could clean the linked tyrosine recombinase to produce a 3′-OH priming site. If SE-mediated cointegrate formation occurs,

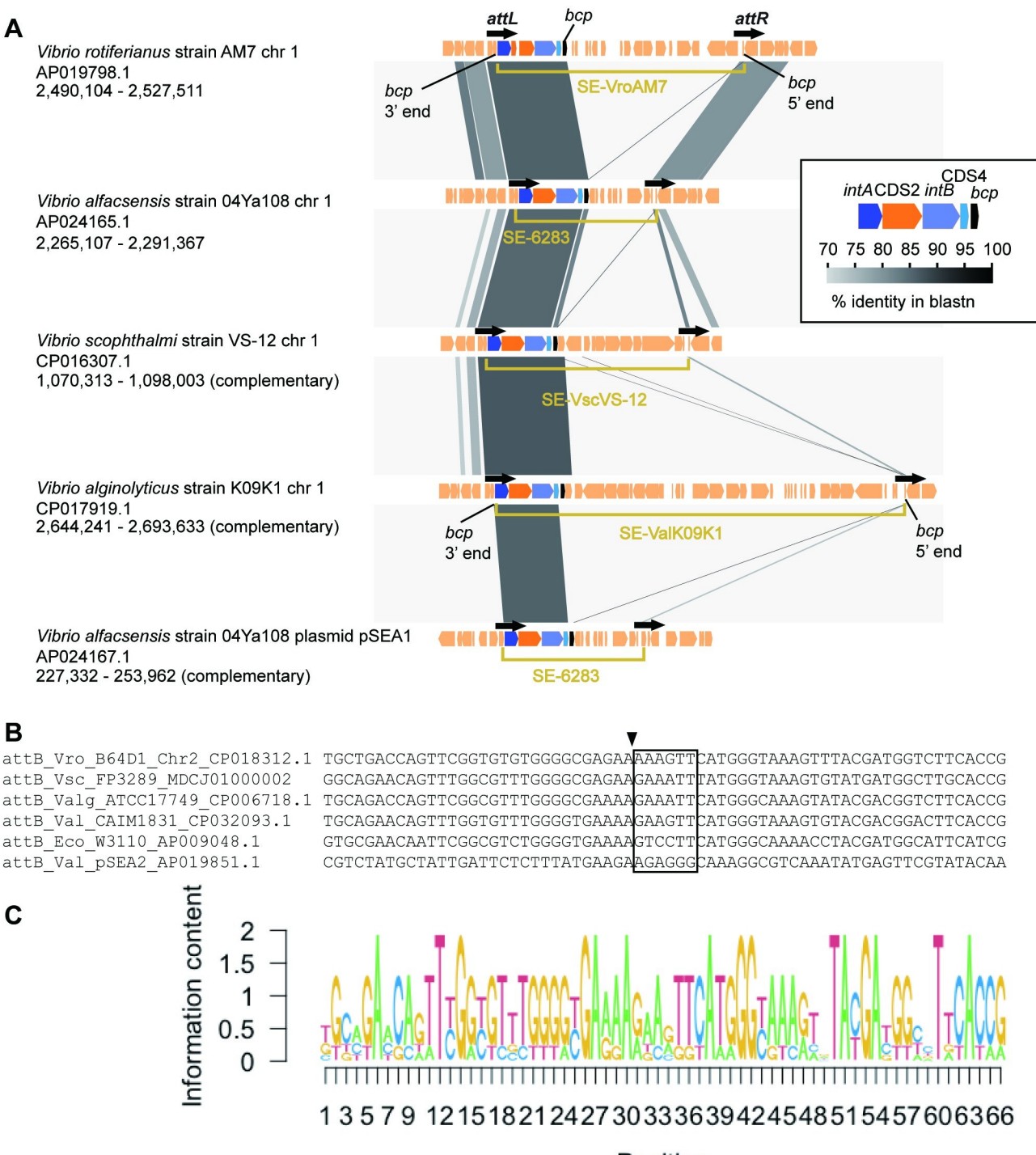

**Fig 3. Location of SE-6283 and its related SEs.** (A) Comparisons of genetic organization around SE elements in five genomic locations. GenBank accession numbers and the coordinates of genomic segments compared are shown below the strain name. Except SE-6283 on pSEA1, all elements are integrated into a *bcp* gene and carry a *bcp* gene. (B) Alignment of *attB* sequences from different bacterial species. (C) Sequence logo showing conservation levels of *attB*. The arrowhead indicates the putative nicking site on the top strand.

then Tc Ap-resistant transconjugants should emerge at a comparable frequency to Ap-resistant transconjugant, which was not the case in our observation.

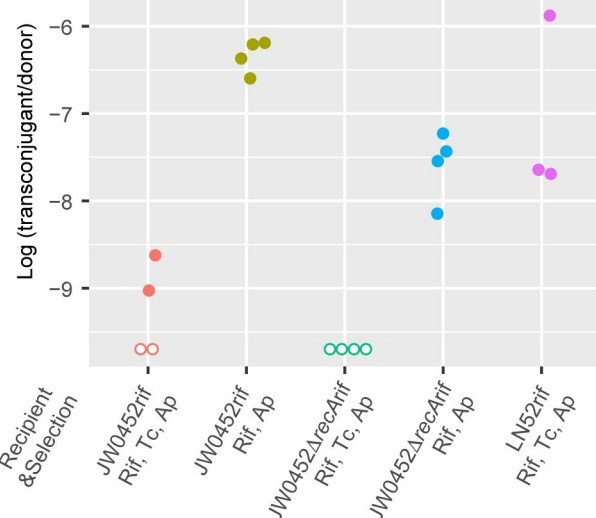

**Fig 4. Resistance gene transfer from *V. alfacsensis*.** Transfer frequency (Y-axis) is the $\log_{10}$-transformed value of the transconjugant colony-forming units (CFU) divided by the donor CFU. Four replicate mating experiments were performed. No transconjugants were detected in any of the four replicate experiments using JW0452Δ*recA*rif with Tc Ap selection or in the two replicate experiments using JW0452rif with Tc Ap selection (indicated by open circles placed at the bottom). In one experiment on LN52rif, the donor CFU was not evaluated, but the transconjugant CFU was obtained at a comparable frequency to that observed in the other three replicate experiments. The detection limit of this experiment was around $1.02 \times 10^{-9}$ per donor.

Collectively, these results and the nature of tyrosine recombinases support the model that pSEA2-chromosome cointegration mainly occurs through a two-step gene transfer mechanism, transposition followed by homologous recombination, rather than through a one-step Tn*3*-like cointegrate formation process.

**Strand-biased circularization of SE-6945.** We previously [30] revealed the following unique features of SE-6283: (i) it does not generate an unoccupied donor site upon circularization *in vivo*, and (ii) the circular form of SE-6283 is generated using only one strand as a template at least in *E. coli*. We note that this assertion is based on the notion that tyrosine recombinases generally do not produce blunt ends at *att* sites on double stranded DNA during strand exchange. Although SE-6283 encodes tyrosine recombinases and not a single DDE transposase, the outcomes described above, which result only from a figure-eight structure containing a single stranded DNA bridge (Path 1 in Fig 5A), is reminiscent of the first step of the 'copy-out paste-in' transposition of some ISs [41] (IS*3* route), rather than the excision of ICEs (ICE/IME route) or the generation of circular integron gene cassettes by *attC* × *attC* recombination (gene cassette route) [25, 42, 43]. If the figure-eight structure containing a single stranded DNA bridge is subject to replication without DPC repair, the products would be a single stranded SE circle, the original copy of donor molecule, and the nicked chromosome, which is eventually lost (Fig 5A bottom left). To test whether circular copy production of SE-6945 also follows Path 1, we created a pSEA2-free 04Ya249 derivative strain LN95 carrying only a single copy of SE-6945 in the chromosome (S5B Fig) and analyzed the production of its circular molecule in this strain and the sequence of strand exchange products.

Pairs of primers were designed such that one primer anneals inside the integrative element and the other outside it (Fig 5A), amplifying *attL* (*intA* side, Fig 5B, product size 304 bp), *attR* (*bla* side, 248 bp), empty donor site (*attB*, 345 bp), occupied *attB* (7.4 kb), and *attS* (joint of motif C and C′ formed on the circular SE-6945, 206 bp). Primer set '1–4' in Fig 5A was used to detect both empty *attB* and occupied *attB*.

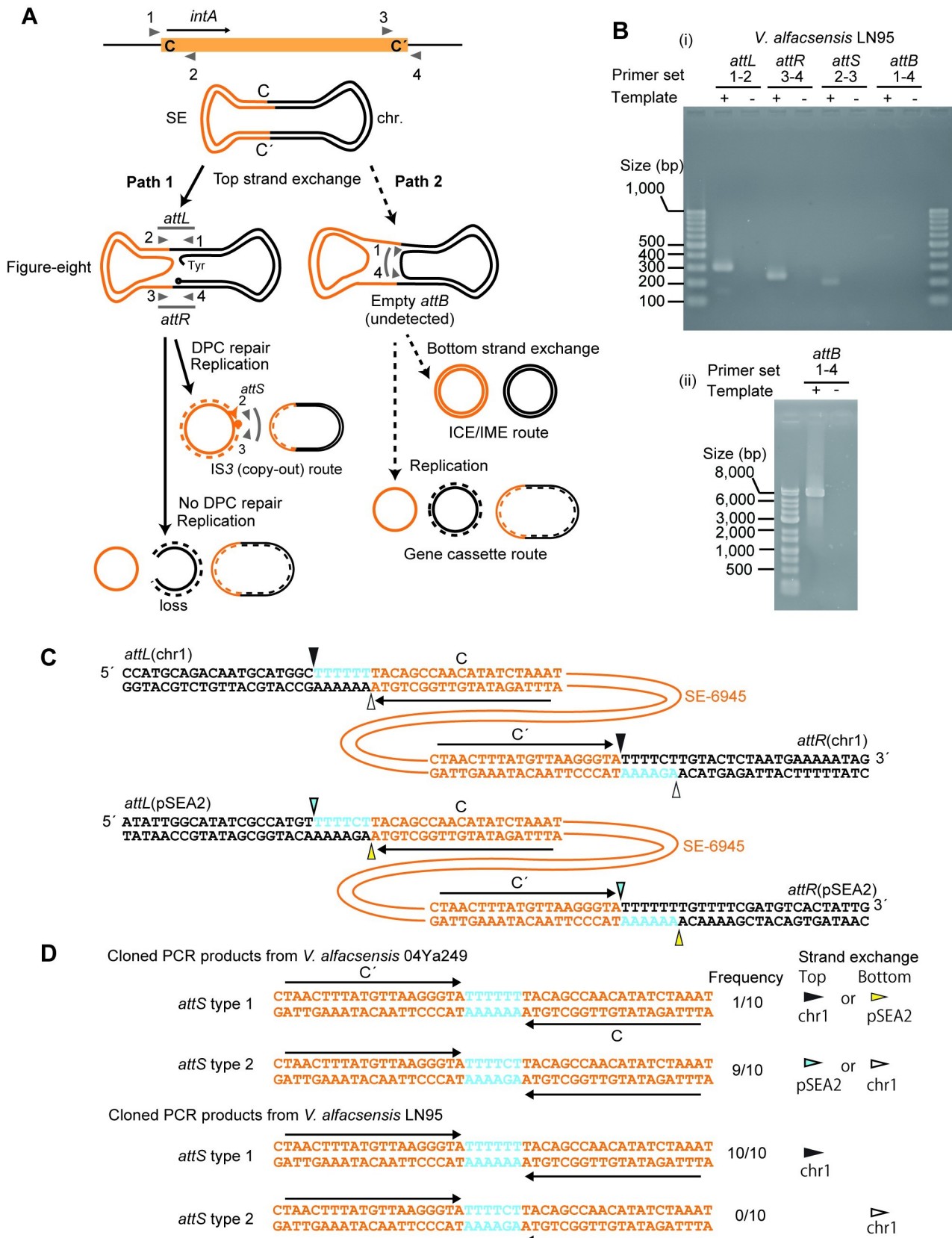

**Fig 5. PCR and sequencing analyses of SE-6945 circle.** (A) PCR experiment to detect SE-6945 circle. Upper panel: primer annealing locations (triangles in gray) on SE-6945 and the *attB* region. The number corresponds to the primer set numbers in B. Primer number indicates the following primers in Table 2: 1, LN_181_attB_L; 2, LN_142_junction2; 3, LN_143_junction2; 4, LN183_attB_R2. Lower panel: possible routes of SE movement. Two strands of DNA are shown as separate lines, with SE in orange. In Path 1 (solid arrows), one strand is nicked, and the SE side is circularized while the host DNA side of the strand remains not joined, leading to a figure-eight structure containing a single stranded DNA bridge. The figure-eight form can be subject to replication, giving rise to a double stranded SE circle (if the recombinase bound 3′ end is cleaned, for example, by DPC repair [40], referred to as 'IS*3* (copy-out) route'), or single stranded DNA circle (if the 3′ end is not cleaned). In Path 2, strand exchange generates a figure-eight containing a double stranded DNA bridge. After going through Path 2, there are two possible outcomes: (i) another round of strand exchange at bottom strand leading to a circular double stranded SE as well as a circular donor molecule carrying empty donor site ('ICE/IME route'), (ii) replication on the figure-eight leading to single stranded SE circle and donor molecule with empty *attB*, and the original donor molecule ('gene cassette route'). (B) The result of PCR. PCR products were electrophoresed in 2% agarose gel (i) or 0.8% agarose gel (ii). (C) Putative nicking sites in *attL* and *attR*. The SE DNA is shown in orange. Nicking and ligation at positions shown as arrowheads in the same color can generate *attS* with a 6 bp spacer. (D) Observed *attS* sequences. *attS* type 1 can be generated from strand exchange at the top strand on chromosome 1 or bottom strand on pSEA2, while *attS* type 1 can be generated from strand exchange at the top strand on pSEA2 or bottom strand on chromosome 1.

The circular form of SE-6945 was detected in LN95 (Fig 5B (i)) as well as the parent strain 04Ya249 (S8A Fig). This suggests that the chromosomal copy of SE-6945 in *V. alfacsensis* is functional. The empty *attB* (345 bp) could not be detected under PCR conditions with short extension times (Fig 5B (i)), while a larger product of occupied *attB* (7.4 kb) was detected when long PCR conditions were used (Fig 5B (ii)).

To estimate the copy number of the circular form of SE-6945 in the cell population, we first searched for Illumina reads of strain LN95 that spanned *attS* or the "hypothetical" *attB*. Although next-generation sequencing reads were obtained at 229× chromosome coverage, no reads spanning *attS* or *attB* were detected [44]. Quantitative PCR further revealed that the mean *attS* to *gyrB* ratio was 0.0012 (S8B Fig). This relative copy number was consistent with the result from a previous study detecting the circular form of SE-6283 in *E. coli*, which showed an *attS* to chromosome ratio of 0.001 [30].

To determine the nicking site on *attL* and *attR* (Fig 5C), we cloned the PCR products of *attS* and then sequenced ten cloned molecules. Based on the *attL*, *attR*, and the original *attB* sequences in the TJ249 chromosome (S3 and S4 Figs), we hypothesized four nicking sites (arrowheads in Fig 5C) in *attL* and *attR* on pSEA2 and chromosome 1 in strain 04Ya249. The cloned *attS* PCR products of 04Ya249 contained a 6 bp spacer sequence between C and C' (Fig 5D). This observation supports the hypothesis that nicking occurs 6 bp upstream at the C end and the C' end and/or 6 bp upstream at the C' end and the C end. Two sequences, 5′-TTTTTT-3′ (*attS* type 1) and 5′-TTTTCT-3′ (*attS* type 2) were detected at a 1:9 ratio in the spacer region between C and C' in strain 04Ya249 (examples of the Sanger sequencing trace files have been posted in figshare [44]). These two types of spacer sequences can arise from both pSEA2 and chromosome 1 (Fig 5D). However, only one sequence (*attS* type 1) was detected in the pSEA2-free strain LN95 (Fig 5D), indicating that strand exchange occurs only on the top strand in *attL* × *attR* recombination on chromosome 1 (Fig 5C). The occurrence of *attS* type 1 and *attS* type 2 in 04Ya249 (Fig 5D) agrees with the top strand exchange in *attL* × *attR* recombinations on chromosome 1 and pSEA2, respectively (Fig 5C). Collectively, these observations are consistent with the model that SE-6945 follows Path 1 (Fig 5A) without undergoing strand exchange of the bottom strand in *V. alfacsensis*.

Nicking of one strand of the double stranded DNA and its circularization is often seen in the reactions mediated by HUH endonucleases (relaxase/rolling-circle replication initiator) [45]. To confirm that observed nicking at *attL* and *attR* is independent of HUH endonucleases produced from plasmids or other mobile elements in *V. alfaccensis*, the production of *attS* was additionally investigated using *E. coli* strain LN52 that carries a single copy of SE-6945 but lacks pSEA2. *attS* was produced even in LN52 (S8C Fig). Therefore, strand-biased (top strand only) *attL* × *attR* recombination is likely mediated by DNA strand exchange enzymes encoded by SE-6945.

### The insertion copy number and location of SE-6945 affects Ap resistance levels of the host cell

The naturally occurring strains 04Ya249 and 04Ya108 carry two copies of SE-6945 in their genomes. However, it remains unclear whether increased SE copies in a genome confer an advantage to the host cell. Three copies of SE-6945 in the transconjugant TJ249 were initially detected by genome sequencing. Southern hybridization analysis of other JW0452 transconjugants revealed one to two copies of SE-6945 in their genomes (S5 Fig). The location of SE-6945 became evident based on Southern hybridization for 15 among 19 transconjugants. Therefore, we used these transconjugant strains to test whether multiple integrations of SE-6945 or its integration location can increase the Ap resistance phenotype of the transconjugants.

Sixteen *E. coli* strains with or without SE-6945 were grown in Luria-Bertani broth with Ap, and then the minimum inhibitory concentrations (MICs) against individual strains were determined using the broth dilution method following the Clinical Laboratory Standards Institute (CLSI) guidelines. The MIC was highest in the clone carrying three copies of SE-6945, followed by the group of transconjugants carrying two copies and the group carrying one copy, and the MIC was the lowest in JW0452 (Fig 6). Although within-group variation was also observed, the increased copy number of SE-6945 in the genome generally resulted in the increased Ap resistance of the host cell.

The MICs of strains are shown in S1 Table. There was a trend that strains with SE-6945 integrated into *insJ* showed a higher MIC than strains with integration into *yjjNt*. Thus, the location of SEs can also affect phenotype variation, possibly due to the differences in the expression levels of their cargo genes.

## Discussion

*Vibrio* is one of the major bacterial genera found in marine sediments [46], and is among the most common microbiota of wild and farmed shrimp [47] and fish [48, 49]. Several members of the genus *Vibrio* are pathogens of fishes reared in aquaculture [50], while other subsets of

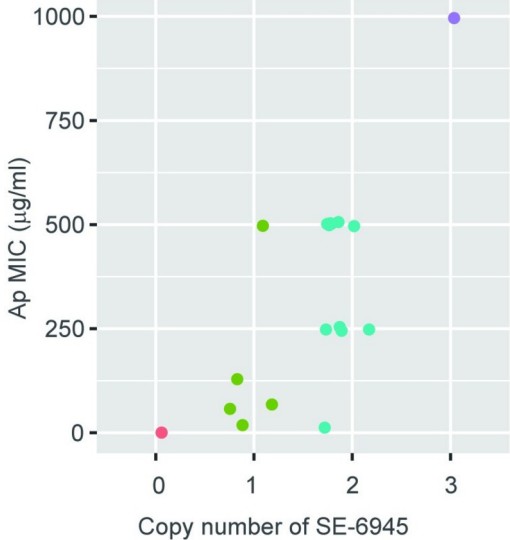

**Fig 6. The copy number of SE-6945 affects the Ap resistance level.** Sixteen transconjugants and a control strain (JW0452) were divided into four groups shown in distinct colors based on the SE-6945 copy number in the genome. MIC of ampicillin is shown according to copy number.

*Vibrio* species, including *V. cholerae*, *V. parahaemolyticus*, and *V. vulnificus*, which are ubiquitous in relatively low-salinity sea water, are human pathogens from seafood and/or water. [50, 51]. Thus, *Vibrio* can be considered as a key genus aiding movement of AMR genes between aquatic environments and human-associated environments. Indeed, the accumulation of AMR genes in this genus has attracted increased research attention, particularly in *V. cholerae* [52, 53]. However, direct experimental evidence for AMR gene transfer from *Vibrio* species to other human-relevant bacteria is limited.

Known genetic mechanisms of AMR gene transfer from *Vibrio* include A/C plasmids [54, 55], unclassified conjugative plasmids [56], pAQU1-type MOB$_H$-family conjugative plasmids [57], pSEA1-type MOB$_H$-family conjugative plasmids [30], ICEs [58], IMEs [19], and a combination of a sedentary chromosomal integron and a conjugative plasmid carrying an integron [59]. The host range of pSEA1-type MOB$_H$ plasmids, discovered in *V. alfacsensis* (for which only one complete genome was available until the present study), is unknown. In this study, we investigated how a pSEA1-like plasmid can contribute to AMR gene transfer in a laboratory setting.

In contrast to A/C plasmids, autonomous replication of pSEA2 and pSEA1 was difficult to achieve in *E. coli*, since transconjugant selection generally yielded *E. coli* clones carrying plasmid DNA integrated into the chromosome. The current most likely mechanism behind this phenomenon is transposition of SE followed by homologous recombination, which enabled AMR gene transfer beyond the plasmid's replication host range. This mode of horizontal gene transfer may be important among marine bacteria, since the plasmid conjugation host range is expected to be wider than the replication host range [60, 61].

pSEA1 was found to carry two SEs (SE-6283 and SE-6945). However, in the previous study [30], only transposition of SE-6283 was observed, likely because we did not recognize SE-6945 on pSEA1 and did not intend to detect its transposition. The newly identified SE-6945 is the smallest SE identified to date (Figs 2 and 3), and the only known SE harboring an AMR gene. As SEs can have more than one target site in a single genome, they may contribute to microbial adaptation to the antimicrobials used at aquaculture sites, with diverse mechanisms beyond mediating the horizontal transmission of AMR genes, such as increasing the resistance level or gene redundancy preceding evolutionary innovation of an AMR gene [62].

The discovery of SE-6945 highlights four potential core genes (*intA*, CDS2, *intB*, and CDS4) present in SEs. However, the specific roles of these gene products in their transposition steps remain to be determined. Among the types of transposon movement classified by Curcio and Derbyshire [15], 'copy-out' is currently the most consistent with our observations of SEs regarding the first step of its location change. One model for strand exchange on single stranded DNA mediated by tyrosine recombinase, leading to a single stranded circle, has been proposed for integron gene cassette excision (replicative resolution model in [43]). In the replicative resolution model, strand exchange between two folded *attC* bottom strands and the following replication generates a gene cassette circle with *attC* and results in the removal of one gene cassette with *attC* from the gene cassette array on the donor molecule. This can cause formation of integron variants with fewer gene cassettes and even integrons with an empty *attI*, as observed in *in vivo* experiments [42]. SE circle formation rarely leaves the empty *attB* (Fig 5B (i) and [30]). Thus the rejoining of the host DNA side at nicking sites is somehow prevented, possibly by the action of multiple proteins involved: possibly the CDS2, CDS4 products, unlike the movement of gene cassettes. Currently, we do not have information on the fate of the nicked donor molecule. Unlike DDE transposases, tyrosine recombinases usually do not generate a free 3′ OH end and are covalently linked to the phosphate of substrate DNA (Fig 5A) [63]. Therefore, a process equivalent to DPC repair [40], which might require both host and SE-encoding factors, is necessary to achieve the IS*3*-like copy-out process. Nonetheless,

we speculate that the alternate single stranded DNA circle generation route, accompanying chromosome loss (bottom left in Fig 5A), is a less likely route as it does not give any advantage to target site-selective and putatively non-transferrable SEs left in dying cells.

While strand-biased *attL* × *attR* recombination was repeatedly observed for the two SEs, not much information has been obtained regarding their integration. While 6 bp from the 5′ end of C at *attL* on the donor molecule is incorporated into a circular SE copy, the incorporated 6 bp tends to be placed next to the 3′ end of C′ at *attR* after integration into the target site (see *attL* on pSEA2 and *attRs* on TJ249, and previously detected SE-6283 in transposition from pSEA1 to *bcp* in *E. coli*). Thus, there seems to be a strict rule in the order of strand exchange at the integration step, but determining the precise order of nicking and nicking sites on the two strands during the integration step requires further investigations.

A notable difference between IS and SE is the strong strand bias unique to SE upon strand exchange: some ISs can generate a single stranded bridge using either of the two strands (but not both simultaneously) [64, 65], while SEs use only top strand. The dissimilarity between SE and ICE/IMEs is that ICE/IMEs encode the recombination directionality factor Xis [25], while SEs do not encode Xis homologs. CDS2 or CDS4 products may function exclusively in either the initial circularization step or the integration as the recombination directionality factor. Another dissimilarity between SE and ICE/IME is that the former does not encode an HUH endonuclease. There is still a possibility that SEs carry an *oriT* recognized by a family of relaxases and are mobilized like CIME [21], though we could not identify an *oriT* in SE-6945 or SE-6283 by local similarity search alone. It may also be possible that IntB, a large tyrosine recombinase, functions as a relaxase, since a conjugative plasmid from *Clostridium* has been shown to use a tyrosine recombinase homolog as a conjugative relaxase [66].

We propose that interplay between SEs and conjugative MDR plasmids contribute to AMR gene transmission in marine bacteria. Further biochemical and bioinformatic studies on SEs are needed to reveal the mechanisms of gene transfer among the genus *Vibrio* and other aquaculture-associated bacteria.

To refer to transposable individual SE members, we used numbers that are compatible with the existing transposon database [33]. However, further debate is needed regarding handling SEs in mobile DNA databases.

## Materials and methods

### Strains, plasmids, and culture media

Strains and plasmids used in the study are listed in Table 1. Strain 04Ya108 was previously identified as *Vibrio ponticus* based on 16S rRNA gene sequence similarity. However, determination of the complete sequence in this study revealed that it shows >96% average nucleotide identity to *V. alfacsensis* strain CAIM 1831 (DSM 24595) (S1B Fig) [67]. Therefore, strain 04Ya108 was re-classified as *V. alfacsensis*. Strain LN95 is a pSEA2-free tetracycline-susceptible derivative of strain 04Ya249. This strain was generated through a repeated batch culture of 04Ya249, and subsequent single-colony isolation. The absence of pSEA2 from LN95 was confirmed by next-generation sequencing.

The presence of a single SE-6945 copy in the LN52 genome was confirmed by Southern hybridization (S5 Fig). The *recA*-null mutant of JW0452, JW0452Δ*recA*, was constructed using the lambda-Red method [69]. A DNA fragment containing the 5′ and 3′ sequences of the *recA* gene and a chloramphenicol resistance gene was amplified by PCR using primers YO-175 and RecA_stop_primingsite_2 (Table 2) and plasmid pKD3 [69] as a template. The PCR products (700 ng) were introduced into electrocompetent cells of JW0452 carrying pKD46 by electroporation using a Gene Pulser Xcell™ (BioRad, Hercules, CA, USA). The occurrence of

recombination at the expected site was confirmed by PCR using primers CAT-584 and BW25113_2815723f (Table 2). The absence of the *recA* gene was also confirmed by PCR using primers LN192_recA1 and LN193_recA2.

*Vibrio alfacsensis* strains were cultured in BD Bacto™ brain heart infusion medium (BD237500; Becton, Dickinson, and Company, Franklin Lakes, NJ, USA) supplemented with up to 2% NaCl at 25°C. *E. coli* strains were cultured in BD Difco™ LB Broth, Miller (BD244520; Becton, Dickinson, and Company) at 37°C or 42°C. BD Difco™ Mueller Hinton Broth (BD 275730; Becton, Dickinson, and Company) was used for antibiotic susceptibility testing of *E. coli*. BD Difco™ Marine Broth 2216 (BD279110; Becton, Dickinson, and Company) was used for filter mating. Solid media were prepared by adding 1.5% agar to the broth. Antibiotics were added to the medium at the following concentrations when required: erythromycin (Nacalai Tesque, Kyoto, Japan), 100 μg/ml; tetracycline (Nacalai Tesque), 10 μg/ml; rifampicin (Sigma-Aldrich, St. Louis, MO, USA), 50 μg/ml; ampicillin (Nacalai Tesque), 100 μg/ml.

## Conjugation

The donor *Vibrio* strain and *E. coli* recipient strains were grown overnight at 25°C and 37°C, respectively. A 500 μl aliquot of each culture was mixed, centrifuged, and resuspended in 50 μl of Luria-Bertani broth. The cell mixture was spotted on a 0.45 μm pore-size nitrocellulose filter (Merck, Millipore Ltd., Tullagreen, Ireland) placed on marine broth agar, and allowed to mate for 24 h at 25°C. After incubation, the cell mixture on the filter was serially diluted in 1× phosphate-buffered saline, and then 100 μl of the mixture was plated on an appropriate agar medium to measure the CFU and incubated at 42°C to suppress growth of *Vibrio* strains. After genome sequence determination of TJ249, more than 17 independent mating experiments were performed to obtain insights into the variation of pSEA2 integration patterns using

**Table 2. Oligonucleotides used.**

| Name | Sequence (5′ to 3′) | Purpose |
|---|---|---|
| LN112 | GGGTTACCTTCCCAATGCGT | Southern hybridization probe for SE-6945 *intA* |
| LN113 | CGACTGTTGGTAGCGACTGT | Southern hybridization probe for SE-6945 *intA* |
| LN_142_junction2 | AAGATGGTAAAAGTGTTCCA | Detection of *attS* (206 bp) by qPCR and *attL* (304 bp) by standard PCR |
| LN_143_junction2 | TTTGTGTGTAGCCCTTGTG | Detection of *attS* (206 bp) by qPCR and *attR* (248 bp) by standard PCR |
| LN_150_intA2 | GGTTATGTGGAGAAGTTGCC | Detection of *intA* (103 bp) by qPCR |
| LN_151_intA2 | TGAGTTCGGTTTCTTGCTTC | Detection of *intA* (103 bp) by qPCR |
| LN_181_attB_L | CGAGGGTAAAGTGCCAACAT | Detection of chromosomal *attB* (345 bp if empty, 7.4 kb if occupied) and *attL* (304 bp) by standard PCR (04Ya249) |
| LN183_attB_R2 | ACATCAGCAGGAGTTAGTTG | Detection of chromosomal *attB* (345 bp, 7.4 kb) and *attR* (248 bp) by standard PCR (04Ya249) |
| LN184_gyrBf1 | AACAGAATTGCACCCAGAAG | Detection of *gyrB* (110 bp) by qPCR (04Ya249) |
| LN185_gyrBr1 | GAAGACCGCCTGATACTTTG | Detection of *gyrB* (110 bp) by qPCR (04Ya249) |
| YO-175 | CAGAACATATTGACTATCCGGTATTACCCGGCATGACAGGAGTAA AAATGTGTAGGCTGGAGCTGCTTCG | Amplification of *cat* gene for Lambda-Red recombination |
| RecA_stop_primingsite_2 | ATGCGACCCTTGTGTATCAAACAAGACGATTAAAAATCTTCGTTAGTTTCCA TATGAATATCCTCCTTA | Amplification of *cat* gene for Lambda-Red recombination |
| CAT-584 | AAGCCATCACAAACGGCATG | Confirmation of *cat* gene insertion |
| BW25113_2815723f | AATACGCGCAGGTCCATAAC | Confirmation of *cat* gene insertion |
| LN192_recA1 | GTTCCATGGATGTGGAAACC | Confirmation of *recA* deletion |
| LN193_recA2 | ATATCGACGCCCAGTTTACG | Confirmation of *recA* deletion |

JW0452 as the recipient, and transconjugants were screened at 42˚C in the presence of Tc alone, Ap alone, or both Ap and Tc. The AMR phenotype of each transconjugant was further determined after single-colony isolation using Ery, Cm, Ap, and Tc (Table 1).

Quantitative mating assays (Fig 4) were performed to address the involvement of homologous recombination in pSEA2 integration using strains JW0452rif, JW0452Δ*recA*rif, and LN95rif as recipients. Selection conditions for transconjugants were Rif Ap Tc 42˚C or Rif Ap 42˚C. The *Vibrio* donor strain was selected on brain heart infusion agar with 2% NaCl supplemented with Tc at 25˚C. CFUs of donor, recipient, and transconjugant were determined after 24 hours of incubation.

## PCR experiments

The PCR detection of *attL*, *attR*, and *attS* was performed using TaKaRa Ex Taq (TaKaRa Bio Inc., Kusatsu, Japan), and genomic DNA of 04Ya249, LN95, and LN52 prepared using QIAGEN DNeasy Blood & Tissue kit (QIAGEN GmbH, Hilden, Germany). PCR premix was prepared in a 50 μl scale, and 1 μl of the template (10 ng/μl) was added to the reaction mix throughout the study. The PCR cycle used was 94˚C for 1 min, followed by 35 cycles of 94˚C for 30 s and 60˚C for 30 s. As the empty *attB* was not amplified using TaKaRa Ex Taq, we conducted additional long PCR using LA Taq polymerase (TaKaRa Bio Inc.) to detect occupied *attB* and confirm the functionality of primers used for detection of empty *attB*. The PCR cycles used for long PCR were 1 cycle of 94˚C for 1 min, followed by 35 cycles of 95˚C for 1 min, 65˚C for 1 min, and 72˚C for 1 min. PCR products of *attS* were further cloned using the pGEM-T easy vector system (Promega, Madison, WI, USA) and DH5α competent cells to analyze the sequence variation of PCR products.

Quantitative PCR was performed using THUNDERBIRD® SYBR qPCR Mix (Toyobo, Osaka, Japan) and a CFX connect Real-Time system (BioRad, Hercules, CA, USA) and a two-step PCR protocol of 40 cycles of 95˚C for 5 s and 60˚C for 30 s. Target quantity was estimated based on a standard curve of the control plasmid DNA (pGEM-*gyrB*, pGEM-*intA*, pGEM-*attS*) constructed by TA cloning. Primers used and their purposes are listed in Table 2 and Fig 5 legend.

## Southern hybridization

The copy number and integration location of SE-6945 in transconjugants (JW0452 derivatives) were analyzed by Southern hybridization using the 5′ end of *intA*, amplified using primers LN112 and LN113 (Table 2) and the PCR DIG Synthesis Kit (Roche, Basel, Switzerland) as a probe. Genomic DNA (2.5 μg) was double digested with either *Nde*I and *Sph*I or *Nde*I and *Hin*dIII (New England Biolabs, Ipswich, MA, USA), then electrophoresed in 0.8% agarose gel. Separated DNA fragments were transferred to the nitrocellulose membranes (Amersham Hybond-N+; GE Healthcare UK Ltd., Buckinghamshire, UK) by capillary transfer. The probe was detected using CDT-star® (Roche). Restriction map for the DNA segments around SE-6945 integration locations and the expected size of DNA segments with probe signal are shown in S5A Fig.

## Antimicrobial susceptibility testing

To examine the antimicrobial susceptibility of transconjugants harboring one to three SE-6945 copies, the MIC of ampicillin was determined using the broth dilution method in 96-well microtiter plate format according to standard M07 of the CLSI [70]. The Ap concentrations tested were 1000, 500, 250, 125, 62.5, 31.25, 16, 8, 4, and 2 μg/ml. The test plates were incubated at 35˚C for 24 h.

## Genome sequencing

Genomic DNA was extracted from 250–500 μl of *V. alfacsensis* strains 04Ya249, 04Ya108, or *E. coli* strain TJ249 culture using the QIAGEN DNeasy Blood & Tissue kit (QIAGEN GmbH). The extracted genomic DNA was sequenced on the PacBio RS II platform using P6C4 chemistry at Macrogen (Tokyo, Japan) in 2016. Genome assembly was conducted using HGAP v.3 [71] for *Vibrio* strains and Flye v 2.8.3-b1695 for *E. coli* TJ249 [72]. Reads were obtained at >120× coverage for the chromosomes of each strain. Reads and the genome sequence of TJ249 have been posted to figshare [73]. Illumina reads of the pSEA2-free strain LN95 were obtained using the TruSeq PCR-free library and NovaSeq 6000 platform at NovogeneAIT Genomics Singapore Pte., Ltd. (Singapore) to confirm the loss of pSEA2 and to identify the circular form of SE-6945. The mapping results have been posted to figshare [44]. Genomes were compared using MUMmer3.23 [74] and GenomeMatcher [75]. AMR genes in the assembly were searched for using AMRFinderPlus [76]. The average nucleotide identity was determined using fastANI [77].

**Searches for SE-6945 and SE-6283-related SEs.** Nucleotide sequences of the SE core gene region (*intA*, CDS2, *intB*, CDS4) (6432 bp for SE-6283, 5541 bp for SE-6945) were used as queries in blastn searches against the NCBI nucleotide collection (nr/nt) database (August 3, 2021). Hit subjects showing > 85% nucleotide identity with > 99% query coverage were regarded as close relatives of SE-6945 or SE-6283, and were further investigated as to their integration locations.

## Supporting information

**S1 Fig. Comparison of the genome structure between strain 04Ya249 with 04Ya108, CAIM 1831 and TJ249.** (A) Structure comparison was performed using nucmer in MUMmer3 [74]. Purple dots indicate a match on the Watson strand (5′ to 3′ on the top strand in GenBank file), and light blue indicates a match on the Crick strand (5′ to 3′ on the bottom strand). (B) Average nucleotide identity (ANI) between two strains as determined by fastANI [77]. The commands used were as follows: (A) $nucmer -minmatch 60../../data/04Ya249_submission.fas../../data/reference.fas$mummerplot -x "[0,6000000]" -y "[0,22000000]" -postscript -p test out. delta; (B) $fastANI -q../../data/04Ya249_submission.fas -r../../data/CAIM1831_Refseq.fas -o 04Ya249vsCAIM1831.txt $fastANI -q../../data/04Ya249_submission.fas -r../../data/04Ya108_submission.fas -o 04Ya249vs04Ya108.txt$fastANI -q../../data/04Ya249_submission.fas -r../../data/04Ya108_submission.fas -o 04Ya249vs04Ya108.txt.
(TIF)

**S2 Fig. Genetic map of pSEA2.** (A) Locations of antimicrobial resistance (AMR) genes and SE-6945 in pSEA2. AMR genes were inferred using AMRFinderPlus [76]. Genes were visualized using CLC Sequence Viewer (Qiagen, Hilden Germany). (B) Location of SEs in pSEA1 and pSEA2. Four SE core genes are indicated by four distinct colors. Red pentagons are the β-lactamase gene.
(TIF)

**S3 Fig. *attL* sequences formed in the SE-6945 integration regions.** Sequences are derived from 04Ya249 genomes (AP019851.1, AP019849.1), TJ239 genome (*insJ*, *yjjNt*), and *Vibrio harveri* strain WXL538 (CP045070.1) carrying a SE-6945-like element carrying a different set of cargo genes. Sequences in red are incorporated into circular copy of SE. 6-bp underlined sequence is incorporated into *attS* during SE circle formation. 19 bp motif C highlighted in yellow form an imperfect inverted repeat with the motif C′ located the other terminus of SE. Sequence highlighted in cyan is a putative transcription termination motif. Asterisks indicate conserved nucleotides.
(DOCX)

**S4 Fig.** *attR* **sequences formed in the SE-6945 integration regions.** Sequences are derived from 04Ya249 genomes (AP019851.1, AP019849.1), TJ239 genome (*insJ*, *yjjNt*), and *Vibrio harveri* strain WXL538 (CP045070.1). Sequences in red are incorporated into circular copy of SE. The 6 bp underlined sequences originate from the original *attB* or SE circle, and they are not incorporated into the *attS* of the SE circle.
(DOCX)

**S5 Fig. Inference of SE-6945 insertion number and its integration location by Southern hybridization.** (A) *in silico* restriction map of DNA segments around SE-6945 integration locations in strain 04Ya249, TJ249, and a hypothetical *E. coli* strain carrying a pSEA2 insertion into SE-6945 integrated into *yjjNt*. Locations of SE-6945 and *intA* probe hybridization region are indicated by the yellow square and black filled square, respectively, above the restriction map. Horizontal black lines indicate the fragments detected by the probe and their sizes. (B) Southern blots of the pSEA2-free Vibrio strain LN95 and the parental strain 04Ya249. Genomic DNA was double digested with *NdeI* and *HindIII* (the left two lanes) or *NdeI* and *SphI* (the right two lanes). The probe used was 5' end of *intA*. Four unique bands originate from fragments shown in the first two rows (04Ya249 pSEA2, 04Ya249 chr1) in panel A. (C) Southern blots of 19 *E. coli* transconjugants obtained from 19 independent mating assays. Upper panel shows digestion with *NdeI* and *SphI*. Lower panel shows digestion with *NdeI* and *HindIII*. The color of the strain name indicates the pattern of SE/pSEA2 insertion deduced from restriction maps in panel A: blue, three SE copies (*yjjNt*::SE-6945, *insJ*::SE-6945::pSEA2); red, two SE copies (*insJ*::SE-6945::pSEA2); purple, (*yjjNt*::SE-6945::pSEA2); lime, one SE copy (*yjjNt*::SE-6945 or chr::pSEA2); Strains with asterisk (*) are expected to carry pSEA2 in an unknown chromosomal location without generating a SE-6945-chromosome junction.
(TIF)

**S6 Fig.** *attL* **sequences formed in the SE-6283 insertion regions.** Sequences are derived from strain 04Ya108 (AP024165.1, AP024167.1), *E. coli* transconjugant TJ108W0 sequenced in our previous study [30], and *Vibrio* strains carrying SE-6283-like elements listed in Fig 3. Sequences in red are incorporated into circular copy of SE. Incorporation of the 6 bp underlined sequence of TJ108W0 and pSEA1 into *attS* was demonstrated in the previous study.
(DOCX)

**S7 Fig.** *attR* **sequences formed in the SE-6283 insertion regions.** Sequences are derived from genomes listed in Fig 3. Regarding SE-6283, the 6 bp underlined sequences in *attR* originate from inserting SE circle, and they are not incorporated into *attS* upon SE circularization.
(DOCX)

**S8 Fig. PCR detection of SE-6945 circle in** *V. alfacsensis* **strain 04Ya249 and** *E. coli* **strain LN52.** (A) PCR detection of chromosomal *attL*, *attR*, *attS* in 04Ya249. (B) Relative copy number of *attS* in *V. alfacsensis* LN95 (no pSEA2). Copy number is represented as relative copy number to *gyrB*. Primers used are shown in Table 2. (C) PCR detection of *attS* in pSEA2 free *E. coli* strain LN52.
(TIF)

**S1 Table. MIC of transconjugant strains.**
(DOCX)

**S2 Table. Data availability.**
(DOCX)

**S1 Raw images.**
(PDF)

## Acknowledgments

We thank the National Bioresource Project of the National Institute of Genetics, Japan, for providing the *E. coli* strains BW25113, JW2669, and JW0452. We thank Atsushi Ota and Fumito Maruyama at Hiroshima University for their support for the annotation on the TJ249 assembly. We thank Yuichi Otsuka at Saitama University for helpful discussions and Yuta Sugimoto at Ehime University for experimental support. We thank Kenji K Kojima at Genetic Information Research Institute for the discussion on transposon classification, and Yasukazu Daigaku at Tohoku University for the discussion on DNA protein cross link repair. We thank Editage for language editing (job number LINON_4_4).

## Author Contributions

**Conceptualization:** Lisa Nonaka, Hirokazu Yano.

**Data curation:** Hirokazu Yano.

**Formal analysis:** Hirokazu Yano.

**Funding acquisition:** Lisa Nonaka, Hirokazu Yano.

**Investigation:** Lisa Nonaka, Hirokazu Yano.

**Resources:** Lisa Nonaka, Michiaki Masuda, Hirokazu Yano.

**Supervision:** Lisa Nonaka, Hirokazu Yano.

**Visualization:** Lisa Nonaka, Hirokazu Yano.

**Writing – original draft:** Lisa Nonaka, Hirokazu Yano.

**Writing – review & editing:** Lisa Nonaka, Hirokazu Yano.

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
