## [Decision Letter · Decision Letter 0]

22 Jul 2021

PONE-D-21-18582

Copy-out-paste-in transposition of a Tn6283-like integrative element assists interspecies antimicrobial resistance gene transfer from Vibrio alfacsensis

PLOS ONE

Dear Dr. Nonaka,

Thank you for submitting your manuscript to PLOS ONE. After careful consideration, we feel that it has merit but does not fully meet PLOS ONE’s publication criteria as it currently stands. Therefore, we invite you to submit a revised version of the manuscript that addresses the points raised during the review process.

I have added some comments to help guide in this revision below.

We look forward to receiving your revised manuscript.

Kind regards,

Ruth Hall, PhD

Academic Editor

PLOS ONE

Additional Editor Comments:

Your manuscript incudes some interesting findings but both reviewers had difficulty following the text and had some concerns about particularly the naming you have used. They have given some quite detailed suggestions all of which look very helpful.

I agree with the reviewers on most points particularly that the text is very hard to follow because the information needed (e.g. sizes of PCR products, predicted sizes of restriction fragments, exactly how PCRs were performed) is not provided.

I have a few comments to reinforce or add.

1. Please do not use Tn with a number for an IE. The reviewers have explained why.

2. Please do not use “copy-out paste-in”, which is a specific transposition route and not applicable to integration. It’s known that some tyrosine recombinases perform only one strand exchange. Replication is then needed. This route could give the outputs you observe. The first Int system of this type was IntI1 the integron integrase and this was discovered about it in the 1990s. Please consider this.

3. Please reduce the use of transposon references and information and replace it with Integrase or IE stories.

4. Conjugation frequencies must be expressed as per donor. Please fix this. I will accept scatter plots.

5. A strain Table is needed.

6. I was surprised that 2 genes should be called Int as they has such different sizes. So, I checked them using BLAST and found that IntA has the required domain but IntB has no known domains. To see domains, once the search is complete click on the graphic summary and look right at the top. Hence, I insist that this orf be annotated as an orf.

6. I also noticed that in the database entry for pSEA2, intA is not annotated as such. Please fix this.

6. IE encoding the same or closely related Int should integrate at the same position. Is your IE in the same gene in Vibrio and E. coli? More on this is needed.

6. The searches I did revealed that there are several IE of this type in Vibrio species. Are they in the same location?

Journal Requirements:

Reviewers' comments:

Reviewer's Responses to Questions

**Comments to the Author**

1. Is the manuscript technically sound, and do the data support the conclusions?

Reviewer #1: Partly

Reviewer #2: Partly

2. Has the statistical analysis been performed appropriately and rigorously? 

Reviewer #1: I Don't Know

Reviewer #2: N/A

3. Have the authors made all data underlying the findings in their manuscript fully available?

Reviewer #1: Yes

Reviewer #2: Yes

4. Is the manuscript presented in an intelligible fashion and written in standard English?

Reviewer #1: No

Reviewer #2: Yes

5. Review Comments to the Author

Reviewer #1: This manuscript describes basic characterisation a non-conjugative integrative element (named “Tn6945”), carrying a beta-lactamase (bla) gene, on the chromosome and a plasmid of a Vibrio ponticus isolate from an aquaculture site in Japan. The element was identified following conjugation with E. coli, when either it or the whole plasmid was transferred to the chromosome, the latter process identified as being dependent on recA. “Tn6945” is related to “Tn6238” previously identified by the same group on a related plasmid in a different Vibrio species. The manuscript seems to imply that “Tn6945” is only the second example of this element to be identified (it would be helpful to make this more explicit if this is the case), in which case describing it would be useful. The approach/results appear technically sound, but it is a bit difficult to tell from the descriptions provided, and the conclusions generally justified, but naming these elements transposons is problematic (see point 1 below). Also, the manuscript requires some reorganization rewriting and condensing, as it not easy to follow and is quite wordy and repetitive in places (see examples below).

1) “Non-conjugative integrative element” vs transposon

Description of “Tn6945”/”Tn6238” elements as a transposons is at best confusing, when they carry tyrosine recombinases rather than true transposases. Inconsistencies/contradictions in the text suggest that the authors are not really sure how to describe them either e.g., “nonconjugative integrative element Tn6283” (line 62), “a new type of transposon” (line 66), “Tn6283-like integrative elements” (line 69), “active integrative element” (lines 115-6) and their is reference to them having both IR (typical if transsponsons) and att sites (typical of integrative elements). While some ICE were historically given Tn numbers it is not helpful to perpetuate such confusion and the nomenclature of these “Tn6945”/”Tn6238” elements needs to be carefully considered and if these are a novel type of element then this needs to be clearer.

2) Lines 111-131 – descriptions of insertions

This section needs to be written more clearly/reorganised/condensed.

Line 111 – this is called “MULTISPECIES: class A beta-lactamase” by NCBI RefSeq. It might be useful to obtain a name for it from NCBI, – “a 7.1 kb repeat region”.

Line 114 – it might be better to avoid “homology” here? Gene product identity would be higher than coding sequence identity, so this might be written the other way round.

Lines 115-116 – the evidence for this is from the conjugation experiments? This is not clear.

Lines 120-121 – what is the evidence for targeting, as opposed to random insertion?

Lines 122-4 – “Tn6945” inserts itself, rather than its IR? It’s not clear whether it is being proposed that “Tn6945” has IR, like a typical e.g., Tn3 family transposon (and why are the repeats called C and C’?) which are internal to such Tn, or it has an att site that recombines with target att site to create hybrid sites, as for integrative elements. att site are mentioned in the Fig. 1 legend but are not discussed until later in the text, which is also a problem.

3) Other scientific points

Lines 63, 85, 88, 354, 359 - why was 42 deg C used? This needs to be explained.

Lines 81 - it might be useful to make it clear when citation are to previous papers from the same group of researchers (i.e., use “We”).

Lines 106, 380-1 – is this version of PacBio alone sufficient to give accurate sequences?

Line 109 – is the accession no. for this given anywhere?

Lines 179-80 – att need explaining before this point in the manuscript

Line 223 – what concentration of ampicillin?

Line 240 – V. cholerae is not always from sea food?

Line 250 – the term “super-integron” should not be used. Use e.g., “sedentary chromomsomal integron” instead.

Line 264-66 – is “Tn6945” the smallest of only two of this type of elements to have been identified to date? Or are there other examples in addition to Tn6283? This needs to be clear, i.e., state in the Introduction that is a new type of element, if this is the case.

Lines 266-70 – is it possible to tell anything about how the bla gene might have ended up in this location? Is there really evidence for an “ancient origin”?

Lines 275-6 – these genes need to be described in Results.

Lines 278-80 – what is the evidence of this for the “latter” i.e., Tn6283? This is not really clear from the Results and both references are about IS.

Lines 292-307 – a table explaining the different strains, with some additional details in the text in the case of more complex constructions, might be clearer. The first sentence should be deleted and the citations included in other places.

Line 347-59 – as some strains used are transconjugants, it might be more logical to describe conjugation experiments earlier in Methods? Line 355 is ambiguous and needs to be written more clearly.

Lines 364-8 – digestion should be described before hybridisation and something like “using the 5'end of intIA, amplified using primers LN112 and LN113 (Table 1) and the PCR DIG… as probe” would be simpler.

4) Minor/formatting/wording/English etc

Some streamlining/reducing repetition in different sections etc. is needed, as well as some reorganization. Some examples are given below.

Line 61 – “The” is not needed here.

Lines 63-4 – suggest “…, following conjugation it could integrate into the chromosome by…”

Line 71 – “in this study we mated”

Lines 84-91 are unclear and need rewording.

Lines 110 etc – change “beta” to beta symbol

Lines 118, 120 etc – suggest present tense for facts that are still true i.e., “contains”. “indicates” etc.

Line 125 – “the previously reported plasmid” can be deleted.

Line 127 – can delete “that of”.

Line 128 – could be condensed to “…from Southern hybridization (19)”

Lines 129-30 – suggest “carries Tn6945 in the same place as in…”

Line 134 – “pSEA2” would be clearer than “the plasmid” here. Also line 143 – “integration of pSEA2”

Line 136 – “homology” is not really the best term here, if they are identical?

Line 142 – suggest “movement of Tn6945 to new locations”

Line 184 – meaning actually “chromosomal copy of Tn6945 in V. alfacsensis is functional”?

Line 198 – suggest “two sequences” rather than “sequence types”

Line 199 – “absence of pSEA-2 from”

Line 206 – “a” of attB needs to be italicised

Lines214-5 – ambiguous and needs rewording, “respective” is probably not needed (nor line 219)

Line 225 – “was highest”

Line 229 – better to say something like increased copy number resulted in increased resistance?

Line 247 – suggest “other conjugative”, might be more logical to mention islands after all of the plasmids?

Lines 347-8 – giving the strain names here is not really necessary.

Lines 393 – this information might be more easily accessible if presented in a supplementary table, with the most relevant accessions quoted in the text.

References – several references are missing page nos. (e.g., Refs 5, 8, 15, but all need checking) and Title Case (i.e., First Letter of Each Word In Upper Case Like This) should be removed ((e.g., Refs 6, 8).

5) Figs.

These could be improved/simplified/split into smaller figures, considering how they would look on a journal page and if they relate to different parts of text.

Fig. 1 - I don’t think that part A is necessary – the information in it is clear from the text. If retained then the brackets need to be explained. Part B(i) is not clear (e.g., the legend does not explain what “CAIM” is and inverting the orientation of pSEA2 would make the comparison clearer) and is probably not needed. I would suggest putting Part C first, then B(ii), with a line underneath showing pSEA2 and then part D.

Fig. 2 could be supplementary. It is not clear what “divided by the donor CFU and recipient CFU” in the legend and “Log(transconjugant/(donor/*recipient)” on the Y axis mean?

Fig. 3B and C could be supplementary. In Part A the Chr2, pSEA and pYa249 diagrams are not really needed and maybe the remainder could be combined with Fig. 1C, B(ii) and D?

Supplementary Information

The order of Figs S1 ad S3 seems to have been switched vs. the legends.

It would be helpful for review if the legends were on the same page as each figure.

Reviewer #2: The manuscript describes the identification of an interesting integrative element containing a bla gene from marine Vibrio. It is detected in chromosomal and plasmid sites and it's transfer and integration into E. coli is demonstrated. These are notable findings. Unfortunately, as noted below, I feel there are a significant number of issues that need attention.

Lines 1 (manuscript title), 133, 135, 164 ,166 and 203 (and possibly elsewhere): To avoid confusion, it would be preferable for movement (translocation) of Tn6945 NOT be described as "transposition"; it is site-specific integration, and it doesn't involve a transposase.

Line 75: "‥.embedded on the plasmid‥." should be "‥.embedded In the ELEMENT‥.".

Line 87: Selection said to include tetracycline and ampicillin, yet next sentence states transfer frequency for tetracycline only. How can this be?

Line 95, Fig 1Bi legend: More description is required - why is CAIM 1831 segment shown? Doesn't actually show "insertion sites", rather comparative locations of Tn6945. This figure doesn't seem to provide any useful information beyond what should be in the text (see below) and hence should be removed.

Line 109: S1 Fig is cited but the legend for S1 is incorrect. The provided S3 legend corresponds to the figure I think. That legend needs to provide the definition for Crick/Watson strands used here since various exist. Is there a meaningful reason why Chr1 of 04Ya108 is shown inverted with respect to Chr 1 of the other strains? If not, it should be reverse complemented for consistency.

Line 112: Rewrite sentence to include that the segment is absent from the corresponding region of CAIM 1831.

Line 114: Indicative values should be provided for "very low gene product identity".

Line 115-117: It is not clear to me what data specifically has "confirmed" that the repeated region is an active integrative element. The evidence for this assertion needs to be stated succinctly. Moreover, if this is the case I find it perplexing that it has been given a Tn number since it is clearly not a transposon. I appreciate that historically some integrative elements were assigned Tn numbers, but the important distinction between IE and Tn (site-specific recombinase verses comparatively random transposase) has been understood for some time. The practice of assigning Tn numbers to IEs promotes confusion and is hence unhelpful; it should not be continuing in 2021 - alternative naming conventions are being used for IEs. A complicating issue here is that at this time it is unclear what type of IE this one represents. It does not appear to be an ICE, but it could be an IME (integrase-like proteins have been found to serve as a relaxase in some conjugative plasmids) or it could be a form of a CIME. In any case, a name that is not misleading would be preferable.

Line 121-124: This section describes insertion sites in E. coli, shown in Fig 1D, but the insertion sites from the Vibrio strains (shown immediately above them in the same figure) are ignored in the text. If they are not informative they should be removed; but surely they are. Is the information consistent between the organisms? I would have thought a consensus motif for attB target sites might be worth deducing/showing?

Line 125: Indicative values should be provided for "very similar".

Line 128: "previously" should be "previous".

Line 130-131: Without first meaningfully describing the similarity relationship between pSEA1 and pSEA2, it is difficult to know whether deducing a precursor based on the presence/absence of a site-specific integrative element is appropriate.

Section starting line 133: This section starts by outlining a plausible scenario for plasmid integration via a two-step process involving homologous recombination. However, analogous mobile element flanked plasmid co-integrate arrangements can also arise in a single step in the absence of homologous recombination. The possibility of this (and possibly other) scenarios needs to be noted here also so the rationale for the subsequent experiments is clear to a reader. The subsequent two paragraphs, and Fig 2, which describe the data, are very difficult to understand in terms of what was done and the data resulting. It all needs to be described much more clearly. Plasmid integration is said to be based on transfer of both ampicillin and tetracycline, but then only tetracycline-resistant transconjugants are described, and only Tet + Ap is shown in Fig 2. Elsewhere in the paper, transfer frequency is provided "per donor" as is established practice. Frequencies indicated in Fig 2 should do likewise, rather than per donor x recipient. Moreover, I think it would be preferable for this data to be provided in a table rather than a graph, so it is clear what data was obtained for each mating. For example, the single dot for Tet+Ap transfer in the delta-recA strain does not really agree with what is stated in the text. What is the explanation for the apparently reduced Ap transfer in the delta-recA strain? Could this be responsible for the failure to detect any Ap + Tet transfer (i.e., insufficient sensitivity)? If so, the data doesn't really provide clear support for the proposed two-step scenario.

Section starting line 168 (to 191) and manuscript title: Copy-out-paste-in has been employed to denote a mechanism of transposition. The mechanism employed by this type of integrative element may indeed share some commonalities (which are certainly worth noting), but the enzymes involved and the biochemistry among other things are going to be different (e.g., the observed strand bias, as noted in the Discussion). I therefore feel that simply using the same Copy-out-paste-in term to describe this integration mechanism will again lead to confusion, leaving some readers with the impression that transposition and integration are the same when they are clearly not. Additionally, mechanisms used by other integrative elements are not explained anywhere in the manuscript. Are they different? Does the data here preclude them? The failure to detect an empty attB site by PCR of itself does not represent compelling evidence that a copy of the element is retained in the donor replicon. Maybe the empty site is not re-joined and is hence lost from the population. Was a positive control performed to show that an attB site could be detected for a different element that is known to generate them? Indeed, could the attB PCR detect the empty site of CAIM 1831? Without additional evidence, it is premature to conclude that a copy-out-like mechanism is used.

Line 215: Why is Tn6283 mentioned here? It doesn't seem to encode resistance so how is it relevant to these studies?

Line 228: Although the results are mostly consistent with a correlation between Tn6945 copy number an Ap MIC, it is an overstatement to say they "clearly" show copy number affects resistance level; there is only a single strain with three copies, some strains with one or two copies exhibit low resistance, while another with one copy exhibits as much resistance as the most resistant isolates with two copies. This suggests that there are other factors at play so the data should be interpreted conservatively.

Fig 4 legend: What do the different colours denote?

Discussion line 256: The conviction in the two step mechanism is not justified by data presented. Other alternatives, and known mechanisms of other integrative elements should be elaborated.

Discussion line 275: The possibility that this element might be mobilisable (i.e., an IME) should be canvassed, particularly given the multiple genes of unknown function. However, all that might be needed is an oriT site. Has a candidate oriT sequence been identified in pSEA1/pSEA2 (likely near their relaxase gene), and is there a similar sequence in Tn6945?

Line 573: The legend shown is presumably for S3 rather than S1. In any case, I doubt enough information is provided for a reader to understand the figure.

Line 581 and S2 legend: "‥.arch‥." should be "‥.arc‥.". I could not find the significance of the "contig 1" described anywhere so it should be removed (from here and the figure). What is the meaning of the gene colours used in B?

Fig S2A: Are putative conjugation or replication initiation genes evident in the pSEA2 sequence? If so, it would be informative if these additional genes at least could be indicated on the figure.

Fig S2B: pSEA1 should be included in the figure label. The legend provided is inadequate - what do the gene colours mean? The figure shows locations of the elements, not "insertion positions", and for both plasmids, not just pSEA1.

6. PLOS authors have the option to publish the peer review history of their article (what does this mean?). If published, this will include your full peer review and any attached files.

Reviewer #1: No

Reviewer #2: No

---

## [Author Response · Author response to Decision Letter 0]

16 Nov 2021

(This space is too small to show all comments and our responses. All responses are described in the attached file.)

Responses to the Reviewer’s comment

PONE-D-21-18582

(Editor’s Comments)

Your manuscript incudes some interesting findings but both reviewers had difficulty following the text and had some concerns about particularly the naming you have used. They have given some quite detailed suggestions all of which look very helpful.

I agree with the reviewers on most points particularly that the text is very hard to follow because the information needed (e.g. sizes of PCR products, predicted sizes of restriction fragments, exactly how PCRs were performed) is not provided.

(Response) Thank you for allowing us to resubmit our manuscript. We sincerely apologize for the inconvenience in interpreting the figures in the previous submission. We found that there were indeed incorrect descriptions about Primers in the Table.

To detect the joint region of the circular form of the integrative element (currently called SE), we showed PCR amplification of 1-kb long attTn (currently called attS) using an additional set of outward SE primers in the previous submission, as the original outward primers (LN_142_junction2, LN_143_junction2) generate a 206 bp product which was very faint in agarose gel in our laboratory setting. The use of two types of outward primers might have confused the readers. In this revision, we have improved the electrophoresis condition such that the 206 bp attS product is sufficiently clear. Therefore, we removed the information on the 1-kb long attS and outward primers used to amplify 1-kb attS in the revised manuscript. Regarding attB, we showed only Long PCR results in the previous submission. This may also be the cause for the confusion. In the revised manuscript, we show results of both standard PCR (neither empty attB nor occupied attB was detected) and Long PCR (only occupied attB was detected) in new Fig. 4. In this revision, we corrected the primer table and updated the figure panel for PCR results. We added a restriction map panel to the Southern hybridization figure and used more words to explain (i) Southern hybridization results (new S5 Fig) and (ii) att sequences (new Fig 4, S3 Fig, S4 Fig): details are provided in the response to the reviewers section below. We believe that the revised manuscript contains sufficient information to explain the experimental results.

(Editor’s Comments)

I have a few comments to reinforce or add.

1. Please do not use Tn with a number for an IE. The reviewers have explained why.

(Response) We appreciate your comment and understand the rationale behind your suggestion. However, currently, there are no definite rules proposed for naming newly identified IE, except for established IE groups, such as ICE and IME. In the revised manuscript, we have explicitly stated that the unclassifiable integrative elements, identified in this study, are a new class of mobile elements. Therefore, we have used an “SE-“ prefix (strand-biased circularizing integrative element) to refer to these IEs in the revised manuscript. We believe that using numbers to identify these DNA elements would make it easy to handle (refer to) them in existing and future mobile DNA databases. Therefore, we have used numbers to refer to the SEs in this manuscript. Regarding the use of the word “transposon/transposition” to refer to the mobile DNA elements, while we are aware that there are scientists who prefer to confine the usage of the term “transposon” to refer to a few classes of mobile elements that use DDE transposase (i.e., IS-composite transposons or unit transposon such as Tn3), the term “transposon” has been explicitly defined as “specific DNA segments that can repeatedly insert into one or more sites in one or more genomes” in Roberts et al. 2008 Plasmid, or “a defined segment of DNA that has the ability to move, or copy itself, into a second location without a requirement for DNA homology” in Curcio and Derbyshire Nat Rev Mol Cell Biol 2003. These definitions share the concept with mobile elements in eukaryotes, and the term transposon is not defined based on the type of DNA strand exchange enzymes used. SEs, handled in this study satisfy these definitions. ICEs are classified as transposons that move using Y-transposase (tyrosine-recombinase) or S-transposase (serine-recombinase) in a comprehensive transposon review by Curcio and Derbyshire (Nat Rev Mol Cell Biol 2003). Regarding mobile DNA in prokaryotes, IS91 uses HUH endonuclease, IS607 uses serine recombinase, Casposons uses CAS nuclease for their location changes, but they have all been called transposons.

(Editor’s Comments)

2. Please do not use “copy-out paste-in”, which is a specific transposition route and not applicable to integration. It’s known that some tyrosine recombinases perform only one strand exchange. Replication is then needed. This route could give the outputs you observe. The first Int system of this type was IntI1 the integron integrase and this was discovered about it in the 1990s. Please consider this.

(Response) We would like to argue with the Editor regrading this point respectfully. “Copy-out paste-in” is the term used in Curcio and Derbyshire (Nat Rev Mol Cell Biol 2003) to classify the type of movements of transposons from a wide range of organisms after the discoveries of ICE, IS91, and IS607, which do not use DDE transposase, though we admit that the word was initially used for the specific IS family. This classification of movements by Curcio and Derbyshire is based on the route/consequences and not based on the proteins used. While we understand the Editor’s concern, we still think that referring to “copy-out paste-in” is particularly useful to explain the route of transposition of SEs. We illustrated all possible routes of SE circle generation in new Fig 4 A, including “copy-out paste-in,” integron gene cassette excision, and ICE/IME excision. Then we explained how the movement of SEs is different from the known routes. As SEs likely use tyrosine recombinases, they require DNA protein crosslink repair to achieve the IS3-like copy-out (to generate 3′OH). Thus, it is indeed not strictly identical to the movement of the original “copy-out paste-in” transposition of IS3, as pointed out by Reviewer 2. We also thank the Editor for spending time speculating the movement of SEs. We assume that the Editor talked about the “replicative resolution model” (Escudero et al., Microbiology Spectrum 2015) of gene cassette excision (cut-out of single-strand DNA circle accompanying generation of empty attC). The SE circle generation does not accompany the generation of empty attB. Therefore, the control of strand exchange is obviously different between Integron and SE. This point is mentioned in the revised manuscript; Line 282-288 in Result and Line 461to 472 in Discussion, and new Fig 4 legend (line 310-333). We newly cited literature about discoveries of attI and 59 bp elements (attC) to emphasize the dissimilarity between attB of SE and attI/attC of Integrons (Stokes and Hall 1989; Stokes et al. 1997), as well as literature relevant to the attC-attC recombination (Collis and Hall 1992; Burrus and Waldor 2003; Escudero et al. 2015): line 214, and line 289 in the revised manuscript.

(Editor’s Comments)

3. Please reduce the use of transposon references and information and replace it with Integrase or IE stories.

(Response) Two transposon papers were removed. However, a few important IS papers are still cited because we think that this research appeals to all audiences studying mobile elements, and we think that dissimilarities (i) between SE and IS, (ii) between SE and gene cassette, and (iii) between SE and ICE/IMEs are all worth stating as emphasized in new Fig 4. We thank the Editor for reminding us of Integron. We agree that literature relevant to the Integron gene cassette was indeed important but missing. We think that the citation of Integron papers/review (Stokes and Hall 1989; Stakes et al. 1997; Collis and Hall 1992; Escudero et al. 2015), ICEs/IME papers/reviews (Caparon and Scott 1989; Auchtung et al. 2005; Burrus and Waldor 2003; Naito et al. 2016, and others), and CTX phage (Val ME et al. 2005) which are relevant to tyrosine recombinase, and IS (Polard &Chandler 1995; Rousseau et al. 2008) is sufficiently fair. We also emphasized the difference between ICE/IMEs and SEs in line 490-497 in the revised manuscript.

(Editor’s Comments)

4. Conjugation frequencies must be expressed as per donor. Please fix this. I will accept scatter plots.

(Response) We updated the figure to comply with this (new Fig 5).

(Editor’s Comments)

5. A strain Table is needed.

(Response) We created a strain table (new Table 1). The information on all transconjugants used is included.

(Editor’s Comments)

6. I was surprised that 2 genes should be called Int as they has such different sizes. So, I checked them using BLAST and found that IntA has the required domain but IntB has no known domains. To see domains, once the search is complete click on the graphic summary and look right at the top. Hence, I insist that this orf be annotated as an orf.

(Response) We appreciate that the Editor showed interest in IntB, and conducted blast searches. As notified by the Editor, at first glance, IntB does not show similarity to known tyrosine recombinase. Homology to Integrase C-terminal domain can be detected by increasing the E-value threshold from 0.01 to 0.05 in Conserved Domain search (which is used by the Editor). The hit subject was cl00213: DNA_BRE_C Superfamily (DNA breaking-rejoining enzymes, C-terminal catalytic domain) in the CD-search. IntB also generated a hit in the Pfam database, and the hit subject was Phage_integrase (PF00589). In this revised manuscript, we made an alignment of IntA, IntB with known tyrosine recombinases, including IntI of R388, using structure-aware prediction program PROMAL3D. We showed that both IntA and IntB possess catalytic (G)-R-H-R-(H)-Y motifs in new Fig 1B. The relevant statements are shown in lines 129-137 in the revised manuscript.

(Editor’s Comments)

7. I also noticed that in the database entry for pSEA2, intA is not annotated as such. Please fix this.

(Response) We have added intA, intB of (SE-6283, and SE-6945) to the Genbank file of pSEA1 (AP024167), pSEA2 (AP019851), Chromosome 1 of 04Ya249 (AP019850), Chromosome 1 of 04Ya108 (AP024165). 

(Editor’s Comments)

8. IE encoding the same or closely related Int should integrate at the same position. Is your IE in the same gene in Vibrio and E. coli? More on this is needed.

(Response) We appreciate the Editor’s comment. Insertion location of SE-6945 was different between the E. coli genome and Vibrio genome (new Fig 2). Interestingly, the insertion location of SE-6283 was the same in the E. coli genome and Vibrio genome (new Fig 6). 

(Editor’s Comments)

9. The searches I did revealed that there are several IE of this type in Vibrio species. Are they in the same location?

(Response) We conducted additional bioinformatic surveys using SE-6945 and SE-6283 core genes as queries to identify their closely related SEs. We focused on closely related SE because target site preference may be different among distant SE members. We identified new members of SE-6945-related elements and SE-6283-related elements in a different Vibrio species, inserted into the equivalent loci (bcp for the SE-6283 related elements; yaaA promoter for the SE-6945 related elements) as predicted by the Editor. Due to an increase in the information of the result section, we substantially reorganized figures related to attB, attL, and attR. In the revised manuscript, attB of SE-6945 is discussed in Lines 210-216 using new Fig 2 and new S3 Fig and new S4 Fig, while attB of SE-6283 is discussed in lines 399-415 using new Fig 6, S7 Fig, and S8 Fig. We are aware that distantly related SEs are present in the database. We think that the analysis for entire SE members requires a huge amount of bioinformatic analysis, and their target sites are likely different. Thus, we think that those analyses are beyond the scope of this study.

(Reviewer #1’s Comments)

Reviewer #1:

1. This manuscript describes basic characterisation a non-conjugative integrative element (named “Tn6945”), carrying a beta-lactamase (bla) gene, on the chromosome and a plasmid of a Vibrio ponticus isolate from an aquaculture site in Japan. The element was identified following conjugation with E. coli, when either it or the whole plasmid was transferred to the chromosome, the latter process identified as being dependent on recA. “Tn6945” is related to “Tn6238” previously identified by the same group on a related plasmid in a different Vibrio species. The manuscript seems to imply that “Tn6945” is only the second example of this element to be identified (it would be helpful to make this more explicit if this is the case), in which case describing it would be useful. The approach/results appear technically sound, but it is a bit difficult to tell from the descriptions provided, and the conclusions generally justified, but naming these elements transposons is problematic (see point 1 below). Also, the manuscript requires some reorganization rewriting and condensing, as it not easy to follow and is quite wordy and repetitive in places (see examples below).

(Response) We appreciate the reviewer for showing an interest in this work despite the presence of an incorrect description in the results in the original submission. We understood the reasons why the reviewer was confused. In the revised manuscript, we used more words to explain the Southern hybridization and PCR results and dissimilarity in movements between Tn6945(SE-6945) and Integron gene cassette or ICE/IMEs. Incorrect descriptions about primers and electrophoresis panels have been replaced with corrected versions (details are stated below). We use the term “transposon” based on the definition “specific DNA segments that can repeatedly insert into one or more sites in one or more genomes” (Roberts et al. 2008 Plasmid) while admitting that some scientists prefer to confine its usage to unit transposon (i.e., Tn3, Tn7) or IS-composite transposon encoding DDE transposase. The term “transposition” has been defined as “the movement of discrete segments of DNA without a requirement for homology” (Curcio and Debyshire 2003 Nat. Rev. Mol. Cell. Biol), and not based on the type of strand exchange enzyme used. In the introduction section of the revised manuscript, we cited transposon reviews about the definition (Roberts et al. 2008 Plasmid) and transposon classification based on strand exchange enzymes and movement (Curcio and Debyshire 2003 Nat. Rev. Mol. Cell. Biol). In the review, ICE/IME was classified as transposons that use Y- or S-transposase (tyrosine recombinase or serine recombinase). We added these explanations for the terms in lines 53-72 in the revised manuscript. As Tn6283/Tn6945 are currently unclassifiable into any mobile element classes proposed to date, we coined the term SE (strand-biased circularizing integrative element) to refer to these integrative elements in the revised manuscript. Upon request from the editor, we conducted an additional database survey and found several new members of SEs carrying a different set of cargo genes while sharing only the 4 core genes (new Fig 2 and new Fig 6). Therefore, we think that SEs are worth being proposed as a new mobile element family. The confusion about the word usage may also stem from the fact that this study needs to handle several distinct intracellular DNA movements using similar or identical words. We, therefore, explicitly stated the rule for the word usage in this manuscript for “transposition,” “integration,” “insertion,” and “translocation” in the introduction section of the revised manuscript (lines 96-107). This would help readers regardless of their backgrounds.

(Reviewer #1’s Comments)

2. “Non-conjugative integrative element” vs transposon

Description of “Tn6945”/”Tn6238” elements as a transposons is at best confusing, when they carry tyrosine recombinases rather than true transposases. Inconsistencies/contradictions in the text suggest that the authors are not really sure how to describe them either e.g., “nonconjugative integrative element Tn6283” (line 62), “a new type of transposon” (line 66), “Tn6283-like integrative elements” (line 69), “active integrative element” (lines 115-6) and their is reference to them having both IR (typical if transsponsons) and att sites (typical of integrative elements). While some ICE were historically given Tn numbers it is not helpful to perpetuate such confusion and the nomenclature of these “Tn6945”/”Tn6238” elements needs to be carefully considered and if these are a novel type of element then this needs to be clearer.

(Response) As stated above, our standpoint is that ICE/IMEs should be handled as one class of transposons based on the idea in the review by Curcio and Derbyshire 2003 Nat Rev Mol Cell Biol. We used the “SE-“ prefix to refer to each SE member in the revised manuscript. We prefer to use 4-digit numbers to keep this compatible with the existing Tn database, and in part considering the ease of handling them in future databases. We used the terminal inverted repeat motifs as a general meaning but not specific TIR of ISs. In the revised manuscript, this phrase was deleted.

(Reviewer #1’s Comments)

3. Lines 111-131 – descriptions of insertions. This section needs to be written more clearly/reorganised/condensed.

Line 111 – this is called “MULTISPECIES: class A beta-lactamase” by NCBI RefSeq. It might be useful to obtain a name for it from NCBI, – “a 7.1 kb repeat region”.

(Response) We requested an update of Genbank annotation files to include gene names suggested by AMRFIndePLuS under “CDS feature”. All AMR-relevant genes in AP019849, AP019850, AP019851, AP024165, AP024166, AP024167 released by now have gene names. “Gene feature” was not amendable by the submitter. It will be automatically updated or added, when necessary, by GenBank.

(Reviewer #1’s Comments)

4. Line 114 – it might be better to avoid “homology” here? Gene product identity would be higher than coding sequence identity, so this might be written the other way round.

(Response) In the revised manuscript, we showed the Int protein alignment as new Fig 1B. Therefore, it is sufficiently clear for the audience that the 4 coding sequences embedded in SE-6283 and SE-6945 are not independently evolved hypothetical proteins but orthologous proteins; thus, using homology (meaning phylogenetic relation) is correct (line 125). 

We showed the gene product identify values as follows in the revised manuscript as follows: “gene product identities between SE-6945 and SE-6283 counterparts were very low: 22.5 % for IntA, 20.5 % for CDS2, 23.2 % for IntB, and 26.6 % for CDS4 in blastp” in lines 126-128.

(Reviewer #1’s Comments)

5. Lines 115-116 – the evidence for this is from the conjugation experiments? This is not clear.

(Response) We used the term “active” based on the evidence for (i) the circular form generation in the absence of pSEA2 in V. alfacsensis (new Fig 4, previous Fig 3) and plasmid-free E. coli (new S6 Fig) and (ii) new junction generation (attL, attR) in E. coli during conjugation experiments. We rephrased this sentence to explicitly state this fact as “This 7.1-kb region was confirmed to be active in generating its circular copy of top strand in E. coli and V. alfacsenis and inserting it into new target locations in E. coli in this study” in lines 129-131.

(Reviewer #1’s Comments)

6. Lines 120-121 – what is the evidence for targeting, as opposed to random insertion?

(Response). The Southern hybridization experiments (the current S5 Fig) were performed to see the target site preference of SE using 19 transconjugants obtained from independently performed mating experiments. However, we forgot to write this motive in the previous submission. In the revised manuscript, we have added a new paragraph regarding the target site preference and have discussed this in lines 181 – 225.

(Reviewer #1’s Comments)

7. Lines 122-4 – “Tn6945” inserts itself, rather than its IR? It’s not clear whether it is being proposed that “Tn6945” has IR, like a typical e.g., Tn3 family transposon (and why are the repeats called C and C’?) which are internal to such Tn, or it has an att site that recombines with target att site to create hybrid sites, as for integrative elements. att site are mentioned in the Fig. 1 legend but are not discussed until later in the text, which is also a problem.

(Response). We understand the reviewer’s concern. We changed the phrase to “SE-6945 inserts itself ending with 5’-GTA-3 (Fig 1 A) along with an additional 6 bp next to the 5’ end of C at attL on the donor molecule into the target site” (line 177-178). C and C’ are from core site of site-specific recombination mediated by tyrosine recombinase (i.e. Lambda Int). The problem with this new integrative element is that we cannot deduce how much longer does the cis-element involved in recombination extend from motif C/C’. Therefore, we simply defined the term attL and attR as “The border regions between SE-6945 and its target locations” and then stated “The border regions between SE-6945 and its target locations were termed attL (intA-proximal side) and attR (intA-distal side) (Fig 2 A)” in the revised manuscript (line 169-170). We also elaborated to explain the feature around nicking sites at attL and attR at lines 170-176 and lines 210-216.

(Reviewer #1’s Comments)

Other scientific points

8. Lines 63, 85, 88, 354, 359 - why was 42 deg C used? This needs to be explained.

(Response) 42°C was used to suppress the growth of Vibrio donor strain on the transconjugant selection plate. We have stated this in the Method section in the revised manuscript (lines 567-568). When we started this research, we did not have rifampicin-resistant mutant of E. coli. Therefore, we used 42°C selection alone to select E. coli from the mating mixture. After obtaining TJ249 and its sequencing, we revisited the experiment design, and started using both rifampicin and 42°C.

(Reviewer #1’s Comments)

9. Lines 81 - it might be useful to make it clear when citation are to previous papers from the same group of researchers (i.e., use “We”).

(Response) In the revised manuscript, we started with “We previously isolated V. alfacsensis strain 04Ya249 from sea sediment at an aquaculture site,” as suggested (line 111-112).

(Reviewer #1’s Comments)

10. Lines 106, 380-1 – is this version of PacBio alone sufficient to give accurate sequences?

(Response) The oldest reads were obtained in 2016 from PacBioRSII at CLR mode; coverage was > 120x, the chemistry was P6C4 chemistry. Currently, error rate information of that chemistry is not available from Pacbio. However, even under the older setting used in 2013 (https://www.pacb.com/uncategorized/a-closer-look-at-accuracy-in-pacbio), the accuracy exceeds QV 60 (99.999%) when the coverage became higher than 25x. Since we obtained 120x coverage, the accuracy of the consensus genome is equivalent to the current HiFi reads or illumina reads assembly. We added the statement “using P6C4 chemistry at Macrogen (Tokyo, Japan) in 2016” in line 623-624 to indicate that these reads were obtained after 2013.

(Reviewer #1’s Comments)

11. Line 109 – is the accession no. for this given anywhere? 

(Response) Accession no. of the genome of 04Ya249 is given in the Accession numbers section. Strain TJ249 is a laboratory strain, and thus we think its genome is not worth registration in Genbank/DDBL/EMBL database. However, its raw reads and assembly are available from figshare (the link is listed in reference in the previous submission) as stated in the method section and new data availability table new S2 Table.

(Reviewer #1’s Comments)

12. Lines 179-80 – att need explaining before this point in the manuscript

(Response) We added explanation for the att sites as follows: The border regions between SE-6945 and its target locations were termed attL (intA-proximal side) and attR (intA-distal side) (Fig 2 A)” (line 169-170), ”unoccupied target site in the host (attB)” (line 173) in the revised manuscript.

(Reviewer #1’s Comments)

13. Line 223 – what concentration of ampicillin?

(Response) 100 ug/ml as stated in the Strains, plasmids, and culture media section in Methods: line 553-554.

(Reviewer #1’s Comments)

14. Line 240 – V. cholerae is not always from sea food?

(Response) V. cholerae is indeed not always from sea foods as described in the cited reviews. These descriptions are correct. Thus we did not make changes to the revised manuscript.

(Reviewer #1’s Comments)

15. Line 250 – the term “super-integron” should not be used. Use e.g., “sedentary chromomsomal integron” instead.

(Response) We changed the super-integron to “sedentary chromosomal integron” in the revised manuscript in line 439.

(Reviewer #1’s Comments)

16. Line 264-66 – is “Tn6945” the smallest of only two of this type of elements to have been identified to date? Or are there other examples in addition to Tn6283? This needs to be clear, i.e., state in the Introduction that is a new type of element, if this is the case.

(Response) “Tn6945/SE-6945” is the smallest of only two of these types of elements to have been identified to date. In the revised manuscript, we first called Tn6283 as “unclassifiable integrative element Tn6283” in line 80, then stated that “we have termed currently unclassifiable integrative element, SE-6283 and its related elements as a new class of mobile elements: strand-biased circularizing integrative element (SE)” in line 93-95 in the Introduction. In this revision, we show more examples of SEs in new Fig 2 and new Fig 6 based on our new survey.

(Reviewer #1’s Comments)

17. Lines 266-70 – is it possible to tell anything about how the bla gene might have ended up in this location? Is there really evidence for an “ancient origin”?

(Response) We omitted the relevant sentience in the revised manuscript.

(Reviewer #1’s Comments)

18. Lines 275-6 – these genes need to be described in Results.

(Response) We appreciate the reviewer’s comment. We explicitly stated that “The biochemical functions of their products were estimated using Pfam and InterProScan in EMBL. The intA and intB products generated a hit on phage_integrase (CL0382) in Pfam and were further aligned with known tyrosine recombinases using structure-aware alignment program PROMAL3D [32] to confirm the presence of conserved motifs. The intA and intB products contained the catalytic R-H-R-Y motif (G-R motif in box I and H-R-H-Y motif in box II (Fig 1B)) conserved in tyrosine recombinases [33], though the N-terminal region of IntB is extraordinarily long among tyrosine recombinases. CDS2 and CDS4 products did not generate hits in the Pfam or InterProScan databases.” in lines 133-140 

(Reviewer #1’s Comments)

19. Lines 278-80 – what is the evidence of this for the “latter” i.e., Tn6283? This is not really clear from the Results and both references are about IS.

(Response) We used more words and figure panels to explain the results in the revised manuscript (lines 283-296, lines 304-308, lines 342-359; new Fig 4 panels A, D, E, new Fig 4 legend). We believe that new Fig 4 would help readers understand our interpretation: the SE’s movement is more like IS than Integron gene cassette or ICEs.　The previous reference for the IS911 review indeed stated that in its Fig 5B, “cleavage of the left or right inverted repeat (IR) and attack of the other end”, and it generates a single-strand DNA bridge: this means that both strands can be nicked, but they do not happen simultaneously, and a single-strand bridge is generated using the top or bottom strand. In the previous reference for the IS (Kosek et al 2016, Figure 2 C) indicates the presence of top strand exchange and bottom strand exchange products, and there is a statement “This suggests that TnpA can use either TIR to strand transfer into the other”.　For clarity, we changed the cited articles to Polard 1995, which focused on the reaction of single stand bridging (figure-eight formation) (Polard 1995) and the bias of two IRs of IS911 (Rousseau 2008). We think that current references are more appropriate to argue about the presence of strand bias of single-strand DNA bridging.

(Reviewer #1’s Comments)

20. Lines 292-307 – a table explaining the different strains, with some additional details in the text in the case of more complex constructions, might be clearer. The first sentence should be deleted and the citations included in other places.

(Response) We made a new Table of strains and plasmid list as new Table 1. SE-insertion location in each strain is shown in the new Table 1. The first sentence was replaced with “Strains and plasmids used in the study are listed in Table 1” (line 508).

(Reviewer #1’s Comments)

21. Line 347-59 – as some strains used are transconjugants, it might be more logical to describe conjugation experiments earlier in Methods? Line 355 is ambiguous and needs to be written more clearly.

(Response) The section on the conjugation method was moved to an earlier position in the revised manuscript. Then, we have described the details of the selection conditions and motives of the experiments in lines 567-579

(Reviewer #1’s Comments)

22. Lines 364-8 – digestion should be described before hybridisation and something like “using the 5'end of intIA, amplified using primers LN112 and LN113 (Table 1) and the PCR DIG… as probe” would be simpler.

(Response) We thank the reviewer for this suggestion. We changed the order of phrases and then added more explanation for the method in the revised manuscript (line 604-605)

(Reviewer #1’s Comments)

Minor/formatting/wording/English etc

Some streamlining/reducing repetition in different sections etc. is needed, as well as some reorganization. Some examples are given below.

23. Line 61 – “The” is not needed here.

(Response) “The” was deleted in the revised manuscript. Line 79.

(Reviewer #1’s Comments)

24. Lines 63-4 – suggest “…, following conjugation it could integrate into the chromosome by…”

(Response) We rephrased that sentence as suggested: “Although pSEA1 could not replicate in E. coli at 42 °C, following conjugation, it could integrate into the E. coli chromosome by homologous recombination between two Tn6283 copies: one on pSEA1 and another that moved from pSEA1 into the chromosome”.

(Reviewer #1’s Comments)

25. Line 71 – “in this study we mated”

(Response) We removed ‘,’ as suggested. Line 89.

(Reviewer #1’s Comments)

26. Lines 84-91 are unclear and need rewording.

(Response) In the revised manuscript, we split descriptions of the conjugation experiments for three different purposes into three distinct sections (line 111, a section starting at line 182, a section starting at line 228). Now, this paragraph only describes the experiment used to isolate TJ249. We hope that this has made it more clear.

(Reviewer #1’s Comments)

27. Lines 110 etc – change “beta” to beta symbol

(Response) The beta of beta-lactamase was changed to beta (β) symbol in the revised manuscript (line 28; line 110; line 122; line 124; line 889)

(Reviewer #1’s Comments)

28. Lines 118, 120 etc – suggest present tense for facts that are still true i.e., “contains”. “indicates” etc.

(Response) We appreciate the reviewer’s suggestion. We modified the text as suggested. Lines 161 and line 167 in the revised manuscript.

(Reviewer #1’s Comments)

29. Line 125 – “the previously reported plasmid” can be deleted.

(Response) We deleted “the previously reported plasmid” in the revised manuscript (line 155).

(Reviewer #1’s Comments)

30. Line 127 – can delete “that of”.

(Response) We deleted “that of” in the revised manuscript (line 157 in the revised manuscript).

(Reviewer #1’s Comments)

31. Line 128 – could be condensed to “…from Southern hybridization (19)”

(Response) We deleted “in our previous study” as suggested (line 158).

(Reviewer #1’s Comments)

32. Lines 129-30 – suggest “carries Tn6945 in the same place as in…”

(Response) We used “carries SE-6945 in the same two locations as” in the revised manuscript (line 159-160).

(Reviewer #1’s Comments)

33. Line 134 – “pSEA2” would be clearer than “the plasmid” here. Also line 143 – “integration of pSEA2”

(Response) We used “pSEA2” as suggested in line 229 in the revised manuscript. A phrase corresponding to the previous line 143 no longer exist in the revised manuscript.

(Reviewer #1’s Comments)

34. Line 136 – “homology” is not really the best term here, if they are identical?

(Response) We use a more direct expression in the revised manuscript as follows: “by homologous recombination between SE-6945 copies: one on the chromosome and the other on pSEA2” in the revised manuscript”. Line 230-231.

(Reviewer #1’s Comments)

35. Line 142 – suggest “movement of Tn6945 to new locations”

(Response) Homology independent location change of DNA segment is actually identical to the term “transposition” as stated in the Curcio and Derbyshire review (2003), regardless of the strand exchange enzymes used. We wish to follow it (for example, line 226; a phrase corresponding to the previous line 142-143 no longer exist in the revised manuscript). For clarity, the rules of word usage in this manuscript are described in the Introduction of the revised manuscript (line 96-107). 

(Reviewer #1’s Comments)

36. Line 184 – meaning actually “chromosomal copy of Tn6945 in V. alfacsensis is functional”?

(Response) Because only one SE-6945 copy is present in LN95, detection of circular SE-6945 in LN95 indicates that the cis-elements (attL/attR) and strand exchange enzymes encoded in the chromosomal SE-6945 is indeed functional, while those of SE-6945 on pSEA2 is obviously functional as its transposition was observed in E. coli. We explicitly stated that “This suggests that the cis-elements (attL/attR) and strand exchange enzymes encoded by the chromosomal copy of SE-6945 are also functional in V. alfacsensis” in lines 304-306 in the revised manuscript.

(Reviewer #1’s Comments)

37. Line 198 – suggest “two sequences” rather than “sequence types”

(Response) We used “two sequences” as suggested in the revised manuscript (line 349).

(Reviewer #1’s Comments)

38. Line 199 – “absence of pSEA-2 from”

(Response) We are afraid that we could not find the relevant points in the manuscript. We used more words to explain the strand bias in attL x attR recombination: “These two types of spacer sequences can arise from both pSEA2 and chromosome 1 (Fig 4D). However, only one sequence (attS type 1) was detected in the pSEA2-free strain LN95 (Fig 4E), indicating that strand exchange occurs only on the top strand in attL � attR recombination on chromosome 1 (Fig 4C). The occurrence of attS type 1 and attS type 2 in 04Ya249 (Fig 4E) agrees with the top strand exchange in attL × attR recombinations on chromosome 1 and pSEA2, respectively (Fig 4D).” We believe that these words would suffice: lines 351-357. 

(Reviewer #1’s Comments)

39. Line 206 – “a” of attB needs to be italicized

(Response) We corrected attB in line 336.

(Reviewer #1’s Comments)

40. Lines 214-5 – ambiguous and needs rewording, “respective” is probably not needed (nor line 219)

(Response) We deleted “respective”. We also clearly stated the observed pSEA2 integration pattern in all transconjugants in a new paragraph in line 181-199 in the revised manuscript. The sentences are now as follows: “Southern hybridization analysis of other JW0452 transconjugants revealed one to two copies of SE-6945 in their genomes (S5 Fig). Location of SE-6945 insertion became evident based on Southern hybridization for 15 among 19 transconjugants” (line 376-377).

(Reviewer #1’s Comments)

41. Line 225 – “was highest”

(Response) We deleted “the” in the revised manuscript (line 383).

(Reviewer #1’s Comments)

42. Line 229 – better to say something like increased copy number resulted in increased resistance?

(Response) We rephrased the sentence as suggested (line 386-387). In this revision, we elaborated to explain the Southern hybridization results. We coincidently found that insertion location also affects MIC (new S1 Table). A new statement was added at the bottom of this section as follows: “MIC of individual strain is shown in Table S1. When focused on a class of MIC 500, there was a trend that strains with SE-6945 insertion into insJ show a higher MIC than strains with insertion into yjjNt. Thus, the location of SEs can also affect phenotype variation, possibly due to the differences in the expression levels of their cargo genes.” (lines 388-391)”.

(Reviewer #1’s Comments)

43. Line 247 – suggest “other conjugative”, might be more logical to mention islands after all of the plasmids?

(Response) We have placed ICE and IME side by side in the manuscript (line 438).

(Reviewer #1’s Comments)

44. Lines 347-8 – giving the strain names here is not really necessary.

(Response) Strain names were deleted (line 559).

(Reviewer #1’s Comments)

45. Lines 393 – this information might be more easily accessible if presented in a supplementary table, with the most relevant accessions quoted in the text.

(Response) We changed the paragraph title from accession numbers to Data availability (line 642). The accession numbers are summarized in the new S2 Table in the revised manuscript.

(Reviewer #1’s Comments)

46. References – several references are missing page nos. (e.g., Refs 5, 8, 15, but all need checking) and Title Case (i.e., First Letter of Each Word In Upper Case Like This) should be removed ((e.g., Refs 6, 8).

(Response) We apologize for the inconvenience. We corrected the page numbers and modified the title case to be consistent.

(Reviewer #1’s Comments)

Figs.

These could be improved/simplified/split into smaller figures, considering how they would look on a journal page and if they relate to different parts of text.

47. Fig. 1 - I don’t think that part A is necessary – the information in it is clear from the text. If retained then the brackets need to be explained. Part B(i) is not clear (e.g., the legend does not explain what “CAIM” is and inverting the orientation of pSEA2 would make the comparison clearer) and is probably not needed. I would suggest putting Part C first, then B(ii), with a line underneath showing pSEA2 and then part D.

(Response) We have deleted panel A. Genome comparisons figure (panel B) was incorporated to new Fig.2, while attL attR figure panel (previous panel C/D) was moved to S3 Fig, S4 Fig, Fig 4, due to increase in related data. We use new Fig 1 just for explaining the genetic features of SEs.

(Reviewer #1’s Comments)

48. Fig. 2 could be supplementary. It is not clear what “divided by the donor CFU and recipient CFU” in the legend and “Log(transconjugant/(donor/*recipient)” on the Y axis mean?

(Response) Showing transfer frequency by log (T/(RxD)) has been becoming more popular (https://www.ncbi.nlm.nih.gov/pmc/articles/PMC3687034;
https://www.frontiersin.org/articles/10.3389/fmicb.2020.02070/full)

as its data distribution becomes more like a normal distribution, and thus parametric tests (such as ANOVA) can be applied when comparing data. As our research does not really require a parametric test, we used a traditional “T/D” presentation per request from the Editor in the revised manuscript. 

(Reviewer #1’s Comments)

49. Fig. 3B and C could be supplementary. In Part A the Chr2, pSEA and pYa249 diagrams are not really needed and maybe the remainder could be combined with Fig. 1C, B(ii) and D?

(Response)We moved the gel image for strain 04Ya249 and qPCR result to supplementary material (new S6 Fig). We updated the figure panel of PCR design to explain the copy-out model in more detail (new Fig 4). As we found the attL, attR sequence useful when explaining the copy-out model, we combined previous Fig. 1 D with the PCR experiment result and design, then generated new Fig 4 panel C and D in this revision.

(Reviewer #1’s Comments)

Supplementary Information

50. The order of Figs S1 ad S3 seems to have been switched vs. the legends.

(Response) We apologize for the incorrect descriptions. We confirmed that Figure legends and supplementary figures match in the revised manuscript.

(Reviewer #1’s Comments)

51. It would be helpful for review if the legends were on the same page as each figure.

(Response) In the PLOS authors guide, it is stated that “List supporting information captions at the end of the manuscript file. Do not submit captions in a separate file”. Therefore, we did not add legends in Supplementary figures.

(Reviewer #2’s comments)

Reviewer #2: The manuscript describes the identification of an interesting integrative element containing a bla gene from marine Vibrio. It is detected in chromosomal and plasmid sites and it's transfer and integration into E. coli is demonstrated. These are notable findings. Unfortunately, as noted below, I feel there are a significant number of issues that need attention.

1. Lines 1 (manuscript title), 133, 135, 164, 166 and 203 (and possibly elsewhere): To avoid confusion, it would be preferable for movement (translocation) of Tn6945 NOT be described as "transposition"; it is site-specific integration, and it doesn't involve a transposase.

(Response) As stated above, we wish to use the terms of transposon/transposition defined by mobility of DNA segments and not by the type of DNA strand exchange enzymes, following the published reviews (Robers et al Plasmid 2008, Curcio and Derbyshire Nat Review Mol Cell Biol 2003), while we understand that some scientists prefer to confine their usage to a few families of transposable elements in prokaryotes. Furthermore, particularly in this work, the use of the term “integration” to refer to “location change” is incorrect in that integration is just the last step of the location change, and it would rather confuse the readers. The term translocation has been used to refer to the rarely occurring location change of chromosomal segments in organisms carrying multiple chromosomes, and thus is not suited to refer to the repeatedly occurring movements of mobile DNA mediated by a sequence-specific strand-exchange enzyme.　Therefore, in the revised manuscript, we first mentioned the word definition and transposon classification by citing reviews (Robers et al., 2008 Plasmid.; Curcio and Derbyshire 2003 Nat Rev Mol Cell Biol): ICEs are classified as transposons that use Y- or S-transposase in Curcio and Derbyshire 2003), then we mentioned rules of term usage in this manuscript at the end of introduction section (line 94-105). We believe that these changes would help all readers who have different images for the word “transposon,” including readers studying genetics in eukaryotes. The current tile no longer contains “transposition” or “movement”. 

(Reviewer #2’s comments)

2. Line 75: "‥.embedded on the plasmid‥." should be "‥.embedded In the ELEMENT‥.".

(Response) One AMR gene (bla) is embedded in SE, but other AMR genes were embedded outside the SE. Those AMR genes outside the SE-6945 are also integrated into E. coli (which is out of the host range of plasmid) using homology of SE-6945. In the revised manuscript, the relevant phrase was deleted. But we still think that our previous description was correct. 

(Reviewer #2’s comments)

3. Line 87: Selection said to include tetracycline and ampicillin, yet next sentence states transfer frequency for tetracycline only. How can this be?

(Response) In the revised manuscript, we stated “Tc Ap resistance” instead of “tetracycline resistance” in line 243.

(Reviewer #2’s comments)

4. Line 95, Fig 1Bi legend: More description is required - why is CAIM 1831 segment shown? Doesn't actually show "insertion sites", rather comparative locations of Tn6945. This figure doesn't seem to provide any useful information beyond what should be in the text (see below) and hence should be removed.

(Response) We understand the reviewer’s concern. In the revised manuscript, we deleted CAIM 1831 genome from the insertion location comparison figure panel (new Fig 2 A), and showed only its unoccupied attB in new Fig 2B.

(Reviewer #2’s comments)

5. Line 109: S1 Fig is cited but the legend for S1 is incorrect. The provided S3 legend corresponds to the figure I think. That legend needs to provide the definition for Crick/Watson strands used here since various exist. Is there a meaningful reason why Chr1 of 04Ya108 is shown inverted with respect to Chr 1 of the other strains? If not, it should be reverse complemented for consistency.

(Response) We corrected the legends accordingly. We added the following modification in the legend in the revised figure legend: “Purple dots indicate a match on the Watson strand (5’ to 3’ on the top strand in Genbank file), and light blue indicates a match on the Crick strand (5’ to 3’ on the bottom strand)”: line 873-874 in the revised manuscript. We arranged Genbank annotation files such that the dnaA position comes first and coded on the “Watson Strand”. There is a small inversion around the dnaA gene in 04Ya108; therefore, a large portion of chromosome 1 was inverted relative to the dnaA region and this may be the reason why the reviewer felt that our presentation is “not consistent”. We did not make changes on the fasta files used in data analysis is to prioritize the reproducibility of results. But we added arrows to indicate the position of SE-6945 such that the reader can easily follow the insertion locations in the inverted chromosome (new S1 Fig).

(Reviewer #2’s comments)

6. Line 112: Rewrite sentence to include that the segment is absent from the corresponding region of CAIM 1831.

(Response) We deleted CAIM 1831 from Fig 1 (relevant to previous line 112) due to major changes in the genome comparison section (new Fig 2 and lines 119-122). We simply used sequence of CAIM 1831 to show unoccupied target sites (attB) in new Fig 2B. 

(Reviewer #2’s comments)

7. Line 114: Indicative values should be provided for "very low gene product identity".

(Response) We added the identity values in the revised manuscript as stated above.

(Reviewer #2’s comments)

8. Line 115-117: It is not clear to me what data specifically has "confirmed" that the repeated region is an active integrative element. The evidence for this assertion needs to be stated succinctly. Moreover, if this is the case I find it perplexing that it has been given a Tn number since it is clearly not a transposon. I appreciate that historically some integrative elements were assigned Tn numbers, but the important distinction between IE and Tn (site-specific recombinase verses comparatively random transposase) has been understood for some time. The practice of assigning Tn numbers to IEs promotes confusion and is hence unhelpful; it should not be continuing in 2021 - alternative naming conventions are being used for IEs. A complicating issue here is that at this time it is unclear what type of IE this one represents. It does not appear to be an ICE, but it could be an IME (integrase-like proteins have been found to serve as a relaxase in some conjugative plasmids) or it could be a form of a CIME. In any case, a name that is not misleading would be preferable.

(Response) In the revised manuscript, we explicitly stated that “This 7.1-kb region was confirmed to be active in generating its circular copy of top strand in E. coli and V. alfacsenis and inserting it into new target location in E. coli in this study“ (thus satisfying the definition of transposon in Roberts et al., 2008) at line 127-129. While the term transposon/transposition has been clearly defined in several reviews and papers (ICE is classified as one form of transposons in Curcio and Derbyshire 2003), the term IE has not been explicitly defined historically. The definition of ICE/IME/CIME, based on their transfer activity, is also problematic in that the scientists cannot tell whether the new IE is CIME or not, if conjugation genes are not embedded in the IE.　In this revision, we conducted additional database surveys and found that Tn6945/SE-6945 and Tn6283/SE-6283 -like IEs clearly have 4 conserved genes (new Fig 2 and new Fig 6), and form a family of new mobile elements. They do not look like degraded ICE. In addition, the first step of their location change is presumably “copy-out” as elaborated in the revised manuscript, although their integration step is still vague (it can be copy-in (replication-dependent) or paste-in (replication-independent) in the classification by Curcio and Derbyshire 2003). Therefore, we coined the term SE for these IEs in the revised manuscript.　We think that frustration for the usage of transposon/transposition/integration/translocation for diverse readers will be solved if we explicitly state the rules of term usage in the introduction as stated above. 

(Reviewer #2’s comments)

9. Line 121-124: This section describes insertion sites in E. coli, shown in Fig 1D, but the insertion sites from the Vibrio strains (shown immediately above them in the same figure) are ignored in the text. If they are not informative they should be removed; but surely they are. Is the information consistent between the organisms? I would have thought a consensus motif for attB target sites might be worth deducing/showing?

(Response) We added new information about insertion locations of SE-6945 like elements (new Fig 2: line 200-225) and SE-6283 like elements (new Fig 6, line 398-423). They indeed show target site preference.

(Reviewer #2’s comments)

10. Line 125: Indicative values should be provided for "very similar".

(Response) We modified the sentence as “As pSEA2 showed > 99.9 % nucleotide identity on the aligned region to pSEA1”: line 155.

(Reviewer #2’s comments)

11. Line 128: "previously" should be "previous".

(Response) We deleted this phrase in the revised manuscript: line 158.

(Reviewer #2’s comments)

12. Line 130-131: Without first meaningfully describing the similarity relationship between pSEA1 and pSEA2, it is difficult to know whether deducing a precursor based on the presence/absence of a site-specific integrative element is appropriate.

(Response) We removed the relevant sentence in the revised manuscript.

(Reviewer #2’s comments)

13. Section starting line 133: This section starts by outlining a plausible scenario for plasmid integration via a two-step process involving homologous recombination. However, analogous mobile element flanked plasmid co-integrate arrangements can also arise in a single step in the absence of homologous recombination. The possibility of this (and possibly other) scenarios needs to be noted here also so the rationale for the subsequent experiments is clear to a reader. 

(Response) We think that a Tn3-like movement (Shapiro’s intermediate formation) is unlikely for SE for a couple of reasons. The formation of Shapiro’s intermediate requires one nicking at the top strand and one nicking at the bottom strand simultaneously. We actually show that strand exchange occurs only at the top stand (Fig 4 DE), thus, cointegrate formation through Shapiro’s intermediate is very difficult to happen. If cointegration formation happens, insertion of Tc Ap should occur as frequently as Ap insertion, which was also not the case for our observation.　To eliminate the frustration of readers, we added the following statement in line 256-261: “The Tn3-like cointegrate formation [15] might happen if SE-6945 could simultaneously introduce nick at one site on the top strand at attL (or attR) and one site on the bottom strand at attR (or attL), and DNA-protein crosslink (DPC) repair [38] could clean the linked tyrosine recombinase to produce a 3′-OH priming site. If SE-mediated cointegrate formation occurs, then Tc Ap-resistant transconjugant should emerge at a comparable frequency to Ap-resistant transconjugant, which was not the case in our observation.” 

(Reviewer #2’s comments)

14. The subsequent two paragraphs, and Fig 2, which describe the data, are very difficult to understand in terms of what was done and the data resulting. It all needs to be described much more clearly. Plasmid integration is said to be based on transfer of both ampicillin and tetracycline, but then only tetracycline-resistant transconjugants are described, and only Tet + Ap is shown in Fig 2. Elsewhere in the paper, transfer frequency is provided "per donor" as is established practice. Frequencies indicated in Fig 2 should do likewise, rather than per donor x recipient. Moreover, I think it would be preferable for this data to be provided in a table rather than a graph, so it is clear what data was obtained for each mating. For example, the single dot for Tet+Ap transfer in the delta-recA strain does not really agree with what is stated in the text.

(Response) We changed the unit of Y axis of transfer frequency data (new Fig 3) to T/R. The reasons why we used log T/(D*R) is already mentioned above. We modified the description of “tetracycline resistance” to “Tc Ap resistance”. Data point for no transconjugant is indicated by open circle in the new Fig 3.

(Reviewer #2’s comments)

15. What is the explanation for the apparently reduced Ap transfer in the delta-recA strain? Could this be responsible for the failure to detect any Ap + Tet transfer (i.e., insufficient sensitivity)? If so, the data doesn't really provide clear support for the proposed two-step scenario.

 (Response) We appreciate the reviewer’s comment. To address this, we added the following statement “Unexpectedly transfer frequency of Ap resistance was also reduced to the 1/17 level of JW0452rif in JW0452ΔrecArif (compare the second and the fourth groups from the left in Fig 3). Thus, the direct effect of recA knockout on pSEA2 integration could not be evaluated due to the reduced detection sensitivity of gene transfer. However, when LN52rif (which already carries one copy of SE-6945 and wild type recA) was used as the recipient, Tc Ap-resistant transconjugants were obtained at a 56-fold higher frequency (in mean) than that observed when using JW0452rif as the recipient (compare the first and fifth groups in Fig 3). This indicates that the presence of the 7.1 kb long homology stretches in both target DNA and pSEA2 DNA promotes pSEA2 integration in the recipient” in lines 247-256.

(Reviewer #2’s comments)

16. Section starting line 168 (to 191) and manuscript title: Copy-out-paste-in has been employed to denote a mechanism of transposition. The mechanism employed by this type of integrative element may indeed share some commonalities (which are certainly worth noting), but the enzymes involved and the biochemistry among other things are going to be different (e.g., the observed strand bias, as noted in the Discussion). I therefore feel that simply using the same Copy-out-paste-in term to describe this integration mechanism will again lead to confusion, leaving some readers with the impression that transposition and integration are the same when they are clearly not. Additionally, mechanisms used by other integrative elements are not explained anywhere in the manuscript. Are they different? Does the data here preclude them? 

(Response) All IEs discovered to date follow excision-(transfer)-integration route. We clearly stated this in the introduced section in the revised manuscript as “ICEs/IMEs so far studied in the laboratory seem to change their location via the ‘excision-integration’ mode only (line 69-71). 

(Reviewer #2’s comments)

17. The failure to detect an empty attB site by PCR of itself does not represent compelling evidence that a copy of the element is retained in the donor replicon. Maybe the empty site is not re-joined and is hence lost from the population. 

(Response) Actually, “not re-joined”, leaving one single strand bridge (figure-eight form), is the key feature of the first step of the original copy-out paste-in transposition of IS (Chandler 2015). The reviewer might have meant to say “Not re-joined” after two rounds of strand exchange. We think this is unlikely, because two rounds of strand exchange at the bottom and top strands generate the so-called ‘couple sequence’ (heteroduplex) observed in the circular form of Tn916, and the sequencing of the spacer sequence between C and C’ in the cloned attS should detect two sequences, but we detected only one spacer sequence in attS in strain LN95. In the revised manuscript, we used more words to explain our interpretation and chose to use Path 1 (new Fig 4) in “copy-out route” instead of “copy-out paste-in transposition” since the integration step of SE is still vague. The updated Fig 4A, Fig 4 legend, and new descriptions added in lines 280-295 and lines 342-358 would help readers grasp all possible routes of transposition and understand our interpretations. 

(Reviewer #2’s comments)

18. Was a positive control performed to show that an attB site could be detected for a different element that is known to generate them? 

(Response) PCR detection of empty attB during IE excisions has been reported for representative ICE/IMEs. Those literature are cited in the introduction with the statement in lines 69-71 (reference 22-25) in the revised manuscript. We believe that it is not necessary for us to conduct the experimental analyses using those elements.

(Reviewer #2’s comments)

Indeed, could the attB PCR detect the empty site of CAIM 1831? Without additional evidence, it is premature to conclude that a copy-out-like mechanism is used.

(Response) We disagree with this reviewer’s comment. Since occupied attB (7.2 kb) was amplified using the same primers as used for empty attB detection (new Fig 4 B), primers and PCR conditions are optimal. It is not necessary to test the genome of CAIM 1831. In our previous study (Nonaka et al 2018) on SE-6283, empty attB was detected, though at a very low frequency. The ratio of attS to empty attB was about 100:1 based on qPCR for SE-6283 in E. coli. 

(Reviewer #2’s comments)

19. Line 215: Why is Tn6283 mentioned here? It doesn't seem to encode resistance so how is it relevant to these studies?

(Response) We here mentioned SE-6283 in 04Ya108 as another example of the occurrence of more than one SE in the genome. The occurrence of more than one copy in a genome may also be a feature of SE. Since there are only two examples of SE to date, SE-6283 is relevant to this section. Line 370-372.

(Reviewer #2’s comments)

20. Line 228: Although the results are mostly consistent with a correlation between Tn6945 copy number an Ap MIC, it is an overstatement to say they "clearly" show copy number affects resistance level; there is only a single strain with three copies, some strains with one or two copies exhibit low resistance, while another with one copy exhibits as much resistance as the most resistant isolates with two copies. This suggests that there are other factors at play so the data should be interpreted conservatively.

 (Response) We appreciate the reviewer’s comment. When polishing the descriptions about the Southern hybridization results and new Table 1 (strain table), we noticed that the insertion location of SE also affects the MIC. Hence the paragraph title was changed as “The insertion copy number and location of SE-6945 affects beta-lactam resistance levels of the host cell” in the revised manuscript” in lines 368-369. We also added the following descriptions in lines 387-390: “MIC of individual strain is shown in Table S1. When focused on a class of MIC 500, there was a trend that strains with SE-6945 insertion into insJ show a higher MIC than strains with insertion into yjjNt. Thus, the location of SEs can also affect phenotype variation, possibly due to the differences in the expression levels of their cargo genes.”

(Reviewer #2’s comments)

21. Fig 4 legend: What do the different colours denote?

(Response) The colors of the dots denote the copy number of SE. The phrase of “four groups shown in distinct colors” was added to the legend (new Fig 5. Line 394).

(Reviewer #2’s comments)

22. Discussion line 256: The conviction in the two step mechanism is not justified by data presented. Other alternatives, and known mechanisms of other integrative elements should be elaborated.

(Response) As we stated in response to reviewer 1’s comment, Tn3-like movement (Shapiro’s intermediate formation) is unlikely for SE, since the formation of Shapiro’s intermediate requires one nicking at the top strand and one nicking at the bottom strand simultaneously, as well as protein-DNA link repair for the SE to generate priming site. We actually found that nicking occurs only at the top stand for SE (Fig 4 DE). Furthermore, the presence of SE in the recipient still increases the pSEA2 integration frequency by 10-fold. For these reasons, homologous recombination is still the primal route of the pSEA2 integration into the chromosome. We rephrased the sentence to be more conservative as follows “The current most likely mechanism of this phenomenon is the transposition of SE followed by homologous recombination” at line 446-447 in the revised manuscript.

(Reviewer #2’s comments)

23. Discussion line 275: The possibility that this element might be mobilisable (i.e., an IME) should be canvassed, particularly given the multiple genes of unknown function. However, all that might be needed is an oriT site. Has a candidate oriT sequence been identified in pSEA1/pSEA2 (likely near their relaxase gene), and is there a similar sequence in Tn6945?

(Response) The difficulty with pSEA1/pSEA2 is that they belong to MobH group, which is the least characterized plasmid/ICE group. The operonic structure of tra operon is not conserved among MobH group members.Inverted repeat near oriT of Neisseria gonorrhoeae genomic island (GGI) encoding MobH family relaxase TraI, has been determined to be 5’-CAAAGGCCTGCATTTTTATGCAGGCCTTTG-3’ (Heilers et al 2019 NAR). oriT of ICE-SXT contain Inverted repeat motif: 5’-ccaaaagccaaacggatagtggttttggcttttgg-3’ near oriT (Ceccarelli et al JB 2008), but it is not located immediately upstream of TraI relaxsase. We could not find a similar motif on pSEA1/pSEA2 and SEs based on similarity search. If CDS2 or CDS4 product was related to relaxase (HUH endonuclease), motif search in Pfam and InterproScan should show hits. But they did not. For these reasons, positive comments on the presence of oriT on SE would become a speculation based on speculation.We understand that readers might show interest in whether SE possesses oriT or not. We added the following statement: “Another dissimilarity between SE and ICE/IME is that the former does not encode HUH endonuclease. There is still a possibility that SEs carry oriT recognized by a family of relaxases and are mobilized like CIME, though we could not identify oriT-like sequences in SE-6945 and SE-6283 by local similarity search alone” at line 495-497.

(Reviewer #2’s comments)

24. Line 573: The legend shown is presumably for S3 rather than S1. In any case, I doubt enough information is provided for a reader to understand the figure.

(Response) We corrected the legend names (current S1 Fig and S5 fig). We added more information for the Southern hybridization results as described above.

(Reviewer #2’s comments)

25. Line 581 and S2 legend: "‥.arch‥." should be "‥.arc‥.". I could not find the significance of the "contig 1" described anywhere so it should be removed (from here and the figure). What is the meaning of the gene colours used in B?

(Response) We appreciate the reviewer’s comments. We added the sentence “The 14.2 kb mphA-cat gene region flanked by repeats in pSEA2 (S2 Fig A) was detected as a separate circular contig in the TJ249 genome assembly” to the main text (Line 165-167) to describe the results clearly. We also add the following statement in the legend ”Four SE core genes are indicated by four distinct colors. Red pentagons are the �-lactamase gene”. Line 888-889 in the revised manuscript.

(Reviewer #2’s comments)

26. Fig S2A: Are putative conjugation or replication initiation genes evident in the pSEA2 sequence? If so, it would be informative if these additional genes at least could be indicated on the figure.

(Response) We added information on TraI relaxase/TraD coupling protein locations to the figure panel. Rep and oriV were difficult to predict (new S2 Fig). 

(Reviewer #2’s comments)

27. Fig S2B: pSEA1 should be included in the figure label. The legend provided is inadequate - what do the gene colours mean? The figure shows locations of the elements, not "insertion positions", and for both plasmids, not just pSEA1.

(Response) We added legend for intA, intB, CDS2, CDS4, then corrected panel B title to be “Location of SEs in pSEA1 and pSEA2” (new S2 Fig).

---

## [Decision Letter · Decision Letter 1]

1 Mar 2022

PONE-D-21-18582R1Atypical integrative element with strand-biased circularization activity assists interspecies antimicrobial resistance gene transfer from Vibrio alfacsensis

PLOS ONE

Dear Dr. Nonaka,

Thank you for submitting your manuscript to PLOS ONE. After careful consideration, we feel that it has merit but does not fully meet PLOS ONE’s publication criteria as it currently stands. Therefore, we invite you to submit a revised version of the manuscript that addresses the points raised during the review process.

I would like to sincerely apologise for the delay you have incurred with your submission. Your manuscript has been reviewed by the two previous reviewers; their comments are available below. The reviewers have indicated that although the revised version has improved respect the original version, there are still significant concerns that need to be addressed.

Please revise the manuscript to address all the reviewer's comments in a point-by-point response in order to ensure it is meeting the journal's publication criteria. Please note that the revised manuscript will need to undergo further review, we thus cannot at this point anticipate the outcome of the evaluation process.

We look forward to receiving your revised manuscript.

Kind regards,

Miquel Vall-llosera Camps

Senior Editor

PLOS ONE

Reviewers' comments:

Reviewer's Responses to Questions

**Comments to the Author**

1. If the authors have adequately addressed your comments raised in a previous round of review and you feel that this manuscript is now acceptable for publication, you may indicate that here to bypass the “Comments to the Author” section, enter your conflict of interest statement in the “Confidential to Editor” section, and submit your "Accept" recommendation.

Reviewer #1: (No Response)

Reviewer #2: (No Response)

2. Is the manuscript technically sound, and do the data support the conclusions?

Reviewer #1: Yes

Reviewer #2: Yes

3. Has the statistical analysis been performed appropriately and rigorously? 

Reviewer #1: I Don't Know

Reviewer #2: N/A

4. Have the authors made all data underlying the findings in their manuscript fully available?

Reviewer #1: Yes

Reviewer #2: Yes

5. Is the manuscript presented in an intelligible fashion and written in standard English?

Reviewer #1: No

Reviewer #2: Yes

6. Review Comments to the Author

Reviewer #1: This revised manuscript includes interesting and important information about a new type of integrative element, and the scientific aspects have been improved by adding/clarifying experimental details. However, further revision is still needed to address some scientific points and better present the data and conclusions, as the manuscript is still hard to follow - information needs to be presented in a more logical order - and the text is still wordy and repetitive and could be condensed in places. I have tried to give some more detailed suggestions of how this could be done in the comments below.

1) Nomenclature for the new integrative elements

(Response) In the revised manuscript, we have explicitly stated that the unclassifiable integrative elements, identified in this study, are a new class of mobile elements. Therefore, we have used an “SE-“ prefix (strand-biased circularizing integrative element) to refer to these IEs in the revised manuscript. We believe that using numbers to identify these DNA elements would make it easy to handle (refer to) them in existing and future mobile DNA databases. Therefore, we have used numbers to refer to the SEs in this manuscript.

NEW COMMENT: “SE” is not a particularly descriptive name. Including at least “I” for “integrative” should be be considered (i.e., “SIE”?). Also, it might be better to introduce this term only after describing identification of the strand biased mechanism. The Results sections could be reorganised to first describe the experiments that led to this conclusion (i.e., move lines 278-367 to after lines 168 and maybe combine lines 169-79 with it, ending with lines 129-131. Also consider the best order for Results sections on lines 181-276 and369-415- e.g., combining the sections on target preference for SE-6945 and SE-6383, or at least placing them together?

In the Abstract, the name SE-6945 is used on line 28, before the term SE is explained on lines 34-5. The explanation should be placed earlier, first stating something like “This 7.kb element has four coding sequences, which are related to those of an element previously referred to as Tn6283, two of which… PCR and sequencing revealed these elements generate a copy of only one strand…, so they are collectively termed SE (…) here.” Also, the name SE-6283 is used on lines 80, before the term SE is explained on lines 93-5 – again suggest “called Tn6283”, but delete “(renamed…)” here and use Tn6283 for the rest of the paragraph, and “a related element” on line 91, then ex[plain the term.

Another part of the response states “We prefer to use 4-digit numbers to keep this compatible with the existing Tn database, and in part considering the ease of handling them in future databases” but on line 205 the names “SE-VhaWXL538” is used , from the strain ID. Is the idea that this would be numbered at some point and if so, would the Tn registry be responsible for assigning the numbers? (also lines 502-3).

2) Introduction, lines 96-107, Use of transposition, integration insertion etc, L

The authors have explained why they prefer to use transposition as a more general term but I think that it would be better to incorporate these definitions with lines 53-72 and end the Introduction with the statement about identification of a new integrative element. Lines 104-7 (translocation) can simply be removed, as they are not relevant. I also think the use of the terms “integration” and “insertion” might be better reversed, to follow the traditional definition of integration, if SE are being called integrative elements. To me, SE-6945 integrates in the traditional sense, but the plasmid “insertion” is more complex, as it requires an additional recombination step.

3) Results, lines 110-180

NEW COMMENT: The title of this section could be reconsidered, as it also discusses “Tn6283” and other points and the section needs to be reorganized and condensed. Results of sequencing 04Ya240 could be moved up and if the sequencing of TJ249 helped identify SE-6945 then this information should be before the descriptions of the proteins. The proposed gene product names are used on lines 127-8, but not explained until lines 132-4 – they should be explained first and then used.

PREVIOUS COMMENT: 5. Lines 115-116 – the evidence for this is from the conjugation experiments?

(Response) We used the term “active” based on the evidence for (i) the circular form generation in the absence of pSEA2 in V. alfacsensis (new Fig 4, previous Fig 3) and plasmid-free E. coli (new S6 Fig) and (ii) new junction generation (attL, attR) in E. coli during conjugation experiments. We rephrased this sentence to explicitly state this fact as “This 7.1-kb region was confirmed to be active in generating its circular copy of top strand in E. coli and V. alfacsenis and inserting it into new target locations in E. coli in this study” in lines 129-131.

NEW COMMENT: As referred to above, I think that this information is in the wrong place. The point about the element being active and the top strand being involved should be after the description of the experiments that identified this (“to be active in generating its circular copy of top strand” also needs rewording).

Lines 164-7 are very unclear. The information about the mph(A) cat region is confusing. What kind of repeat? A repeated mobile element? Direct repeats/TSD? I couldn’t access the Figshare directly from the link in Ref 72/Table S2 (please check this) but I did manage to find the sequences. Contig names seem to be the other way round, i.e. contig2 is 14.2 kb. It looks like pSEA2 was transferred to E. coli and inserted, but a composite-transposon like structure was lost as a circular molecule? This all needs to be written so that it is much easier to follow and so that it does not get in the way of the main point about the two insertions in the chromosome.

Line 122 – the gene name would be blaCARB-19, with bla in italics and CARB-19 as subscript, but the protein is not the same as CARB-19 in the NCBI reference gene catalog. As NCBI now assigns bla protein names/numbers it would be better to use their nomenclature, obtaining a name/number from NCBI if possible.

PREVIOUS COMMENT 7. Lines 122-4 – “Tn6945” inserts itself, rather than its IR? It’s not clear whether it is being proposed that “Tn6945” has IR, like a typical e.g., Tn3 family transposon (and why are the repeats called C and C’?) which are internal to such Tn, or it has an att site that recombines with target att site to create hybrid sites, as for integrative elements. att site are mentioned in the Fig. 1 legend but are not discussed until later in the text, which is also a problem.

(Response). We understand the reviewer’s concern. We changed the phrase to “SE-6945 inserts itself ending with 5’-GTA-3 (Fig 1 A) along with an additional 6 bp next to the 5’ end of C at attL on the donor molecule into the target site” (line 177-178). C and C’ are from core site of site-specific recombination mediated by tyrosine recombinase (i.e. Lambda Int). The problem with this new integrative element is that we cannot deduce how much longer does the cis-element involved in recombination extend from motif C/C’. Therefore, we simply defined the term attL and attR as “The border regions between SE-6945 and its target locations” and then stated “The border regions between SE-6945 and its target locations were termed attL (intA-proximal side) and attR (intA-distal side) (Fig 2 A)” in the revised manuscript (line 169-170). We also elaborated to explain the feature around nicking sites at attL and attR at lines 170-176 and lines 210-216.

NEW COMMENT on lines 169-179 - see comments above on moving this section. Line 141 – please explain in the text why the names C and C’ are used. Also please consider whether just referring to border regions, rather than att, until the ends of the element have been properly defined would be better.

4) Other scientific points

Line 61 – it is the attC site only that is folded

Line 164 etc -is insJ associated with a known/named IS? If so, it would be better to use the IS name.

Line 239 – it would be helpful to note here that Ap resistance is on SE-6945, Tc resistance on the plasmid.

Line 255 – the “homology stretches” are actually identical?

Lines 287, 489 – do these descriptions apply to all IS? If not, “some IS” should be used.

Lines 288-9, 469– please reconsider descriptions relating to integron (upper case I not needed) gene cassettes and “empty donor sites”. The attC site belonging to a particular cassette moves with that cassette. Excision may create an empty attI if it was the only cassette, otherwise the downstream cassette moves up the array. Also, Ref. 40 on line 469 dates from before the single stranded mechanism was identified.

5) Table 1

This is useful but can be simplified and condensed. The Vibrio isolates need to be listed before the transconjugants and would also probably be better before the E. coli, with LN95 under 04Ya249. A heading “transconjugants” could be added, grouping these to reduce the size of the table, i.e., put LN# numbers for all isolates with identical characteristics on a single row (or even just list the number of transconjugants of each type, if LN numbers are not referred to elsewhere in the manuscript). This would allow footnote c to be simplified e.g., note the selection used in the table itself and the text already states in several places that selections were done at 42 deg. Footnote b is hard to understand. Please also align text in the first column on the left and at the top of each row, to make it easier to read. W0452deltarecA should be JW0452deltarecA? Are all resistance genes are on the plasmid in each case? This should be made clear.

Lines 531-3, 565-7 could also be incorporated into Table 1.

6) Figures

Fig. 1A– the arrowheads might be better left off, as the reasons for proposing these nicking sites have not been introduced at this point in the text and showing them on a sequence is clearer /more informative. I would also suggest placing part C first to better follow the order in the text.

Fig. 2 – see suggestions on reorganisation of text. As the two strains at the bottom of Part A don’t appear to have been used in Part B maybe these diagrams could be moved to Supplementary?

Legend - line 220 “coordinates”. Line 221 “sequences were” “unpredictable” is the wrong word here – suggest “the bases in the putative crossover region could not be defined and are shown as X”. “one site on the left side on the top strand” is unclear and needs rewording.

Fig. 3 – line 270 – “no transconjugants were”.

Fig.4 - Legend – line 310 “Design of” can be deleted. Primer names are very long – can PCR reaction numbers be put in primer table? Line 215 – suggest “as shown as separate lines, with SE in orange”. “IS3 route”, “ICE/IME route” “gene cassette route” shown on the figure are not referred to in the legend. Reduce overlap between description of parts A and B. Line 329 – “as arrowheads”. Lines 372-8 – legends should generally not contain methods.

Fig. 5 – mention Ap in the title and add “Ap” before MIC on vertical axis label.

Fig. 6 - line 420 – “and carry”

7) Supplementary

Fig. S1 legend, line 870 “and TJ249”

Fig. S2 legend, line 885 – “genes were”, line 887 “PacBio assembly”

Fig. S3 legend – reword “part of the mobile DNA unit of the SE element” (also Fig. S4 lines 902-3, Fig. S7 line 930).

Fig. S6 legend – suggest “(B) Relative copy number of attS in V. alfansensis LN95 (no pSEA2)” Relative to what?

8) Minor/formatting/English

The Methods section could be simplified/condensed by restricting information to what was done, rather than too much description of why, especially if already explained in Results. Some minor rewording for accuracy/clarity is also needed to make the manuscript easier to read (some suggestions are included below).

PREVIOUS COMMENT: 14. Line 240 – V. cholerae is not always from sea food?

(Response) V. cholerae is indeed not always from sea foods as described in the cited reviews. These descriptions are correct. Thus we did not make changes to the revised manuscript.

NEW COMMENT: now line 430. I don’t understand this response. If these Vibrios are not always from sea food then the wording should be changed.

PREVIOUS COMMENT: 36. Line 184 – meaning actually “chromosomal copy of Tn6945 in V. alfacsensis is functional”?

(Response) Because only one SE-6945 copy is present in LN95, detection of circular SE-6945 in LN95 indicates that the cis-elements (attL/attR) and strand exchange enzymes encoded in the chromosomal SE-6945 is indeed functional, while those of SE-6945 on pSEA2 is obviously functional as its transposition was observed in E. coli. We explicitly stated that “This suggests that the cis-elements (attL/attR) and strand exchange enzymes encoded by the chromosomal copy of SE-6945 are also functional in V. alfacsensis” in lines 304-306 in the revised manuscript.

NEW COMMENT: This was a comment on wording only. I think that what I previously suggested is clearer

PREVIOUS COMMENT: 38. Line 199 – “absence of pSEA-2 from”

NEW COMMENT: Sorry, this should have referred to line 299, now line 515, and is simply a wording issue. If pSEA2 is absent then it is not “in” LN95, so “from” is clearer/more correct.

OTHER NEW COMMENTS:

Lines 37-8 – could delete “in addition to conjugative plasmids”

Line 61 - “of an integron cassette” or “of integron gene cassettes” and “the phage CTX“.

Line 33 – “The copy number and location”

Line 64 – “a Y-transposase”, “an S transposase.

Line 67 – please give full names for these elements in addition to abbreviations.

Line 92 – “the E. coli chromosome”

Line 99 – “as” before transposition should be omitted.

Line 114 – “and transconjugants selected on erythromycin”

Line 120 – please give the accession no. for the CAIM 1831 sequence here or in Fig. S1.

Line 121 – “plasmid” is not needed before pSEA2.

Line 122 – “a 7.1 kb repeat region”, not “the 7.1 kb repeat region”

Line 133 – suggest “predicted” rather than “estimated”.

Lines 159-60 – maybe “also carries two copies of SE-6945, in the same locations on the chromosome and plasmid as in 04Ya249”?

Line 186 – “transconjugants were screened on Tc alone, …”

Line 195 - “other transconjugants”

Line 198 – delete “the”

Line 201 – “searched for”

Line 214 - “integrons”.

Lines 234-5 – “altogether in a recA-null”

Lines 243 – this needs rewording - transconjugants are not transferred

Line 246 – “in any of the four”

Line 248 – suggest “the transfer frequency of Ap resistance to JW0452deltarecArif was about 17 fold lower than transfer to JW0452rif”. Was this for the means?

Line 253 - “in mean” needs rewording.

Line 256 – “integration into the recipient chromosome”

Line 281 – fix “E. col”

Lines 282-3 – could this be expressed the other way round? i.e., tyrosine recombinase typically produce staggered cuts at att sites?

Lines 288, 291, 360 etc – “single stranded”, “double stranded”

Line 297 – “anneals inside the integrative element and the other outside it”

Line 301 – “primer sets”.

Line 308 – “long PCR conditions were used”

Line 372 – “their genome”

Lines 379-80, 387- need rewording - only Ap MICs were obtained.

Line 388 – “MICs against individual strains are”. Not clear what “when focused on a class of MIC 500” means.

Line 405 – “screened for”, or “identified”

Line 406 – “and related SEs”

Line 431 – “linking” not really the right word here, “aiding movement between”?

Line 462 - “transposon mechanisms”?

Line 469 – needs rewording

Line 473 - “unjoined”?

Line 475 – “a process equivalent to”

Line 496 – “relaxases”, “and oriT in SE-6945 or SE-6282”

Lines 567-8 would be better incorporated into lines 563-5 i.e. add “and incubated at 42deg C to supress growth of Vibrio strains” to the end of the sentence. See also comments above on incorporating information into Table 1.

References – species names should be in italics.

Reviewer #2: The revision of the manuscript has clarified many aspects of the original version. I don't think it will be productive to argue the utility of general verses specific use of the words transposon/transposition. More importantly, I feel that the use of "SE" in place of "Tn" is a satisfactory solution that clearly implies distinctions between these particular integrative elements from others, and Tn-named elements.

Further suggestions:

Line 25: "multidrug resistance conjugative plasmid" is probably better written as "conjugative multidrug resistance plasmid".

Line 31: "of its one specific" can just be "of one specific".

Line 33: "Copy numbers and location" should be "The copy number and location".

Line 48: Sentence should be: Conjugative plasmids and integrative and conjugative elements (ICE) [6] are DNA units that can move from one cell to another through conjugation machinery that they encode.

Line 100: Just a suggestion. Replicon fusion between plasmids and/or chromosome could be termed "cointegration", leaving "integration" to be used for site-specific recombination, since "insertion" is commonly used for events not involving site-specific recombination.

Line 103: I appreciate the authors explaining why they haven't used the term "translocation". However, since it is not used the explanation isn't needed in the paper - it might appear perplexing to a reader.

Line 114: "and the transconjugant was first selected" should be "and transconjugants were selected".

Line 166 and Fig S2: The significance of contig 1 is not clear to me. How does it indicate movement of SE-6945?

Line 186: "transconjugant was" should be "transconjugants were".

Line 190: " generated intA-containing fragments originating from" should be "possessed intA-containing fragments located at".

Line 197: It is more correct to say that pSEA2 integration relies on SE-6945 transposition, since generally transposition of SE-6945 will NOT result in plasmid integration (it will do so only rarely).

Line 212 and elsewhere: "Sequence logos" should be "Sequence logo".

Line 257: "The Tn3-like" should be "Tn3-like".

Line 347 and 353: (Fig 4E) should be (Fig 4D).

Line 371: "in genomes" should be "in their genomes".

Line 372: Why/how would/could SE-6283 be expected to influence beta lactam resistance? Seems like a complete red herring to me.

Line 388-391 paragraph: Could just be: The MIC of individual strains is shown in Table S1. There was a trend for strains with SE-6945 insertion into insJ to show a higher MIC than strains with insertion into yjjNt. Thus, the location of SEs can result in phenotype variation, possibly due to the differences in the expression levels of their cargo genes.

Line 414: As written, the sentence implies the opposite of what is meant. I suggest change to "and, like attB of SE-6945 did not possess an inverted repeat structure."

Line 418: Isn't it five genomic locations shown in Fig 6A?

Line 462: "movements" instead of "moves"

Line 471: "possibly the CDS2" rather than "i.e., CDS2".

Line 473: " transposases" instead of "transposase".

Line 497: It might be worth mentioning here that a tyrosine recombinase-like protein has been shown to function as a conjugative relaxase, particularly since these elements contain two such proteins. See reference https://doi.org/10.1111/mmi.13270.

7. PLOS authors have the option to publish the peer review history of their article (what does this mean?). If published, this will include your full peer review and any attached files.

Reviewer #1: No

Reviewer #2: No

---

## [Author Response · Author response to Decision Letter 1]

5 May 2022

Responses to the Reviewer’s comment

PONE-D-21-18582

Reviewer #1: This revised manuscript includes interesting and important information about a new type of integrative element, and the scientific aspects have been improved by adding/clarifying experimental details. However, further revision is still needed to address some scientific points and better present the data and conclusions, as the manuscript is still hard to follow - information needs to be presented in a more logical order - and the text is still wordy and repetitive and could be condensed in places. I have tried to give some more detailed suggestions of how this could be done in the comments below.

(Reviewer #1’s Comments)

Nomenclature for the new integrative elements

(Response) In the revised manuscript, we have explicitly stated that the unclassifiable integrative elements, identified in this study, are a new class of mobile elements. Therefore, we have used an “SE-“ prefix (strand-biased circularizing integrative element) to refer to these IEs in the revised manuscript. We believe that using numbers to identify these DNA elements would make it easy to handle (refer to) them in existing and future mobile DNA databases. Therefore, we have used numbers to refer to the SEs in this manuscript.

1. NEW COMMENT: “SE” is not a particularly descriptive name. Including at least “I” for “integrative” should be be considered (i.e., “SIE”?). 

(Response) We thank the reviewer for their comment. However, SIE may be confused with eukaryotic “SINE” transposons. Thus, we did not use it. When priorizing distinction among mobile DNA families, SE is rather apt. Putative integrative element Crypton in fungi does not even include I or E. 

(Reviewer #1’s Comments)

2. Also, it might be better to introduce this term only after describing identification of the strand biased mechanism. 

(Response) We have revised the manuscript based on this suggestion. In the Abstract, the word "Tn6283" has been used before the SE term definition (line 36). Similar to the Abstract, the word "Tn6283" has been used before the SE term definition at line 95 in the Introduction. As SE is defined in the Introduction, we see no issues with using the term "SE" in the Results. 

(Reviewer #1’s Comments)

3. The Results sections could be reorganised to first describe the experiments that led to this conclusion (i.e., move lines 278-367 to after lines 168 and maybe combine lines 169-79 with it, ending with lines 129-131. 

(Response) I am afraid that the section order suggested by Reviwer 1 is not really the best to explain the findings in this study. We attempted to formulate the text following Reviewer 1’s suggestion, but it resulted in restricted word usage in the figures (we could not use the terms “SE”, “attL”, “attR”, “attS” in the figure showing the PCR results) and induced repetition of the words ‘7.1 kb element’ in the Results section. Those were not really helpful.

Strand-biased circularization mechanisms were identified in our previsous study on Tn6283, and this has been alluded to in the revised manuscript (line 93). We then introduced the term “SE“ in the Introduction (line 95). Whether the 7.1 kb element is SE or not is relatively easy to deduce based on the results of genome sequencing. Therefore, introducing "SE-6945" at a relatively early location in the Results with belief reasoning and inclusion of the statement “(see below)” help reduce repetition of "the 7.1 kb element" in Figures and text and is rather apt. 

(Reviewer #1’s Comments)

4. Also consider the best order for Results sections on lines 181-276 and 369-415- e.g., combining the sections on target preference for SE-6945 and SE-6383, or at least placing them together?

(Response) We agree with Reviwer 1 regarding this point. We combined the target site preference sections for SE-6945 and SE-6283 and made one new section "Target site preference of SE-6945 and SE-6283" (line 174). 

(Reviewer #1’s Comments)

5. In the Abstract, the name SE-6945 is used on line 28, before the term SE is explained on lines 34-5. The explanation should be placed earlier, first stating something like “This 7.kb element has four coding sequences, which are related to those of an element previously referred to as Tn6283, two of which… PCR and sequencing revealed these elements generate a copy of only one strand…, so they are collectively termed SE (…) here.”

(Response) We modified the phrases and changed the order of sentences in the Abstract as suggested (lines 28 to 34).

(Reviewer #1’s Comments)

6. Also, the name SE-6283 is used on lines 80, before the term SE is explained on lines 93-5 – again suggest “called Tn6283”, but delete “(renamed…)” here and use Tn6283 for the rest of the paragraph, and “a related element” on line 91, then explain the term.

(Response) “SE-6283” in Introduction in the previsous mansucript was replaced with Tn6283 before line 95. Introductory sentences for the SE term are now “Based on the findings in the previous study and this study, we have termed the currently unclassifiable integrative element, Tn6283, and a related element as a new class of mobile elements: strand-biased circularizing integrative element (SE). Tn6283 is hereafter referred to as SE-6283”.

 (Reviewer #1’s Comments)

7. Another part of the response states “We prefer to use 4-digit numbers to keep this compatible with the existing Tn database, and in part considering the ease of handling them in future databases” but on line 205 the names “SE-VhaWXL538” is used, from the strain ID. Is the idea that this would be numbered at some point and if so, would the Tn registry be responsible for assigning the numbers? (also lines 502-3).

(Response) Yes, Tn registry assinged "9645". Tn registry assigns four digits to all DNA segments that satisfy the criteria used to define transposon (evidence of transposition) when requested. Bioinformatically screened SEs do not exhibit "evidence of transposition". Thus, the Tn registry would not assign numbers to them, but we need to refer to those putative transposons/SEs somehow in the text. Strain ID is a convenient and reasonablely clear solution for handling putative SEs in manuscript. 

In the revised manucript we clearly mentioned the connection between name and the Tn registry at line 131: “Thus, the 7.1 kb element was labelled a transposon and assigned a four digits identifier, 6945, by the transposon registry “, and at line 506: “To refer to transposable individual SE members, we used digits that are compatible with the existing transposon database”.

(Reviewer #1’s Comments)

8. Introduction, lines 96-107, Use of transposition, integration insertion etc, L 

The authors have explained why they prefer to use transposition as a more general term but I think that it would be better to incorporate these definitions with lines 53-72 and end the Introduction with the statement about identification of a new integrative element. Lines 104-7 (translocation) can simply be removed, as they are not relevant. I also think the use of the terms “integration” and “insertion” might be better reversed, to follow the traditional definition of integration, if SE are being called integrative elements. To me, SE-6945 integrates in the traditional sense, but the plasmid “insertion” is more complex, as it requires an additional recombination step.

(Response) We removed the terms “translocation”, and “insertion” and then introduced “cointegration” to refer to plasmid insertion into chromosome (as suggested by reviewer 2). “Integration” was used to refer to site-specific recombination, as suggested. However, we feel incorporation of these general definitions to an earlier paragraph addressing MGE classification (previous lines 53-72) would distract readers. Hence, we have not moved these lines (line 90-98).

(Reviewer #1’s Comments)

9. Results, lines 110-180　NEW COMMENT: The title of this section could be reconsidered, as it also discusses “Tn6283” and other points and the section needs to be reorganized and condensed. Results of sequencing 04Ya240 could be moved up and if the sequencing of TJ249 helped identify SE-6945 then this information should be before the descriptions of the proteins. 

(response) The paragraph describing the sequencing of TJ249 has been moved up as suggested (line 115), and previous Fig 1 Panel C has also been changed to Panel A. 

(Reviewer #1’s Comments)

10. The proposed gene product names are used on lines 127-8, but not explained until lines 132-4 – they should be explained first and then used.

(Response) We removed the explanations regarding product names because gene/product names were given to Tn6283/SE-6283 in the previous work.

(Reviewer #1’s Previous comments)

PREVIOUS COMMENT: 5. Lines 115-116 – the evidence for this is from the conjugation experiments?

(Response) We used the term “active” based on the evidence for (i) the circular form generation in the absence of pSEA2 in V. alfacsensis (new Fig 4, previous Fig 3) and plasmid-free E. coli (new S6 Fig) and (ii) new junction generation (attL, attR) in E. coli during conjugation experiments. We rephrased this sentence to explicitly state this fact as “This 7.1-kb region was confirmed to be active in generating its circular copy of top strand in E. coli and V. alfacsenis and inserting it into new target locations in E. coli in this study” in lines 129-131.

11. NEW COMMENT: As referred to above, I think that this information is in the wrong place. The point about the element being active and the top strand being involved should be after the description of the experiments that identified this (“to be active in generating its circular copy of top strand” also needs rewording).

(Response) We disagree with this comment. By providing the information about SE-6945 in the first, we can avoid repetitively describing “the 7.1 kb element” in figures and text. Adding “(see below)” would suffice since the readers can validate it after reading the paragraph about circularization detection. 

Regarding rewording, we rephrased the sentence as follows: “As the 7.1-kb element was confirmed to generate a circular copy of one specific strand without leaving an empty site like SE-6283 in this study (see below)”. (line 132)

(Reviewer #1’s Comments)

12. Lines 164-7 are very unclear. The information about the mph(A) cat region is confusing. What kind of repeat? A repeated mobile element? Direct repeats/TSD? I couldn’t access the Figshare directly from the link in Ref 72/Table S2 (please check this) but I did manage to find the sequences. Contig names seem to be the other way round, i.e. contig2 is 14.2 kb. It looks like pSEA2 was transferred to E. coli and inserted, but a composite-transposon like structure was lost as a circular molecule? This all needs to be written so that it is much easier to follow and so that it does not get in the way of the main point about the two insertions in the chromosome.

(Response) The following sentence was deleted, as it was not relevant to the SEs (also mentioned by Reviewer 2).

 “The 14.2 kb mphA-cat gene region flanked by repeats in pSEA2 (contig 1 in S2 Fig A) was detected as a separate circular contig in the TJ249 genome assembly” (previous lines 165-167). 

(Reviewer #1’s Comments)

13. Line 122 – the gene name would be blaCARB-19, with bla in italics and CARB-19 as subscript, but the protein is not the same as CARB-19 in the NCBI reference gene catalog. As NCBI now assigns bla protein names/numbers it would be better to use their nomenclature, obtaining a name/number from NCBI if possible.

(Response) We communicated with the Pathogen-Detection group at NCBI, and they have assigned a new family name and a new allele name for this beta-lactamase gene. Now, blaGMA-1 is used in the manuscript. We have added the following sentence at line 120: “One �-lactamase gene (60 % product identity to blaVHH-1 in the CARD database) was located within a 7.1-kb repeat region found in both chromosome 1 and pSEA2 (S2 Fig B). This �-lactamase gene has been assigned a new family name, blaGMA (Gammaproteobacterial Mobile Class A �-lactamase), and allele name, blaGMA-1, in NCBI.”.

(Reviewer #1’s Comments)

PREVIOUS COMMENT 7. Lines 122-4 – “Tn6945” inserts itself, rather than its IR? It’s not clear whether it is being proposed that “Tn6945” has IR, like a typical e.g., Tn3 family transposon (and why are the repeats called C and C’?) which are internal to such Tn, or it has an att site that recombines with target att site to create hybrid sites, as for integrative elements. att site are mentioned in the Fig. 1 legend but are not discussed until later in the text, which is also a problem.

(Response). We understand the reviewer’s concern. We changed the phrase to “SE-6945 inserts itself ending with 5’-GTA-3 (Fig 1 A) along with an additional 6 bp next to the 5’ end of C at attL on the donor molecule into the target site” (line 177-178). C and C’ are from core site of site-specific recombination mediated by tyrosine recombinase (i.e. Lambda Int). The problem with this new integrative element is that we cannot deduce how much longer does the cis-element involved in recombination extend from motif C/C’. Therefore, we simply defined the term attL and attR as “The border regions between SE-6945 and its target locations” and then stated “The border regions between SE-6945 and its target locations were termed attL (intA-proximal side) and attR (intA-distal side) (Fig 2 A)” in the revised manuscript (line 169-170). We also elaborated to explain the feature around nicking sites at attL and attR at lines 170-176 and lines 210-216.

14. NEW COMMENT on lines 169-179 - see comments above on moving this section. Line 141 – please explain in the text why the names C and C’ are used. Also please consider whether just referring to border regions, rather than att, until the ends of the element have been properly defined would be better.

(Response) We have added a statement explaining why the terms C and C’ are used at line 152: “; the terms C and C’ were used because they were thought to be functionally equivalent to the core-type sites of phages [35].”. 

Regarding attL/R, we think that the use of “border region” instead of attL/R would rather confuse readers (particularly in the figure). Therefore we did not follow this suggestion. Most Phage/ICE -related studies use the term attL/R without having a clear definition.

(Reviewer #1’s Comments)

4) Other scientific points

15. Line 61 – it is the attC site only that is folded.

(Response) We have deleted “a folded single-strand DNA” from the sentence and simply stated that “DNA insertion involving DNA replication can also be seen in the integration of integron gene cassettes into attI, and the phage CTX into dif” (line 62).

(Reviewer #1’s Comments)

16. Line 164 etc -is insJ associated with a known/named IS? If so, it would be better to use the IS name. 

(Response) ‘insJ’ was replaced with ‘insJ of IS3’ (line 128). 

(Reviewer #1’s Comments)

17. Line 239 – it would be helpful to note here that Ap resistance is on SE-6945, Tc resistance on the plasmid.

(Response) We have added the following line to the manuscript: “Note that Tc resistance is on pSEA2, while Ap resistance is on SE-6945”. (line 263)

(Reviewer #1’s Comments)

18. Line 255 – the “homology stretches” are actually identical?

(Response) Yes. The term “homology stretches” were deleted to avoid confusion and was replaced with “identical sequence” (line 280).

(Reviewer #1’s Comments)

19. Lines 287, 489 – do these descriptions apply to all IS? If not, “some IS” should be used. 

(Response) We replaced “IS” with “some ISs” (line 313).

(Reviewer #1’s Comments)

20. Lines 288-9, 469– please reconsider descriptions relating to integron (upper case I not needed) gene cassettes and “empty donor sites”. The attC site belonging to a particular cassette moves with that cassette. Excision may create an empty attI if it was the only cassette, otherwise the downstream cassette moves up the array. Also, Ref. 40 on line 469 dates from before the single stranded mechanism was identified.

(Response) We modified the Discussion (not Results) section as follows: “In the replicative resolution model, strand exchange between two folded attC bottom strands and the following replication generates a gene cassette circle with attC and results in the removal of one gene cassette with attC from the gene cassette array on the donor molecule. This can cause formation of integron variants with reduced number of gene cassettes and even integrons with an empty attI, as observed in in vivo experiments [41].” (line464-469).

(Reviewer #1’s Comments)

21. 5) Table 1 This is useful but can be simplified and condensed. The Vibrio isolates need to be listed before the transconjugants and would also probably be better before the E. coli, with LN95 under 04Ya249. A heading “transconjugants” could be added, grouping these to reduce the size of the table, i.e., put LN# numbers for all isolates with identical characteristics on a single row (or even just list the number of transconjugants of each type, if LN numbers are not referred to elsewhere in the manuscript). 

(Response) We reorganized Table 1 and reduced the size as per the suggestion. Now Vibrio isolates are listed first, followed by LN95, E. coli, and transconjugants. LN strains with identical characteristics were placed in a single row. (revised Table 1) 

21.2 This would allow footnote c to be simplified e.g., note the selection used in the table itself and the text already states in several places that selections were done at 42 deg. 

(Response) Table 1 contains information regarding Cm and Erm which were not used for transconjugants selection. Therefore, we cannot mix footnote information with Table 1 contents. Therefore, we have added the following lines: “Selection was done at 42 °C” (line 532) and removed three “42 °C”. However, antibiotic selection information remains unchanged.

The following sentences in previous manuscript have been deleted from footnote c. “E. coli strain LN52, which carries a single copy of SE-6945, is a JW0452 transconjugant obtained by ampicillin selection at 42 °C. TJ249 is also a JW0452 transconjugant obtained by erythromycin selection at 42 °C.” 

(Reviewer #1’s Comments)

22. Footnote b is hard to understand. 

(Response) We modified footnote b as follows: “b. Cointegration of plasmid pSEA2 with chromosome, independent of SE-6945 transposition. Location of pSEA2 in the transconjugant genome could not be defined based on Southern hybridization results.” (line 527).

(Reviewer #1’s Comments)

23. Please also align text in the first column on the left and at the top of each row, to make it easier to read. 

(Response) The texts in the first column of Table 1 have been aligned to the left (revised Table 1).

(Reviewer #1’s Comments)

24. W0452deltarecA should be JW0452deltarecA? 

(Response) Yes. We have replaced “W0452deltarecA” with “JW0452deltarecA” (new Table 1).

(Reviewer #1’s Comments)

25. Are all resistance genes are on the plasmid in each case? This should be made clear.

(Response) We have added the following sentence: “It is unclear whether the resistance genes on pSEA2 are still in the original locations for other transconjugants besides TJ249.” (line 525). 

(Reviewer #1’s Comments)

26. Lines 531-3, 565-7 could also be incorporated into Table 1.

(Response) Lines 531-3 and lines 565-7 from the previous version of the manuscript have been deleted and we used more simple expression: “Ap for LN52” (line 533) and “Ery for TJ249” (line 534) in footnote c.

(Reviewer #1’s Comments)

27. 6) Figures Fig. 1A– the arrowheads might be better left off, as the reasons for proposing these nicking sites have not been introduced at this point in the text and showing them on a sequence is clearer /more informative. I would also suggest placing part C first to better follow the order in the text.

(Response) We have removed the arrowheads from previous Fig. 1 panel A (new panel B) and placed previous panel C fist in the new Fig.1, as suggested.

(Reviewer #1’s Comments)

28. Fig. 2 – see suggestions on reorganisation of text. As the two strains at the bottom of Part A don’t appear to have been used in Part B maybe these diagrams could be moved to Supplementary?

(Response) As mentioned in previous comments, reorganization would not enhance readers’ understanding. We chose to mention target site selection immediately after the sequencing results, as in the original manuscript. We think that those two diagrams are very useful to show that chromosomal target location of SE-6945 in genus Vibrio is limited. As attB in the chromosomes of strains WXL538 and 04Ya249 is occupied by an SE, we cannot define the ancestral attB sequences of those two strains. Hence, we used the unoccupied attB sequences of V. harveyi WXL538 and V. alfacsensis CAIM1831 as the ancestral attB sequences of strain WXL538 and 04Ya249 respectively in panel B. Therefore, SE target sites in the two strains of V. harveyi WXL538 and V. alfacsensis CAIM1831 still remain in main Fig 2 panel A.

(Reviewer #1’s Comments)

29. Legend - line 220 “coordinates”.

(Response) We have made the suggested modification (line 218)”.

(Reviewer #1’s Comments)

30. Line 221 “sequences were” “unpredictable” is the wrong word here – suggest “the bases in the putative crossover region could not be defined and are shown as X”. 

(Response) We have rephrased the sentence as follows: “the bases on pSEA2 in the putative crossover region could not be defined and are shown as X” as suggested (line 221).

(Reviewer #1’s Comments)

31. “one site on the left side on the top strand” is unclear and needs rewording. 

(Response) We have mentioned that “position is shown by black arrowhead” (Line 222) in the revised manuscript.

(Reviewer #1’s Comments)

32. Fig. 3 – line 270 – “no transconjugants were”.　

(Response) We have replaced “transconjugant” with “transconjugants were” (line 297). 

(Reviewer #1’s Comments)

33. Fig.4 - Legend – line 310 “Design of” can be deleted. 

(Response) “Design of” has been deleted (line 335). 

(Reviewer #1’s Comments)

34. Primer names are very long – can PCR reaction numbers be put in primer table?

(Response) We have shortened the primer name (line 338, Table 2).

(Reviewer #1’s Comments)

35. Line 215 – suggest “as shown as separate lines, with SE in orange”. 

(Response) We have rephrased the sentence as follows: “Two strands of DNA are shown as separate lines, with SE in orange” (line 339).

(Reviewer #1’s Comments)

36. “IS3 route”, “ICE/IME route” “gene cassette route” shown on the figure are not referred to in the legend. 

(Response) We have added “, referred to as ‘IS3 (copy-out) route’ at line 344. ‘ICE/IME route’ and ‘gene cassette route’ have been added at line 349 and line 350, respectively.

(Reviewer #1’s Comments)

37. Reduce overlap between description of parts A and B. 

(Response) We have deleted “Numbers in the primer set correspond to primer numbers in panel A” (line 351).

(Reviewer #1’s Comments)

38. Line 329 – “as arrowheads”. 

(Response) We have made the suggested correction (line 353).

(Reviewer #1’s Comments)

39. Lines 372-8 – legends should generally not contain methods.

(Response) Lines 372-378 in the previous version of the manuscript were not parts of figure legends.

(Reviewer #1’s Comments)

40. Fig. 5 – mention Ap in the title and add “Ap” before MIC on vertical axis label.

(Response) We added “Ap” to the title (line 393) and the vertical axis label in Fig 5.

(Reviewer #1’s Comments)

41. Fig. 6 - line 420 – “and carry”

(Response) We used “and carry” in the revised manuscript (line 248).

(Reviewer #1’s Comments)

42. 7) Supplementary Fig. S1 legend, line 870 “and TJ249”

(Response) We used “and TJ249” (line 648).

(Reviewer #1’s Comments)

43. Fig. S2 legend, line 885 – “genes were”, 

(Response) We used “genes were” in the revised manuscript (line 888).

(Reviewer #1’s Comments)

44. line 887 “PacBio assembly”

(Response) We have removed the information regarding contig 1 from the main text and from Fig S2 in the revised manuscript as it was not relevant to SEs (comment from reviewer 2). “PacBio assembly” has also been deleted.

(Reviewer #1’s Comments)

45. Fig. S3 legend – reword “part of the mobile DNA unit of the SE element” (also Fig. S4 lines 902-3, Fig. S7 line 930). 

(Response) We rephrased the sentence as follows: “Sequences in red are incorporated into circular copy of SE.” (line 896, line 904, line 926).

(Reviewer #1’s Comments)

46. Fig. S6 legend – suggest “(B) Relative copy number of attS in V. alfansensis LN95 (no pSEA2)”

(Response) We have modified the figure title as suggested (line 936).

(Reviewer #1’s Comments)

47. Relative to what?

(Response) We used “relative copy number to gyrB” (line 938).

(Reviewer #1’s Comments)

48. 8) Minor/formatting/English

The Methods section could be simplified/condensed by restricting information to what was done, rather than too much description of why, especially if already explained in Results. Some minor rewording for accuracy/clarity is also needed to make the manuscript easier to read (some suggestions are included below).

(Response) We have simplified and condensed the text. We agree with almost all of Reviewer 1’s comments and have revised the manuscript according to the reviewer’s suggestion.

(Reviewer #1’s Comments)

PREVIOUS COMMENT: 14. Line 240 – V. cholerae is not always from sea food?

(Response) V. cholerae is indeed not always from sea foods as described in the cited reviews. These descriptions are correct. Thus we did not make changes to the revised manuscript.

49. NEW COMMENT: now line 430. I don’t understand this response. If these Vibrios are not always from sea food then the wording should be changed. Vibrio is one of the major bacterial genera found in marine sediments [26], and is among the most common microbiota of wild and farmed shrimp [27] and fish [28, 29]. Several of the genus Vibrio are pathogens of fishes reared in aquaculture [30], while other subsets of Vibrio species, including V. cholerae, V. parahaemolyticus, and V. vulnificus, which are ubiquitous in relatively low-salinity sea water, are seafood-borne human pathogens [30, 31].

(Response) We have modified the sentence as follows: “, which are ubiquitous in relatively low-salinity sea water, are human pathogens from seafood and/or water etc. [49, 50]” (line 427).

(Reviewer #1’s Comments)

PREVIOUS COMMENT: 36. Line 184 – meaning actually “chromosomal copy of Tn6945 in V. alfacsensis is functional”?

(Response) Because only one SE-6945 copy is present in LN95, detection of circular SE-6945 in LN95 indicates that the cis-elements (attL/attR) and strand exchange enzymes encoded in the chromosomal SE-6945 is indeed functional, while those of SE-6945 on pSEA2 is obviously functional as its transposition was observed in E. coli. We explicitly stated that “This suggests that the cis-elements (attL/attR) and strand exchange enzymes encoded by the chromosomal copy of SE-6945 are also functional in V. alfacsensis” in lines 304-306 in the revised manuscript.

50. NEW COMMENT: This was a comment on wording only. I think that what I previously suggested is clearer

(Response) We have stated that “the chromosomal copy of SE-6945 in V. alfacsensis is functional” as suggested (line 329).

 (Reviewer #1’s Comments)

PREVIOUS COMMENT: 38. Line 199 – “absence of pSEA-2 from”

51. NEW COMMENT: Sorry, this should have referred to line 299, now line 515, and is simply a wording issue. If pSEA2 is absent then it is not “in” LN95, so “from” is clearer/more correct.

(Response) We have used “the absence of pSEA2 from LN95” in the revised manuscript (line 519).

(Reviewer #1’s Comments)

OTHER NEW COMMENTS: 

52. Lines 37-8 – could delete “in addition to conjugative plasmids”

(Response) “in addition to conjugative plasmids” before “SEs are” have been deleted (line 39).

(Reviewer #1’s Comments)

53. Line 61 - “of an integron cassette” or “of integron gene cassettes” and “the phage CTX“

(Response) We used “integration of integron gene cassettes into attI [16] and the phage CTX into dif” as suggested (line 63).

(Reviewer #1’s Comments)

54. Line 33 – “The copy number and location” 

(Response) We used “The copy number and location” in the revised manuscript (line 36).

(Reviewer #1’s Comments)

55. Line 64 – “a Y-transposase”, “an S transposase.

(Response) We have used “a Y-transposase” and “an S transposase”, as suggested (line 65).

(Reviewer #1’s Comments)

56. Line 67 – please give full names for these elements in addition to abbreviations. 

(Response) We expanded abbreviated terms as follows: “IME (integrative mobilizable element) [18], MGI (mobilizable genomic island), CIME (cis-mobilizable element) [19], or MTn (mobilizable transposon) [20].” (line 68-70). 

(Reviewer #1’s Comments)

57. Line 92 – “the E. coli chromosome”

(Response) We have used “the E. coli chromosome” as suggested (line 94).

(Reviewer #1’s Comments)

58. Line 99 – “as” before transposition should be omitted.

 (Response) We have omitted “as” (line 102). 

(Reviewer #1’s Comments)

59. Line 114 – “and transconjugants selected on erythromycin” 

(Response) We have used “transconjugants were selected on erythromycin at 42 °C” as suggested (line 113-114).

(Reviewer #1’s Comments)

60. Line 120 – please give the accession no. for the CAIM 1831 sequence here or in Fig. S1.

(Response) The accession numbers for CAIM 1831 replicons have been added to revised Fig. S1.

(Reviewer #1’s Comments)

61. Line 121 – “plasmid” is not needed before pSEA2.

(Response) The term “plasmid” has been deleted (line 122).

(Reviewer #1’s Comments)

62. Line 122 – “a 7.1 kb repeat region”, not “the 7.1 kb repeat region”

(Response) We have used “a” (line 122).

(Reviewer #1’s Comments)

63. Line 133 – suggest “predicted” rather than “estimated”.

(Response) We have used “predicted” (line 144).

(Reviewer #1’s Comments)

64. Lines 159-60 – maybe “also carries two copies of SE-6945, in the same locations on the chromosome and plasmid as in 04Ya249”? 

(Response) We have rephrased the sentence as follows: “also carries two copies of SE-6945, in the same locations on the chromosome and plasmid as in 04Ya249” (line 140-141).

(Reviewer #1’s Comments)

65. Line 186 – “transconjugants were screened on Tc alone, …”

(Response) We have used “transconjugants were screened on tetracycline (Tc) alone,” (line184).

(Reviewer #1’s Comments)

66. Line 195 - “other transconjugants” 

(Response) We have used “other transconjugants” (line 193).

(Reviewer #1’s Comments)

67. Line 198 – delete “the” 

(Response) We deleted “the” before “SE-6945 transposition” (line 196). 

(Reviewer #1’s Comments)

68. Line 201 – “searched for”

(Response) We have added “for” (line 199).

(Reviewer #1’s Comments)

69. Line 214 - “integrons”.

(Response) We replaced “Integron” with “integrons” (line 212).

(Reviewer #1’s Comments)

70. Lines 234-5 – “altogether in a recA-null”

(Response) We replaced “altogether in the chromosome of a recA-null mutant” with “altogether in a recA-null mutant” (line 260).

(Reviewer #1’s Comments)

71. Lines 243 – this needs rewording - transconjugants are not transferred.

(Response) We want to emphasize the fact that transconjugants were obtained, albeit at a low frequency. Thus, the statement remains the same in the revised manuscript (line 266). We confirmed that these colonies received resistance genes and not spontaneous mutants by conducting single colony isolation and PCR.

(Reviewer #1’s Comments)

72. Line 246 – “in any of the four”

(Response) We have used “in any of the four” (line 272).

(Reviewer #1’s Comments)

73. Line 248 – suggest “the transfer frequency of Ap resistance to JW0452deltarecArif was about 17 fold lower than transfer to JW0452rif”. Was this for the means?

(Response) Yes. These are for mean values. We rephrased the sentence as follows: “the mean transfer frequency of Ap resistance to JW0452ΔrecArif was about 17-fold lower than transfer to JW0452rif” (line 274).

(Reviewer #1’s Comments)

74. Line 253 - “in mean” needs rewording.

(Response) We rephrased the sentence as follows: “However, the mean transfer frequency of Tc Ap resistance to LN52rif (which already carries one copy of SE-6945 and wild type recA) was about 56-fold higher than transfer to JW0452rif.” (line 277). 

(Reviewer #1’s Comments)

75. Line 256 – “integration into the recipient chromosome”

(Response) We rephrased the sentence as follows: “promotes cointegration of pSEA2 and the recipient chromosome” in the revised manuscript (line 281).

(Reviewer #1’s Comments)

76. Line 281 – fix “E. col”

(Response) Corrected. (line 307).

(Reviewer #1’s Comments)

77. Lines 282-3 – could this be expressed the other way round? i.e., tyrosine recombinase typically produce staggered cuts at att sites?

(Response) Staggered cuts indicate occurrence of a double-strand break, and are misleading (the product is different from the staggered cuts made by restriction enzyme). Therefore, we did not change the phrase (line 308-309). 

(Reviewer #1’s Comments)

78. Lines 288, 291, 360 etc – “single stranded”, “double stranded”

(Response) We corrected “single-strand” and “double-strand” to “single stranded” (lines 311, 314, 316, 317, 342, 344, 349, 462, 463, 477, 490) and “double stranded” (lines 309, 343, 346, 348, 384) respectively, through the whole text.

(Reviewer #1’s Comments)

79. Line 297 – “anneals inside the integrative element and the other outside it”

(Response) We modified the sentence as suggested (line 323).

(Reviewer #1’s Comments)

80. Line 301 – “primer sets”.

(Response) The primer set name was 1-4, but not 1, 2, 3, or 4. Therefore, to avoid confusion, we added ‘ ’ and used ‘1-4’ instead of 1-4 (line 326).

(Reviewer #1’s Comments)

81. Line 308 – “long PCR conditions were used” 

(Response) We used “were used” (line 332).

(Reviewer #1’s Comments)

82. Line 372 – “their genome”

(Response) We have used “their genomes” (line 396). 

(Reviewer #1’s Comments)

83. Lines 379-80, 387- need rewording - only Ap MICs were obtained. 

(Response) We used “Ap” instead of “antibiotic” (Line 403, line 411, new Fig 5).

(Reviewer #1’s Comments)

84. Line 388 – “MICs against individual strains are”. Not clear what “when focused on a class of MIC 500” means. 

(Response) We have added the phrase “against individual strains” (line 405). The line “when focused on a class of MIC 500” has been deleted. (line 412).

(Reviewer #1’s Comments)

85. Line 405 – “screened for”, or “identified” 

(Response) We have used “screened for” (line 231).

(Reviewer #1’s Comments)

86. Line 406 – “and related SEs” 

(Response) We have omitted “its” as suggested (line 232).

(Reviewer #1’s Comments)

87. Line 431 – “linking” not really the right word here, “aiding movement between”?

(Response) We have used “aiding movement of” instead of “linking” as suggested (line 428).

(Reviewer #1’s Comments)

88. Line 462 - “transposon mechanisms”? 

(Response) Here, we have used “movements” in the revised manuscript (as suggested by reviewer 2) (line 459).

(Reviewer #1’s Comments)

89. Line 469 – needs rewording

(Response) Please refer to our response to comment 20.

(Reviewer #1’s Comments)

90. Lines 288-9, 469– please reconsider descriptions relating to integron (upper case I not needed) gene cassettes and “empty donor sites”. The attC site belonging to a particular cassette moves with that cassette. Excision may create an empty attI if it was the only cassette, otherwise the downstream cassette moves up the array. Also, Ref. 40 on line 469 dates from before the single stranded mechanism was identified.

(Response) Please see our response to comment number 20.

(Reviewer #1’s Comments)

91. Line 473 - “unjoined”? 

(Response) We used “nicked” because this word was suggested by the Language editor at Editage (line 487).

(Reviewer #1’s Comments)

92. Line 475 – “a process equivalent to”

(Response) We have rephrased the sentence as suggested (line 475).

(Reviewer #1’s Comments)

93. Line 496 – “relaxases”, 

(Response) We have replaced “relaxase” with “relaxases” (line 497).

(Reviewer #1’s Comments)

94. “and oriT in SE-6945 or SE-6282” 

(Response) We have replaced “SE-6945 and SE-6283” with “in SE-6945 or SE-6283” (line 498). 

(Reviewer #1’s Comments)

95. Lines 567-8 would be better incorporated into lines 563-5 i.e. add “and incubated at 42deg C to supress growth of Vibrio strains” to the end of the sentence. See also comments above on incorporating information into Table 1.

(Response) We have moved up Lines 567-8: “and incubated at 42°C to suppress growth of Vibrio strains” (line 567).

The following sentence have been deleted: “The transconjugant strain TJ249 was obtained by the first mating experiment of strain 04Ya249 with E. coli strain JW0452 as the donor and recipient, respectively, followed by Ery selection at 42 °C”.

(Reviewer #1’s Comments)

96. References – species names should be in italics. 

(Response) We have looked at the reference sections, carefully and confirmed that all species names are in italic (Lines 663-869).

Reviewer #2: The revision of the manuscript has clarified many aspects of the original version. I don't think it will be productive to argue the utility of general verses specific use of the words transposon/transposition. More importantly, I feel that the use of "SE" in place of "Tn" is a satisfactory solution that clearly implies distinctions between these particular integrative elements from others, and Tn-named elements.

Further suggestions:

(Reviewer #2’s Comments)

1. Line 25: "multidrug resistance conjugative plasmid" is probably better written as "conjugative multidrug resistance plasmid".

(Response) We have used "multidrug resistance conjugative plasmid" as suggested (line 27).

(Reviewer #2’s Comments)

2. Line 31: "of its one specific" can just be "of one specific".

(Response) We have used "of one specific" (line 33).

(Reviewer #2’s Comments)

3. Line 33: "Copy numbers and location" should be "The copy number and location".

(Response) We added “The” before “copy number (line 36).

(Reviewer #2’s Comments)

4. Line 48: Sentence should be: Conjugative plasmids and integrative and conjugative elements (ICE) [6] are DNA units that can move from one cell to another through conjugation machinery that they encode. 

(Response) We have added “that they encode” as suggested (line 51). 

(Reviewer #2’s Comments)

5. Line 100: Just a suggestion. Replicon fusion between plasmids and/or chromosome could be termed "cointegration", leaving "integration" to be used for site-specific recombination, since "insertion" is commonly used for events not involving site-specific recombination.

(Response) We appreciate the reviewer’s suggestion. We introduced the term “cointegration” to refer to plasmid-chromosome cointegration, and used “integration” for site-specific integration (also suggested by Reviewer 1). We did not declare usage for either “insertion” or ” translocation” in the revised manuscript. The terms definition paragraph (line 99) is now: “For clarity, terms transposition, cointegration, and integration are used here based on the following rules. We call the movement of a DNA segment from one location to the other location in one or two genomes, mediated by any DNA strand exchange enzymes, transposition. The physical incorporation of a plasmid into a chromosome is called cointegration regardless of the molecular mechanisms and genomic locations involved. The physical incorporation of a DNA molecule into a specific site in another molecule mediated by a site-specific recombinase (so-called integrase) are called integration” (line 99-106).

(Reviewer #2’s Comments)

6. Line 103: I appreciate the authors explaining why they haven't used the term "translocation". However, since it is not used the explanation isn't needed in the paper - it might appear perplexing to a reader. 

(Response) We have deleted the sentence as suggested (see above).

(Reviewer #2’s Comments)

7. Line 114: "and the transconjugant was first selected" should be "and transconjugants were selected".

(Response) We have modified the sentence as suggested (line 113). 

(Reviewer #2’s Comments)

8. Line 166 and Fig S2: The significance of contig 1 is not clear to me. How does it indicate movement of SE-6945?

(Response) All statements related to “contig 1” have been deleted from the revised manuscript and revised Fig. S2, as it is not relevant to SEs.

(Reviewer #2’s Comments)

9. Line 186: "transconjugant was" should be "transconjugants were".

(Response) We have used “transconjugants were” (line 184).

(Reviewer #2’s Comments)

10. Line 190: " generated intA-containing fragments originating from" should be "possessed intA-containing fragments located at".

(Response) We have used “possessed” (line 188).

(Reviewer #2’s Comments)

11. Line 197: It is more correct to say that pSEA2 integration relies on SE-6945 transposition, since generally transposition of SE-6945 will NOT result in plasmid integration (it will do so only rarely).

(Response) We added the lines ‘pSEA2 integration relies on SE-6945 transposition’ (line 195). 

‘Since generally transposition of SE-6945 will NOT result in plasmid integration’ was not added, as this additional phrase might be rather distracting.

(Reviewer #2’s Comments)

12. Line 212 and elsewhere: "Sequence logos" should be "Sequence logo".

(Response) We replaced “Sequence logos” with "Sequence logo" (line 223).

(Reviewer #2’s Comments)

13. Line 257: "The Tn3-like" should be "Tn3-like".

(Response) We corrected "The Tn3-like" to "Tn3-like" (line 283).

(Reviewer #2’s Comments)

14. Line 347 and 353: (Fig 4E) should be (Fig 4D). 

(Response) We have changed (Fig 4E) to (new Fig 5D) in line 377 and line 379.

(Reviewer #2’s Comments)

15. Line 371: "in genomes" should be "in their genomes".

(Response) The manuscript was revised as per the reviewer’s suggestion (line 395).

(Reviewer #2’s Comments)

16. Line 372: Why/how would/could SE-6283 be expected to influence beta lactam resistance? Seems like a complete red herring to me. 

(Response) We used SE-6283 here as an example of an SE having two target sites in a genome, to emphasize that SE copy number in a genome can increase. In this study, we used SE-6945 carrying bla to illustrate that copy number of SEs in a genome can affect any phenotype. However, we felt that the phrase, “Furthermore, 04Ya108 also carries two copies of SE-6283” was not necessary to express our message. We have deleted this sentence in the revised manuscript (line 396).

(Reviewer #2’s Comments)

17. Line 388-391 paragraph: Could just be: The MIC of individual strains is shown in Table S1. There was a trend for strains with SE-6945 insertion into insJ to show a higher MIC than strains with insertion into yjjNt. Thus, the location of SEs can result in phenotype variation, possibly due to the differences in the expression levels of their cargo genes. 

(Response) We have modified the sentence as suggested (line 412).

(Reviewer #2’s Comments)

18. Line 414: As written, the sentence implies the opposite of what is meant. I suggest change to "and, like attB of SE-6945 did not possess an inverted repeat structure."

(Response) We appreciate Reviewer 2’s suggestion. We have added “and, like attB of SE-6945 did not possess an inverted repeat structure” to the revised manuscript (line 240). 

(Reviewer #2’s Comments)

19. Line 418: Isn't it five genomic locations shown in Fig 6A?

(Response) We appreciate the Reviewer 2’s comment. We have changed “six” to “five” (line 245). 

(Reviewer #2’s Comments)

20. Line 462: "movements" instead of "moves"

(Response) We have changed “moves” to “movements” (line 459).

(Reviewer #2’s Comments)

21. Line 471: "possibly the CDS2" rather than "i.e., CDS2".

(Response) We have changed "i.e., CDS2" to “possibly the CDS2” (line 471).

(Reviewer #2’s Comments)

22. Line 473: " transposases" instead of "transposase".

(Response) We have replaced "transposase" with "transposases" (line 473).

(Reviewer #2’s Comments)

23. Line 497: It might be worth mentioning here that a tyrosine recombinase-like protein has been shown to function as a conjugative relaxase, particularly since these elements contain two such proteins. See reference https://doi.org/10.1111/mmi.13270.

(Response) We appreciate this useful information and suggestion by Reviewer 2. We added the following lines about the possible function of IntB as relaxase to the Discussion: “It may also be possible that IntB, a large tyrosine recombinase, functions as a relaxase, since a conjugative plasmid from Clostridium has been shown to use a tyrosine recombinase homolog as a conjugative relaxase.” (line 498-501). The relevant paper has been cited in the revised manuscript.

---

## [Decision Letter · Decision Letter 2]

17 Jun 2022

PONE-D-21-18582R2Atypical integrative element with strand-biased circularization activity assists interspecies antimicrobial resistance gene transfer from Vibrio alfacsensisPLOS ONE

Dear Dr. Nonaka,

Thank you for submitting your manuscript to PLOS ONE. After careful consideration, we feel that it has merit but does not fully meet PLOS ONE’s publication criteria as it currently stands. Therefore, we invite you to submit a revised version of the manuscript that addresses the points raised during the review process. Please consider the comments of the reviewers to improve the manuscript.

We look forward to receiving your revised manuscript.

Kind regards,

Axel Cloeckaert

Academic Editor

PLOS ONE

Reviewers' comments:

Reviewer's Responses to Questions

**Comments to the Author**

1. If the authors have adequately addressed your comments raised in a previous round of review and you feel that this manuscript is now acceptable for publication, you may indicate that here to bypass the “Comments to the Author” section, enter your conflict of interest statement in the “Confidential to Editor” section, and submit your "Accept" recommendation.

Reviewer #1: (No Response)

Reviewer #2: (No Response)

2. Is the manuscript technically sound, and do the data support the conclusions?

Reviewer #1: Yes

Reviewer #2: Yes

3. Has the statistical analysis been performed appropriately and rigorously? 

Reviewer #1: I Don't Know

Reviewer #2: N/A

4. Have the authors made all data underlying the findings in their manuscript fully available?

Reviewer #1: Yes

Reviewer #2: Yes

5. Is the manuscript presented in an intelligible fashion and written in standard English?

Reviewer #1: No

Reviewer #2: Yes

6. Review Comments to the Author

Reviewer #1: This revised manuscript is improved, and I have not commented further on the organisation, as the authors have provided reasons why they do not want to change this. The manuscript is essentially now acceptable for publication, but as PLoS One does not copyedit accepted manuscripts, addressing the few small points below, which include suggestions to further improve English/wording, would enhance clarity/readability.

1) ORIGINAL COMMENT: 71. Lines 243 – this needs rewording - transconjugants are not transferred.

(Response) We want to emphasize the fact that transconjugants were obtained, albeit at a low frequency. Thus, the statement remains the same in the revised manuscript (line 266). We confirmed that these colonies received resistance genes and not spontaneous mutants by conducting single colony isolation and PCR.

NEW COMMENT: Sorry if this was not clear – it was simply a comment on the wording – “the transfer of Tc Ap-resistant transconjugants” does not make sense - transconjugants are not transferred, they are the result of transfer. It could be changed to e.g., “transfer of both Ap and Tc resistance”.

2) ORIGINAL COMMENT 77. Lines 282-3 – could this be expressed the other way round? i.e., tyrosine recombinase typically produce staggered cuts at att sites?

(Response) Staggered cuts indicate occurrence of a double-strand break, and are misleading

(the product is different from the staggered cuts made by restriction enzyme). Therefore, wedid not change the phrase (line 308-309).

NEW COMMENT: Lines 308-9 - “this assertion is based on the notion that tyrosine recombinases have a nature to not produce blunt ends at att sites on double stranded DNA during strand exchange”. This wording is quite convoluted – would e.g. “recombinases generally do not produce blunt ends” work?

3) Minor wording suggestions

Lines 34-5 – “of integron gene cassettes”

Lines 76-7, 83-4 – abbreviation AMR could be explained on lines 76-7.

Line 89 – “elements”

Line 96 – suggest delete “the currently unclassifiable integrative element”

Line 99 – “movement”

Line 100 – “the terms”

Line 106 – “is called”

Lines 121-3 – “VHH-1”, “GMA” and “GMA-1” should not be in italics.

Line 130 – “insJ”

Lines 132-5 – need rewording for clarity, delete “member of”

Line 136 – “in the aligned region”

Lines 151-3 suggest “SE-6283 and SE-6945 carry imperfect 19 bp inverted repeat motifs at their ends, (Fig 1 B), termed C and C’ based apparent on functional equivalence to the core-type sites of phages” (assuming “Fig. 1 fB is not correct here?).

Line 169 – “the two E. coli chromosomal locations and the two locations”

Line 182 – “a specific location in the”

Line 185 – “The SE-6945 integration”

Line 187, 194 – “selection with Ap alone”

Line 188 – “transconjuants, 12 had integrase”

Line 199 – “the NCBI”

Line 201- “contains”

Line 203 – “provisionally termed”

Lines 209-10 need rewording

Line 211- “is T rich”, “structures”

Line 240 “are G”

Lines 241-2 - “suggest that, like ICEs, SEs target a few selected locations”.

Line 284 – “nicks”

Line 291 – “mechanism, transposition followed by homologous recombination, rather than through

Line 306 – “upon circularization”

Line 314 – “or the generation of circular integron gene cassettes”, why “donor sites”

Line 332 - “when long”

Lines 372-3 – “at the C end”

Line 384 – “nicking of one strand of”

Line 400 “The location”

Line 412 – “of strains are”

Line 413 – “integrated”

Line 484-5 – “in transposition”

Line 459 “Among the types of transposon movement”

Line 486-7 – “with fewer gene cassettes”

Line 469 – “barely accompanies” not clear

Line 472 “of gene cassettes”

Line 481 – “was repeatedly”?

Lie 482 - delete “step”

Line 496 - “an HUH”

Line 505 - “numbers” rather than “digits”?

Line 514 – replace “strain 04Ya108” with “it”

Line 516 “re-classified”

Line 530 – “The 19 transconjugants listed ()”

Lines 616-7 – “Ap concentrations”

Line 632 – “searched for”

Reviewer #2: The authors have satisfactorily addressed all my previous comments. A few minor corrections are listed below.

Minor corrections:

Line 27: "multidrug resistance conjugative plasmid" should have been changed to "conjugative multidrug resistance plasmid".

Line 106: "are called" should be "is called".

Fig. 1C: The location of the conserved R at the top of Box I appears to be misaligned by two residue positions.

After line 180: It would be helpful for readers for S5 Fig B was referred to specifically at the appropriate place.

Line 196: "but not" should be "rather than".

Line 209: "nucleotide" should be "nucleotides".

Line 210: "Sequence" should be "sequence".

Line 222: "trshown" should presumably just be "shown".

Line 249: "Sequence logos" should be "sequence logo".

Line 287: "transconjugant" should be "transconjugants".

Line 348: "circular a" should be "a circular".

Line 412: "strain" should be "strains".

Line 428: "water etc" should just be "water".

Line 496: "carry oriT" should be "carry an oriT".

Line 498: "identify oriT" should be "identify an oriT".

Line 502: "MDR conjugative plasmids" should be "conjugative MDR plasmids".

S5 Fig B legend. This legend is inadequate. It should at least say it is a Southern blot and give some insight into the probe used - can refer to S5 Fig A legend where it is shown (but not described). Likewise for the S5 Fig C legend.

7. PLOS authors have the option to publish the peer review history of their article (what does this mean?). If published, this will include your full peer review and any attached files.

Reviewer #1: No

Reviewer #2: No

---

## [Author Response · Author response to Decision Letter 2]

4 Jul 2022

Responses to the Reviewer’s comment

PONE-D-21-18582

Reviewer #1: This revised manuscript is improved, and I have not commented further on the organisation, as the authors have provided reasons why they do not want to change this. The manuscript is essentially now acceptable for publication, but as PLoS One does not copyedit accepted manuscripts, addressing the few small points below, which include suggestions to further improve English/wording, would enhance clarity/readability.

(Reviewer #1’s Comments)

1) ORIGINAL COMMENT: 71. Lines 243 – this needs rewording - transconjugants are not transferred.

(Response) We want to emphasize the fact that transconjugants were obtained, albeit at a low frequency. Thus, the statement remains the same in the revised manuscript (line 266). We confirmed that these colonies received resistance genes and not spontaneous mutants by conducting single colony isolation and PCR.

NEW COMMENT: Sorry if this was not clear – it was simply a comment on the wording – “the transfer of Tc Ap-resistant transconjugants” does not make sense - transconjugants are not transferred, they are the result of transfer. It could be changed to e.g., “transfer of both Ap and Tc resistance”.

(Response)

We have used as “transfer of both Ap and Tc resistance” as suggested by reviewer 1 (line 268).

(Reviewer #1’s Comments)

2) ORIGINAL COMMENT 77. Lines 282-3 – could this be expressed the other way round? i.e., tyrosine recombinase typically produce staggered cuts at att sites?

(Response) Staggered cuts indicate occurrence of a double-strand break, and are misleading

(the product is different from the staggered cuts made by restriction enzyme). Therefore, we did not change the phrase (line 308-309).

NEW COMMENT: Lines 308-9 - “this assertion is based on the notion that tyrosine recombinases have a nature to not produce blunt ends at att sites on double stranded DNA during strand exchange”. This wording is quite convoluted – would e.g. “recombinases generally do not produce blunt ends” work?

(Response) We have used “recombinases generally do not produce blunt ends” as suggested by reviewer 1 (line 307).

3) Minor wording suggestions

(Reviewer #1’s Comments)

Lines 34-5 – “of integron gene cassettes”

(Response) We have changed “of the integron gene cassettes” to “of integron gene cassettes” as suggested (line 34-35).

(Reviewer #1’s Comments)

Lines 76-7, 83-4 – abbreviation AMR could be explained on lines 76-7.

(Response) We have introduced “antimicrobial resistance (AMR) genes” at lines 76-77, then used “AMR genes” at lines 83-84 as suggested by reviewer 1.

Line 89 – “elements” 

(Response) We have changed "interspecies gene transfer assisted by Tn6283-like element is common" to “interspecies gene transfer assisted by Tn6283-like elements is common” as suggested (line 88-89).

Line 96 – suggest delete “the currently unclassifiable integrative element”

(Response) We deleted this phrase.

Line 99 – “movement”

(Response) We have changed “several types of intracellular DNA movements” to "several types of intracellular DNA movement”, as suggested (line 98).

Line 100 – “the terms”

(Response) We have changed “terms” to “the terms” (line 99).

Line 106 – “is called”

(Response) We have changed “are called” to “is called” (line 105).

Lines 121-3 – “VHH-1”, “GMA” and “GMA-1” should not be in italics.

(Response) We corrected (line 120, line 122 and line 123).

Line 130 – “insJ”

(Response) We have changed “InsJ” to “insJ” (line 129).

Lines 132-5 – need rewording for clarity, delete “member of”

(Response) “member of “ has been deleted.

Line 136 – “in the aligned region”

(Response) We have changed “of the aligned region” to “in the aligned region” as suggested (line 135).

Lines 151-3 suggest “SE-6283 and SE-6945 carry imperfect 19 bp inverted repeat motifs at their ends, (Fig 1 B), termed C and C’ based apparent on functional equivalence to the core-type sites of phages” (assuming “Fig. 1 fB is not correct here?).

(Response) We rephrased the sentence according to the reviewer’s suggestion. However, reference to “Fig. 1B remains because it actually shows the sequences of C and C’ below genetic map (line 150-152).

Line 169 – “the two E. coli chromosomal locations and the two locations”

(Response) We have changed “two E. coli chromosomal locations” to “the two E. coli chromosomal locations” as suggested (line 168).

Line 182 – “a specific location in the”

(Response) We have changed “a specific location of E. coli chromosome” to “a specific location in the E. coli chromosome” as suggested (line 181).

Line 185 – “The SE-6945 integration”

(Response) We have changed “SE-6945 integration pattern” to “The SE-6945 integration pattern” as suggested (line 184). 

Line 187, 194 – “selection with Ap alone”

(Response) We have changed “Ap alone selection” to “selection with Ap alone” (line 186 and line 193).

Line 188 – “transconjuants, 12 had integrase…”

(Response) We used “transconjuants, 12 strains had intA…”, because integrase is protein name and gene name intA is rather appropriate here (line 187).

Line 199 – “the NCBI”

(Response) We added “the” before “NCBI” (line 198).

Line 201- “contains”

(Response) We used “contains” instead of “contained” (line 200).

Line 203 – “provisionally termed”

(Response) We changed “tentatively termed” to “provisionally” as suggested (line 202).

Lines 209-10 need rewording

(before update) Sequences of unoccupied target sites (attB) in E. coli, V. alfacsensis, and V. harveyi are shown in Fig 2 B.

 (Response) We have removed “unoccupied” from the original sentence and the sentence is now simply “Sequences of target sites (attB) in E. coli, V. alfacsensis, and V. harveyi are shown in Fig 2 B (line 208-209).”

Line 211- “is T rich”, “structures”

(Response) We changed “was” to “is”, and “structure” to “structures”, respectively. The updated sentences are the followings: “The central 6 bp putative crossover region in attB is T rich. However, other motifs, such as imperfect inverted repeat structures observed in attC/attI of integrons (IntI target), or dif of CTX phage integration site (XerC/XerD target), could not be detected in attB, attL (S3 Fig), or attR (S4 Fig) for SE-6945 in the current data set” (line 209-213).

Line 240 “are G”

(Response) We used “are G” instead of “were G” (line 239).

Lines 241-2 - “suggest that, like ICEs, SEs target a few selected locations”.

(Response) We rephrased according to the reviewer’s suggestion (line 240-241).

Line 284 – “nicks”

(Response) We changed “nick” to “nicks” (line 283).

Line 291 – “mechanism, transposition followed by homologous recombination, rather than through

(Response) We changed “mechanism; transposition followed by homologous recombination than” to “mechanism, transposition followed by homologous recombination, rather than” as suggested by reviewer 1 (line 290).

Line 306 – “upon circularization”

(Response) We changed “upon its circularization” to “upon circularization” as suggested (line 305).

Line 314 – “or the generation of circular integron gene cassettes”, why “donor sites”

(Response) We changed “or the single stranded circle generation of Integron gene cassette by attC × attC recombination (gene cassette route), both of which generate empty donor sites” to “or the generation of cirucular integron gene cassettes by attC × attC recombination (gene cassette route)” (line 312-314). 

We used “empty donor sites” to emphasize the common feature of ICE excision and gene cassette generation. The reviewer 1 might have confused with the definition of “donor site” on gene cassettes. We have removed “generation of empty donor site” from the sentence to simplify the sentence, as it will not affect our main message. 

Line 332 - “when long”

(Response) We changed “when the long PCR conditions “to” “when long PCR conditions” (line 330).

Lines 372-3 – “at the C end”

(Response) We have changed “nicking occurs 6 bp upstream at the 5’ of C end and 3’ of C end” to “nicking occurs 6 bp upstream at the C end and the C′ end and/or 6 bp upstream at the C′ end and the C end”, as suggested. 

(Response 2) We have changed nicking occurs 6 bp upstream of the 5′ end of C side and 3′ end of C′ side and/or 6 bp upstream of the 5′ end of C′ side and 3′ end of C side to “nicking occurs 6 bp upstream at the C end and the C′ end and/or 6 bp upstream at the C′ end and the C end”, as suggested (line 369-370). 

Line 384 – “nicking of one strand of”

(Response) We have changed “Nicking at one strand on the double stranded DNA” to “Nicking of one strand of the double stranded DNA” as suggested (line 381).

Line 400 “The location”

(Response) We have changed “Location of SE-6945” to “The location of SE-6945” (line 397).

Line 412 – “of strains are”

(Response) “The MIC of individual strain is shown in Table 1” to “The MICs of strains are shown in Table S1.” (line409).

Line 413 – “integrated”

(Response) We have changed “SE-6945 integration into insJ “ to” to “SE-6945 integrated into insJ” (line 409-410).

Line 484-5 – “in transposition”

(Response) We have changed “SE-6283 transposition from” to “SE-6283 in transposition from”, as suggested (line 481-482).

Line 459 “Among the types of transposon movement”

(Response) We have changed “Among the transposon movements” to “Among the types of transposon movement” (line 456).

Line 486-7 – “with fewer gene cassettes”

(Response) We have changed “with reduced number of gene cassettes” to “with fewer gene cassettes” as suggested (line468-9).

Line 469 – “barely accompanies” not clear

(Response) We have changed “SE circle formation barely accompanies the generation of empty attB” to “SE circle formation rarely leaves the empty attB.”

Line 472 “of gene cassettes”

(Response) We have changed “of the gene cassette” to “of gene cassettes” (line 468-469).

Line 481 – “was repeatedly”?

(Response) We have changed “is repeatedly” to “was repeatedly” (line 477).

Lie 482 - delete “step”

(Response) We have deleted “step”.

Line 496 - “an HUH”

(Response) We have changed “HUH endonuclease” to “an HUH endonuclease” (line 493).

Line 505 - “numbers” rather than “digits”?

(Response) We have changed “digits” to “numbers” as suggested (line 503).

Line 514 – replace “strain 04Ya108” with “it”

(Response) We updated the sentence as follows: 

(before update) Strain 04Ya108 was previously identified as Vibrio ponticus based on 16S rRNA gene sequence similarity. However, determination of the complete sequence in this study revealed that strain 04Ya108 shows >96% average nucleotide identity to V. alfacsensis strain CAIM 1831 (DSM 24595) S1 Fig B [67]. 

(after update) Strain 04Ya108 was previously identified as Vibrio ponticus based on 16S rRNA gene sequence similarity. However, determination of the complete sequence in this study revealed that it shows >96% average nucleotide identity to V. alfacsensis strain CAIM 1831 (DSM 24595) S1 Fig B [67] (line 509-512).

Line 516 “re-classified”

(Response) We have changed “newly classified” to “re-classified” (line 513).

Line 530 – “The 19 transconjugants listed ()”

(Response) We have changed “19 transconjugants listed” to “The 19 transconjugants listed” (line 527).

Lines 616-7 – “Ap concentrations”

(Response) We have changed “antibiotics concentrations” to “Ap concentrations” (line 613-614).

Line 632 – “searched for”

(Response) We have changed “searched“ to “searched for” (line 629).

Reviewer #2: The authors have satisfactorily addressed all my previous comments. A few minor corrections are listed below.

Minor corrections:

Line 27: "multidrug resistance conjugative plasmid" should have been changed to "conjugative multidrug resistance plasmid".

(Response) We have changed "multidrug resistance conjugative plasmid" to "conjugative multidrug resistance plasmid" (line 27).

Line 106: "are called" should be "is called".

(Response) Corrected (line 102) (reviewer 1 also pointed out).

Fig. 1C: The location of the conserved R at the top of Box I appears to be misaligned by two residue positions.

(Response) Thank you for pointing out this. Indeed, "R” was misaligned in the updated figure. We have uploaded corrected Fig F1.

After line 180: It would be helpful for readers for S5 Fig B was referred to specifically at the appropriate place.

(Response) We referred to S5Fig B in the following sentence (line 318-322). “we created a pSEA2-free 04Ya249 derivative strain LN95 carrying only a single copy of SE-6945 in the chromosome (S5 Fig B) and analyzed the production of its circular molecule in this strain and the sequence of strand exchange products.”

Line 196: "but not" should be "rather than".

(Response) We replaced “but not” with "rather than" (line 195).

Line 209: "nucleotide" should be "nucleotides".

(Response) We have changed "nucleotide" to "nucleotides" (line 208).

Line 210: "Sequence" should be "sequence".

(Response) We have changed “Sequence logo” to “sequence logo” (line 209).

Line 222: "trshown" should presumably just be "shown".

(Response) We corrected (line 221).

Line 249: "Sequence logos" should be "sequence logo".

(Response) We corrected (line 248).

Line 287: "transconjugant" should be "transconjugants".

(Response) Corrected (line 286).

Line 348: "circular a" should be "a circular".

(Response) Corrected (line 346).

Line 412: "strain" should be "strains".

(Response) Corrected (line 410) (reviewer 1 also pointed out).

Line 428: "water etc" should just be "water".

(Response) Corrected (line 425).

Line 496: "carry oriT" should be "carry an oriT".

(Response) Corrected (line 493-434).

Line 498: "identify oriT" should be "identify an oriT".

(Response) Corrected (line 495).

Line 502: "MDR conjugative plasmids" should be "conjugative MDR plasmids".

(Response) Corrected (line 499).

S5 Fig B legend. This legend is inadequate. It should at least say it is a Southern blot and give some insight into the probe used - can refer to S5 Fig A legend where it is shown (but not described). Likewise for the S5 Fig C legend.

(Response) We have increased information of S5 Fig B C legend, and referred to panel A as follows (line 911-918). 

(B) Southern blots of the pSEA2-free Vibrio strain LN95 and the parental strain 04Ya249. Genomic DNA was double digested with NdeI and HindIII (the left two lanes) or NdeI and SphI (the right two lanes). The probe used was 5’ end of intA. Four unique bands originate from fragments shown in the first two rows (04Ya249 pSEA2, 04Ya249 chr1) in panel A. (C) Southern blots of 19 E. coli transconjugants obtained from 19 independent mating assays. Upper panel shows digestion with NdeI and SphI. Lower panel shows digestion with NdeI and HindIII. The color of the strain name indicates the pattern of SE/pSEA2 insertion deduced from restriction maps in panel A

---

## [Editor Report · Decision Letter 3]

6 Jul 2022

Atypical integrative element with strand-biased circularization activity assists interspecies antimicrobial resistance gene transfer from Vibrio alfacsensis

PONE-D-21-18582R3

Dear Dr. Nonaka,

We’re pleased to inform you that your manuscript has been judged scientifically suitable for publication and will be formally accepted for publication once it meets all outstanding technical requirements.

Kind regards,

Axel Cloeckaert

Academic Editor

PLOS ONE
---

## [Editor Report · Acceptance letter]

22 Jul 2022

PONE-D-21-18582R3 

Atypical integrative element with strand-biased circularization activity assists interspecies antimicrobial resistance gene transfer from *Vibrio alfacsensis*

Dear Dr. Nonaka:

I'm pleased to inform you that your manuscript has been deemed suitable for publication in PLOS ONE. Congratulations! Your manuscript is now with our production department. 

Kind regards, 

on behalf of

Dr. Axel Cloeckaert 

Academic Editor

PLOS ONE